# SPACEL: deep learning-based characterization of spatial transcriptome architectures

Hao Xu[1,6], Shuyan Wang[2,3,6], Minghao Fang[2], Songwen Luo[1], Chunpeng Chen[1], Siyuan Wan[2,3], Rirui Wang[1], Meifang Tang[1], Tian Xue [4], Bin Li [5] ✉, Jun Lin [1,2] ✉ & Kun Qu [1,2,3] ✉

Spatial transcriptomics (ST) technologies detect mRNA expression in single cells/spots while preserving their two-dimensional (2D) spatial coordinates, allowing researchers to study the spatial distribution of the transcriptome in tissues; however, joint analysis of multiple ST slices and aligning them to construct a three-dimensional (3D) stack of the tissue still remain a challenge. Here, we introduce spatial architecture characterization by deep learning (SPACEL) for ST data analysis. SPACEL comprises three modules: Spoint embeds a multiple-layer perceptron with a probabilistic model to deconvolute cell type composition for each spot in a single ST slice; Splane employs a graph convolutional network approach and an adversarial learning algorithm to identify spatial domains that are transcriptomically and spatially coherent across multiple ST slices; and Scube automatically transforms the spatial coordinate systems of consecutive slices and stacks them together to construct a 3D architecture of the tissue. Comparisons against 19 state-of-the-art methods using both simulated and real ST datasets from various tissues and ST technologies demonstrate that SPACEL outperforms the others for cell type deconvolution, for spatial domain identification, and for 3D alignment, thus showcasing SPACEL as a valuable integrated toolkit for ST data processing and analysis.

Spatial transcriptomics (ST) technologies have enabled researchers to detect the spatial distribution of, in principle, the entire transcriptome in histological tissue slices and thereby substantially improved our understanding of organ architecture[1–5] and disease microenvironments[6–10]. There are two broad categories of experimental ST techniques that can either i) detect the expression of a partial transcriptome at single cell resolution or ii) detect an entire transcriptome without single-cell resolution. ST techniques based on in situ hybridization and fluorescence microscopy (image-based), such as seqFISH[11], osmFISH[12], and MERFISH[13] probe only hundreds to thousands of transcripts in a slice, but can achieve single cell and even subcellular resolution. ST techniques based on next-generation sequencing (seq-based), such as 10X Visium[14], Slide-seq[15,16] and Stereo-seq[4] detect the expression of the whole transcriptome, but the resolution is restricted by the size of spots in the ST slices. Owing to limitations of these experimental ST techniques, many analytical methods have been developed to impute the undetected transcripts on each slice[17–20] and/or to deconvolute the cell types in each spot[21–29], aiming to detect an entire transcriptome while retaining single-cell resolution.

[1]Department of Oncology, The First Affiliated Hospital of USTC, School of Basic Medical Sciences, Division of Life Sciences and Medicine, University of Science and Technology of China, Hefei 230027, China. [2]Institute of Artificial Intelligence, Hefei Comprehensive National Science Center, Hefei 230088, China. [3]School of Data Science, University of Science and Technology of China, Hefei 230027, China. [4]Division of Life Sciences and Medicine, University of Science and Technology of China, Hefei 230027, China. [5]National Institute of Biological Sciences, Beijing 102206, China. [6]These authors contributed equally: Hao Xu, Shuyan Wang. ✉e-mail: libin@nibs.ac.cn; linjun7@ustc.edu.cn; qukun@ustc.edu.cn

A recent computational task for ST data analysis is the identification of clusters of cells/spots that are coherent in both transcriptome expression and spatial coordinates—i.e., dissecting the so-called spatial domains to characterize the spatial architecture present within an ST slice. Several methods have been developed to accomplish this task. For example, BayesSpace employs a Bayesian model and a Markov chain Monte Carlo approach to improve the resolution of ST data to the subspot level and then identifies spatial domains by clustering these subspots[30]. SpaGCN adopts a graph convolutional network to integrate the transcript expression level data with their spatial location information to identify spatial domains[31]. STAGATE uses an adaptive graph attention auto-encoder model to learn the similarities between the neighboring spots to identify spatial domains[32]. stLearn leverages a Louvain clustering algorithm and k-d tree nearest neighbor searching to identify spatial domains in ST slices[33]. The above-mentioned methods were designed for analyzing single ST slices.

The current trends for the rapid development and application of ST technologies suggest that a large number of ST slices from diverse tissues (and conditions) will be generated in scientific and medical studies for the foreseeable future[13,34–37]. Accordingly, there is an urgent need for computational tools that can quickly and efficiently implement integrated analysis of data from multiple ST slices. STACI[38], PRECAST[39], and STAligner[40] aimed to identify spatial domains jointly across multiple ST slices. These methods utilize gene expression as the input for their models, specifically, STACI adopts an over-parameterization approach, PRECAST uses simple projections of the batch effects onto the space of biological effects and STAligner employs triplet-loss for training. However, it has been reported that cell-type distribution is more robust for identifying uniform domains across multiple slices[9].

Accurately identifying functional spatial domains across multiple slices and reconstructing the 3D architecture of tissues offer invaluable opportunities for significant biological discoveries in various practical applications. The PASTE[41] and STAligner[40] methods made an effort to address these difficulties by integrating multiple slices into a single center slice, and/or aligning the consecutive slices to construct a 3D architecture of the tissue. PASTE assumes complete 2D overlap of slices, which restricts its applicability to cases where slices have only partial overlap. STAligner relies on selected landmark domains shared across slices for alignment and does not utilize correspondences for each individual spot in adjacent slices. As a result, these approaches can encounter misalignment issues in accurately capturing the global structure. Overall, constructing a stacked 3D alignment of the tissue still remains a great challenge for ST datasets with substantial defects.

Here, we developed SPACEL, a deep-learning-based toolkit comprising three modules—Spoint, Splane, and Scube—covering three analysis tasks for ST data (Fig. 1a). Spoint predicts the cell type composition of spots obtained by seq-based ST technologies such as 10X Visium. Splane jointly analyzes multiple ST slices and identifies spatial domains based on the cell type composition and spatial coordinate information of spots/cells. Scube aligns multiple consecutive slices and constructs a stacked 3D architecture of the tissue based on the coordinates of the spatial domains identified by Splane. We applied SPACEL to analyze 11 ST datasets comprising 156 slices acquired using the 10X Visium, STARmap, MERFISH, Stereo-seq, and Spatial Transcriptomics[1] technologies (Supplementary Data 1). SPACEL outperformed other state-of-the-art methods for each of the three examined analysis tasks and thus represents a valuable integrated toolkit for ST data processing and analysis.

## Results

### Workflow of SPACEL
The full architecture of SPACEL is illustrated in Fig. 1a, including the modules Spoint, Splane and Scube. Our motivation for developing the Spoint module is the need to perform cell type deconvolution on ST slices generated by seq-base ST technologies, which is also required for subsequent analysis of SPACEL. Spoint embedded a multiple-layer perceptron (MLP) with a probabilistic model to deconvolute the cell type composition of each ST spot (Fig. 1b, Methods). Spoint leverages a combination of simulated pseudo-spots, neural network modeling, and statistical recovery of expression profiles, which allows it to provide a more robust and accurate framework for estimating cell-type proportions in real ST data. For image-based ST data (i.e., single-cell resolution), users can simply use single-cell analysis tools (e.g., Seurat[42] and Scanpy[43]) to cluster cells and use marker genes to identify the cell type of each cell cluster.

Splane employs a graph convolutional network (GCN) approach[44,45] and an adversarial learning algorithm[46] to identify spatial domains by jointly analyzing multiple ST slices (Fig. 1c, Methods). First, for each ST slice, Splane calculates an adjacency matrix of cells/spots based on their distance in space. Then, Splane constructs a GCN model from the adjacency matrix and the cell type composition of cells/spots, and employs an adversarial learning algorithm[46] to learn the latent features shared across all of the analyzed ST slices; we term this a joint analysis scheme. Next, Splane applies a K-means clustering algorithm[47] to aggregate cells/spots with similar patterns of the shared latent features, where each of these cell/spot clusters is referred to as a spatial domain. All existing spatial-domain identification tools commonly utilize gene expression as input for their analyses and most of them follow a single analysis scheme. Splane distinguishes itself by utilizing cell-type composition as the input and adopting a joint analysis scheme. Furthermore, Splane incorporates an adversarial training to explicitly tackle and eliminate batch effects across multiple slices.

For ST datasets containing consecutive slices, Scube aligns the Splane predicted coordinate systems for each pair of adjacent slices and then constructs a stacked 3D architecture of the tissue, which is fully automated and does not require any manual alignment (Fig. 1d, Methods). Briefly, Scube builds a mutual nearest neighbor (MNN) graph between the cells/spots of two adjacent slices based on the coordinate information of spots and constructs an alignment objective function between them. This function serves as the foundation for aligning the slices and treats the alignment task as an optimization problem. Unlike existing tools like PASTE or STAligner, Scube is specifically designed to handle partially overlapped slices by incorporating a penalty term that accounts for the proportion of overlapped spots in adjacent slices. Scube then employs a differential evolution optimization algorithm[48] to search for the optimal translation vectors and rotation angles for each slice, and aligns them by transforming the coordinates of every slice accordingly. Finally, Scube stacks the transformed slices to construct the 3D architecture of the tissue.

In summary, Spoint, Splane, and Scube accomplished three analysis tasks of ST data in turn: cell type deconvolution, spatial domain identification, and 3D architecture construction. We have carefully designed each module of our method in an innovative way to address the potential limitations of current state-of-the-art methods (Supplementary Fig. 1). Specifically, Spoint employs a statistical model, a pseudo-spot simulation, a deep learning technique, and an elimination of variation between reference and ST data. In contrast, other tools often miss one or more of these features. Splane is unique in its combination of a cell type composition as input − a feature not utilized by any other tools, and adversarial training in the GCN model − a common approach first been employed to mitigate batch effects in ST data. Scube employs a unique global optimization strategy for 3D alignment, setting it apart from its peers. While each of these modules can operate independently, they are also optimized to work seamlessly in a unified workflow. This adaptability enables SPACEL to analyze data from diverse experimental platforms while simultaneously providing an all-encompassing and streamlined solution for ST data interpretation.

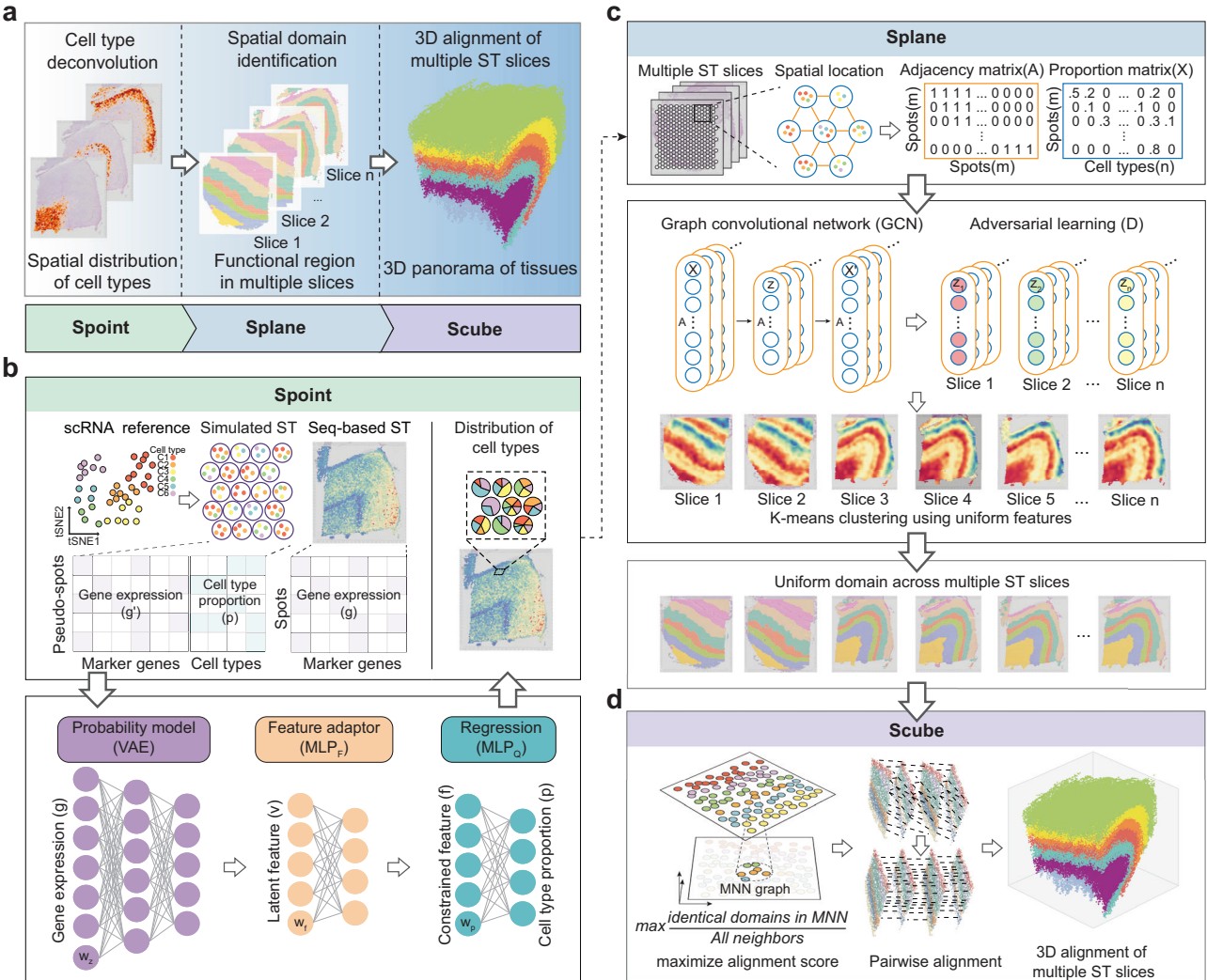

**Fig. 1 | Workflow of SPACEL. a** Three modules of SPACEL: Spoint, Splane, and Scube. **b** Spoint deconvolutes cell types of spots using an MLP and a probability model. MLP, multiple-layer perceptron; VAE, variational autoencoder. **c** Splane employs a GCN model and an adversarial learning algorithm to identify spatial domains across multiple slices. GCN, graph convolutional network. **d** For consecutive slices, Scube adopts a mutual nearest neighbor (MNN) graph and the differential evolution algorithm to transform slices and construct a stacked 3D alignment of a tissue.

To ensure the robustness of SPACEL, we conducted extensive experiments to evaluate its performance across various hyperparameter settings. Our results demonstrate that Spoint, Splane, and Scube exhibit superior robustness to hyperparameter variation compared to other state-of-the-art methods (Supplementary Fig. 2a–c), highlighting the effectiveness of SPACEL in providing reliable and consistent results across different experimental settings.

**Spoint accurately deconvolutes cell-type composition**

To construct the training set for Spoint and other deconvolution methods, we simulated pseudo-spots using the scRNA-seq dataset by assuming a normal distribution of the number of cells and the number of cell types per spot. Our simulations generated pseudo-spots similar to real ST data (MERFISH data from human brain tissue[49]; Supplementary Fig. 2d–k). We compared the performance of Spoint with state-of-the-art methods for cell type deconvolution, including Cell2location, SpatialDWLS, RCTD, STRIDE, Stereoscope, Tangram, DestVI, Seurat, SPOTlight, and DSTG using 32 simulated datasets from a benchmark study[50] (Fig. 2a, Supplementary Data 2). We adopted Pearson's correlation coefficient (PCC) and structural similarity index measure (SSIM) to assess the similarity between the predicted and true cell type compositions, and used root mean square deviation (RMSE) and Jensen-Shannon divergence (JSD) to assess the error of each method. Spoint yielded the highest average PCC/SSIM values (= 0.73/0.69), and the lowest average RMSE/JSD values (=0.05/0.41) among the 11 deconvolution methods (Fig. 2a). Additionally, we applied the accuracy score (AS) defined in the benchmark study to evaluate the performance of each method: the average AS of Spoint (= 0.93) was obviously higher than that of the other methods (AS = 0.24−0.82; Fig. 2b).

To evaluate Spoint's performance in cell type deconvolution on real ST experimental data, we used two datasets as input: a 10X Visium dataset for the human dorsolateral prefrontal cortex (DLPFC) that contains 12 ST slices[34] and a human brain scRNA-seq dataset downloaded from the Allen Brain Map[51]. As there is no experimental evidence for the cell type composition of each spot in this dataset, we took the cortical layers annotated by the original studies as ground truth (Fig. 2c, Supplementary Fig. 3a−c). As one example, we found that the distribution of excitatory L3/4 neurons predicted by Spoint had a higher AS value (= 0.85) than the other examined deconvolution methods (AS = 0.29−0.64; Fig. 2c). Collectively, for all 56 cell types[52,53] in all 12 ST slices, the average AS of Spoint (= 0.60) was again higher than those of the other methods (average AS = 0.30−0.48; Fig. 2d). Furthermore, we calculated the significance (using the Wilcoxon Rank

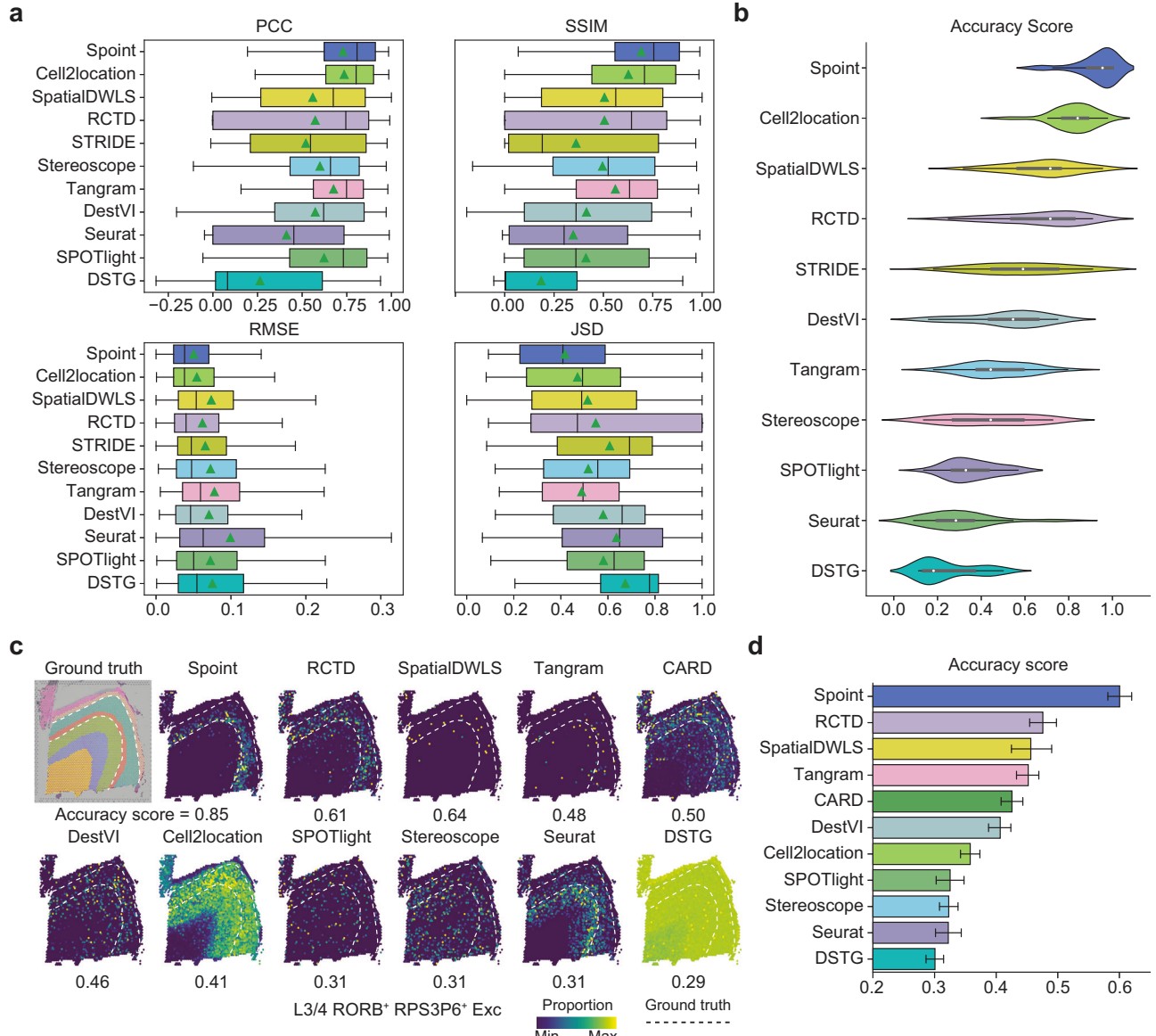

**Fig. 2 | Cell type deconvolution of spots using Spoint and other deconvolution methods. a** Average PCC, SSIM, RMSE, and JSD values of the deconvolution methods for 32 simulated datasets from the benchmark study[50]; PCC, pearson's correlation coefficient; SSIM, structural similarity index measure; RMSE, root mean square error; JSD, Jensen-Shannon divergence; center line, median value; box limits, upper and lower quartiles; whiskers, 1.5× interquartile range; *n* = 32 datasets. **b** Accuracy scores of the deconvolution methods for the 32 simulated datasets.

Center line, median value; box limits, upper and lower quartiles; whiskers, 1.5× interquartile range; *n* = 32 datasets. **c** Spatial distributions of excitatory layer 3/4 RORB⁺RPS3P6⁺ neurons predicted by the deconvolution methods for the DLPFC dataset. **d** Accuracy scores of the deconvolution methods for the DLPFC dataset. Bar height, mean value; whiskers, mean values ± 95% confidence intervals; *n* = 56 cell types. Source data are provided as a Source Data file.

Sum test) of any difference in the proportion of one cell type between the layers, with a lower *P*-value representing increased deconvolution accuracy[29]. We found that the average *P*-value of Spoint ( = 0.01) was lower than that of the other methods (average *P*-values = 0.05–0.64; Supplementary Fig. 3d).

To further evaluate Spoint's performance in real ST datasets, we collected three single-cell resolution ST datasets: Chen et al. mouse embryo brain (Stereo-seq)[4], Chen et al. mouse brain (Stereo-seq)[4] and Fang et al. human brain (MERFISH with 4000 genes)[49] to simulate spot-level ST data with known cell-type composition and spatial context. We also obtained the corresponding scRNA-seq data from the same tissue types as reference. Using these data, we aggregated approximately 10 cells into each pseudo spot, creating three distinct spot-level ST datasets (Supplementary Fig. 4a). As an example, we observed that Spoint achieved the highest PCC compare to other

cell type deconvolution methods for forebrain glutamatergic neuroblast in dataset 1 (Supplementary Fig. 4b). We then evaluated the deconvolution performance of the Spoint module on all cell types across the three datasets using four evaluation metrics. Our findings demonstrate that Spoint consistently outperformed existing methods, achieving the highest accuracy scores in all three datasets (Supplementary Fig. 4c, d). In addition, we evaluated the relative error between the summed predicted cell type proportions and the ground truth, and found that SpatialDWLS (with rankings of 1, 2, and 3) and Spoint (with rankings of 4, 3, and 1) registered the lowest average relative error across the three datasets (Supplementary Fig. 4e). These results provide strong evidence of the Spoint module's superior performance over other methods, and an accurate deconvolution result will facilitate subsequent identification of spatial domains using Splane.

## Splane identifies spatial domains for multiple slices

After cell types were deconvoluted with Spoint, we applied Splane to identify the spatial domains of the aforementioned ST slices of the DLPFC dataset (Supplementary Fig. 5). We used the original study's manual annotation of cortical layers as the ground truth[34], and used three metrics to evaluate the prediction performance. Specifically, we used the Jaccard index (JI) and Adjusted Rand Index (ARI) between a spatial domain identified by Splane and the corresponding cortical layer from the ground truth to assess the accuracy of the method, and computed the shifting distance (SD) between the prediction and the ground truth to quantify the error of Splane (see Methods). A higher JI/ARI value or a lower SD value indicates that the prediction is closer to the ground truth.

First, we compared the results obtained by processing all slices jointly (joint analysis, by default) to those obtained by processing each slice individually (single analysis). To enable this comparison, we built a test version of Splane, namely, Splane-single, that constructs a GCN model for each slice individually. Taking slices 151508, 151510, 151670, and 151673 from the DLPFC dataset as examples, the JI values obtained by the joint analysis were higher than those obtained by the single analysis, and the SD values obtained by the joint analysis were lower than those obtained by the single analysis (Fig. 3a, b). Moreover, upon processing the entire 12 slices dataset, the average JI/median ARI values by joint analysis were 0.61/0.61, higher than that by the single analysis (= 0.53/0.57; Fig. 3c, d); the average SD value by the joint analysis was 50 μm, much lower than that by the single analysis (= 262 μm; Fig. 3e). This comparison implies that the joint analysis scheme improves the accuracy of Splane for spatial identification.

We then compared the performance of Splane against other state-of-the-art spatial-domain-identification methods, including methods designed for analyzing multiple ST slices such as STAligner, PRECAST and STACI, as well as methods that are designed for analyzing single ST slice such as STAGATE, SpaGCN, BayesSpace, and stLearn, using the same evaluation criteria (Supplementary Data 2). In analyzing all 12 ST slices of the DLPFC dataset, Splane exhibited the highest accuracy (average JI/median ARI = 0.61/0.61) and the lowest error (average SD = 53 μm), while other tools had JI/ARI and SD ranges of 0.41–0.56/0.36–0.54 and 75–548 μm (Fig. 3a–e, Supplementary Fig. 6). Thus, Splane outperforms available alternative methods for spatial domain identification.

Spatial expression variation of genes can reflect the states, communications, and dynamics of cells, and therefore accurate identification of spatial variable genes (SVGs) from ST slices is crucial for determining the functions and phenotypes of cells in spatial domains. We thereby compared these methods for the task of identifying SVGs that represent specific cortical layers. We adopted the SVGs for cortical layers 1 ~ 6 and white matter (WM) annotated by the original study[34] as the ground truth, and evaluated the performance of each method by computing the overlap between the identified SVGs and the ground truth. Using a fold-change > 0.5 and a P-value < 0.01 as cut-offs for SVGs, we found that tools designed for analyzing multiple ST slices identified more SVGs compared to those designed for analyzing single ST slice. Splane, PRECAST, STAligner and STACI identified 1714, 1671, 1669 and 1527 out of 1917 SVGs of cortical layers 1 ~ 6 and WM, while the other examined methods identified only 179 ~ 1336 SVGs for each slice when the same cut-offs were applied (Fig. 3f). Similarly, using the receiver operating characteristic (ROC) curves as indicators for SVG prediction accuracy, we found that Splane, PRECAST and STAligner yielded the highest area under the curve value (AUC = 0.90, 0.89 and 0.88 respectively) among all examined methods (AUCs < 0.83; Fig. 3g). These results emphasize the power of the Splane's joint analysis scheme for identifying SVGs from multiple ST slices.

To assess the importance of the Spoint module in facilitating accurate spatial-domain identification using the Splane module, we conducted two comparative analyses on the human DFPLC datasets.

First, we compared the performance of Splane using three different input types: highly variable gene expression matrix, PCA reduction of highly variable gene expression matrix, and cell-type proportions predicted by Spoint. Results showed that using only the cell-type proportion predicted by Spoint as input yielded spatial-domain identification results that closely resembled the ground truth, with significantly higher JI values in all 12 slices compared to the other input types (Supplementary Fig. 7). Second, we evaluated the performance of Splane using cell-type deconvolution results from other methods as input. Results demonstrated that using the cell-type proportions predicted by Spoint yielded the best spatial-domain identification outcome (Supplementary Fig. 8). Notably, the identification of Layer4 was only possible when using the cell-type proportions predicted from Spoint as input.

## Splane identifies spatial domains from cancer slices

To test the performance of Splane in identifying spatial domains from disease slices, we applied Splane to jointly analyze 11 breast cancer 10X Visium slices from three experimental batches, including six slices reported in ref. 35, four slices released by 10X Genomics[54], and one slice from the study conducted in ref. 30. We first performed cell type deconvolution in each slice using Spoint and obtained the cell type identification (based on the expression of selected marker genes) and composition of each spot (Supplementary Fig. 9).

Subsequently, we applied Splane and identified ten spatial domains across the 11 slices (Fig. 4a). Notably, the extent of the batch effect between slices from different experiments was significantly reduced in Splane compared to that of the raw data and Splane without adversarial learning (Supplementary Fig. 10). We then explored the cell type composition of each predicted spatial domain and annotated these domains accordingly: domains D0-3, D4–6 and D7–9 were defined as tumor (tumor cell-enriched), intermediate (mixed cell types) and immune (immune cell-enriched) domains, respectively (Supplementary Fig. 11a–d, Supplementary Data 3).

We then applied two distinct approaches to assess the accuracy of the Splane predictions for the tumor and immune domains. Since tumor cells in breast cancer patients often bear chromosomal copy number variations (CNVs)—including chromosome 1q and 8q gains and/or chromosome 1p losses[55,56]—our first approach was to calculate the CNVs of each ST spot and check for enrichment of CNVs in the predicted tumor domains. Using inferCNV[57], we calculated the CNVs for all spots in all 11 ST slices from the expression matrix of the ST data. Taking slice 11 as an example, the four tumor domains predicted by Splane had copy number gains in chromosomes 1q and 8q (inferred CNV ≥ 0.2) as well as copy number losses for chromosome 1p (inferred CNV ≤ −0.2; Fig. 4b, c). In contrast, fewer or no copy number gains or losses were detected in the Splane-predicted intermediate or immune domains (inferred CNV = −0.2 ~ 0.2). Taking all 11 slices into account by the average CNV values, we also observed chromosome 1q and 8q gains (average CNV > 0.2) and chromosome 1p losses (average CNV < −0.2) in tumor domains, but less in intermediate and none in immune domains (Fig. 4d, e).

Our second approach was to calculate the percentage of experimentally annotated immune spots and check for enrichment of immune spots in spatial domains. We took the H&E-staining-marked lymphocyte-enriched spots in slice S1, S2, S5, and S6 from ref. 35 and the CD3⁺ immunofluorescence(IF)-marked T cell-enriched spots in slice S10 from ref. 30 as the ground truth for immune spots. We then calculated the percentage of immune spots in each of the Splane identified domains. We found higher percentages of H&E-staining-marked immune spots in immune domains D9, D8, and D7 in slice S1, S2, S5, and S6 (44%, 8%, and 6%, respectively) than in the intermediate and tumor domains (< 1%) (Fig. 4f, g). Similarly, in slice S10, we found more CD3⁺ IF-marked immune spots in immune domains D9, D8, and D7 (72%, 70%, and 63% respectively) than in intermediate and tumor

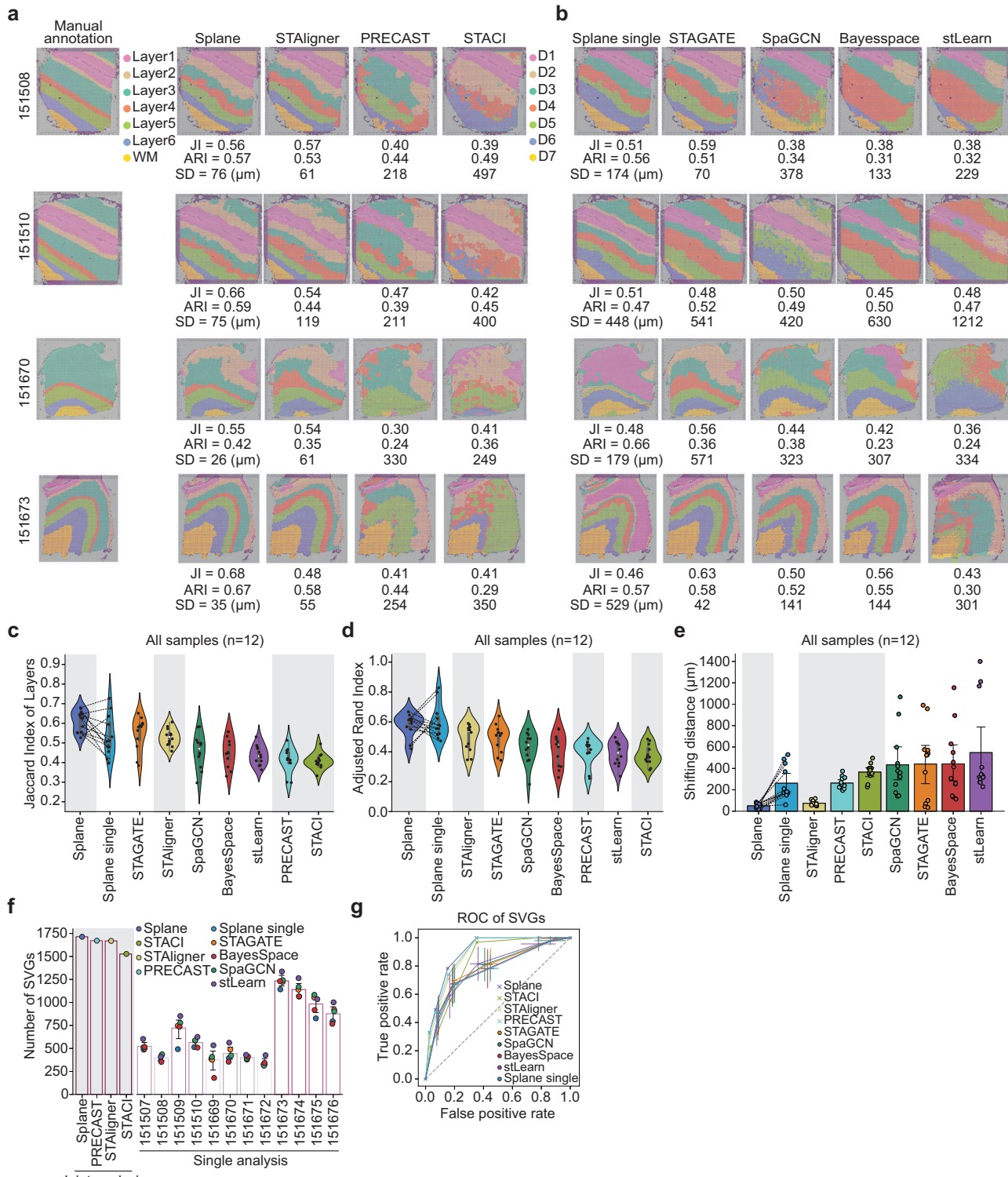

**Fig. 3 | Identification of spatial domains from 12 10X Visium slices of DLPFC. a** Comparison of spatial domains identified by Splane, STAligner, PRECAST, and STACI for slice 151508, 151510, 151670, and 151673. Layer1-Layer6, cortical layer 1 - 6; WM, white matter; JI, Jaccard index; SD, shifting distance. **b** Spatial domains identified by Splane-single, STAGATE, SpaGCN, BayerSpace, and stLearn for the four slices. **c**–**e** Jaccard indexes (**c**), Adjusted Rand Indexes (**d**), and shifting distances (**e**) between cortical layers and corresponding spatial domains identified by Splane, STAligner, PRECAST, STACI, Splane-single, STAGATE, SpaGCN, BayesSpace, and stLearn. The gray background represents for the methods for multiple slice analysis. Center line, median value; bar height, mean value; box limits, upper and lower quartiles; whiskers, 1.5× interquartile range for box plots, mean values ± 95% confidence intervals for bar plots; *n* = 12 slices. **f** Proportion of SVGs identified by the spatial-domain-identification methods when using fold-change > 0.5 and *P*-value < 0.01 (two-sided Wilcoxon rank-sum test) as cut-offs. The gray background represents the methods used for multiple slice analysis. SVGs, spatial variable genes; bar height, mean value; whiskers, mean values ± 95% confidence intervals; *n* = 5 methods for single analysis. **g** Receiver operating characteristic (ROC) curves of SVGs identified by Splane, STAligner, PRECAST, STACI, Splane-single, STAGATE, SpaGCN, BayesSpace, and stLearn. Whiskers, standard errors; *n* = 12 slices. Source data are provided as a Source Data file.

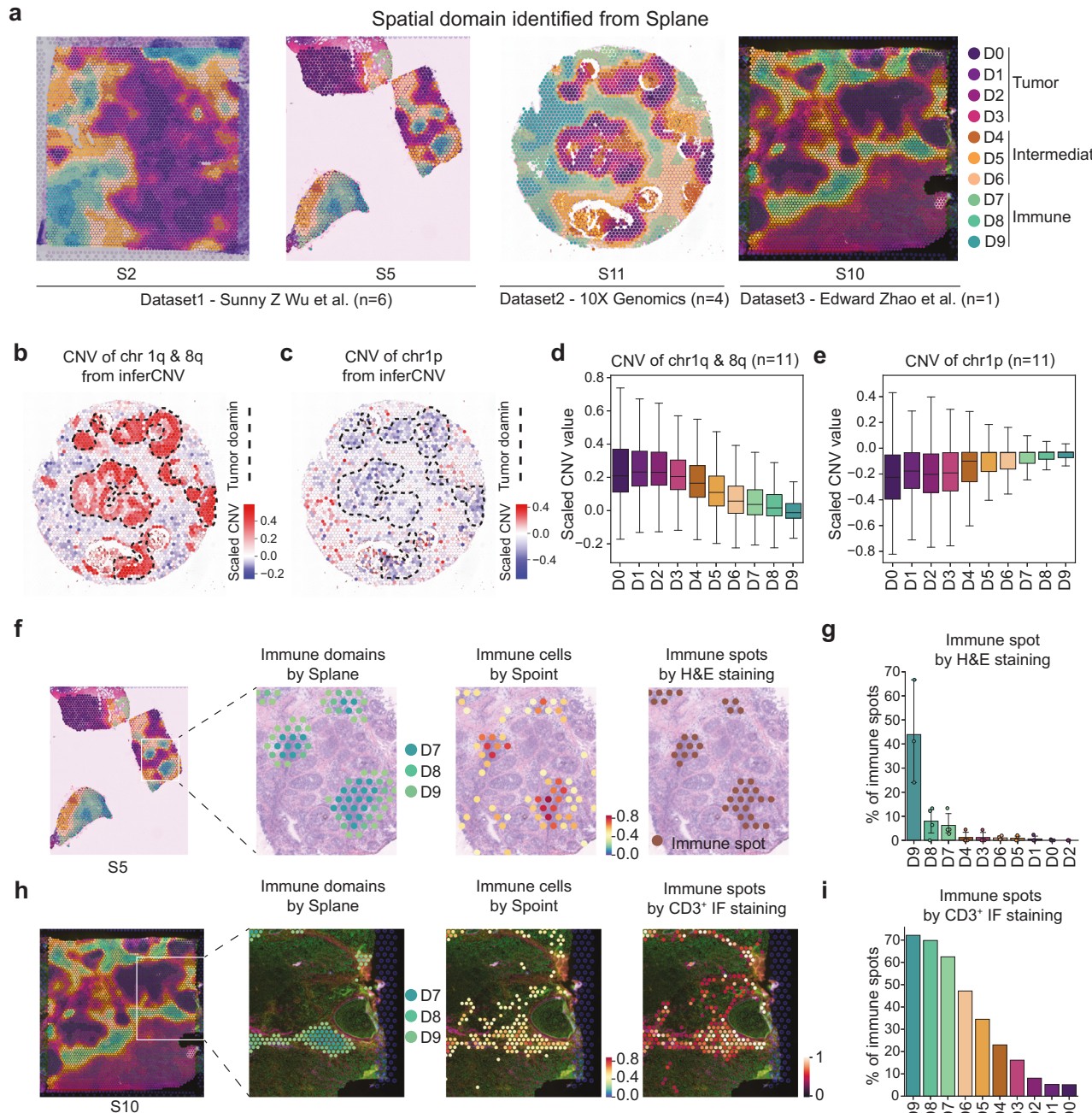

**Fig. 4 | Joint analysis of 11 breast cancer ST slices from three different datasets. a** Spatial domains identified by Splane in slice S2 and S5 from Wu et al. dataset, slice S10 from Zhao et al. dataset, and slice S11 released by 10X Genomics. **b, c** Spatial distribution of chromosome 1q&8q copy number gains (**b**) and 1p copy number losses (**c**) of ST spots in slices S11, calculated by inferCNV. Dashed lines represent the tumor domain. **d, e** CNVs of chromosome 1q & 8q (**d**) and chromosome 1p (**e**) in each spatial domain calculated by inferCNV. CNVs, copy number variations; center line, median value; box limits, upper and lower quartiles; whiskers, 1.5× inter-quartile range; $n = 11$ slices. **f** From left to right: Splane predicted spatial domains in slice S5, distribution of Splane predicted immune domains D7/D8/D9, distribution

of Spoint predicted immune cells, and distribution of H&E staining marked immune spots. **g** Percentage of H&E staining marked immune spots in each domain of slice S1, S2, S5, and S6. The four slices were H&E stained in the original study. Bar height, mean value; whiskers, mean values ± 95% confidence intervals; $n = 4$ slices. **h** From left to right: Splane predicted spatial domains in slice S10, distribution of Splane predicted immune domains D7, D8, and D9, distribution of Spoint predicted immune cells, and distribution of CD3⁺ immunofluorescence (IF) staining marked immune spots. **i** Percentage of CD3⁺ IF staining marked immune spots in each domain of slice S10. Source data are provided as a Source Data file.

domains ( < 47%) (Fig. 4h, i). These results support Splane's prediction of tumor and immune domains.

In the context of tumor systems, accurately identifying tumor boundaries is crucial for immuno-oncology research. We compared the ability of Splane and other multi-slice analysis algorithms, including STAligner, PRECAST, and STACI, to identify tumor boundaries. In the human breast cancer 10X Visium data, only Splane accurately

identified tumor regions with distinct boundaries (Supplementary Fig. 12a). The tumor regions predicted by Splane were consistent with CNV gains or losses in specific chromosomes, and they aligned well with evident tumor tissue regions in H&E images. Furthermore, we observed a significant correlation between the proportion of tumor cells and CNV scores specifically in the spatial domains identified by Splane (Supplementary Fig. 12b). These results highlight Splane's

ability to identify consistent tumor regions and boundaries across multiple slices in tumor systems.

## Scube constructs 3D architecture of tissue from consecutive 2D slices

Most current experimental ST techniques measure the distribution of transcripts in 2D space, yet tissues are obviously 3D, and functions of cells are distributed in space. We thereby developed the Scube module of SPACEL to construct and investigate the 3D architecture of a given tissue. We first employed a 3D STARmap dataset of the mouse brain from ref. 58 (originally $1400 \mu m \times 1700 \mu m \times 100 \mu m$) and divided it into ten slices of $10 \mu m$ thickness each as the ground truth (Fig. 5a). We randomly rotated, flipped, and cropped each slice to simulate a perturbed ST dataset comprising consecutive 2D slices (Fig. 5b, see Methods). We defined the percentage of a slice that was randomly cropped out as the crop ratio of that slice. At each level of crop ratio, we used Splane to identify the spatial domains across the ten slices, and applied Scube to align these slices by transforming their coordinate systems based on the Splane-identified spatial domains.

We adopted SSIM and PCC as the metrics to assess the similarity between the transformed coordinate system and the ground truth for each slice. Higher SSIM and PCC values indicate better alignment. Recall that STAligner and PASTE are two state-of-the-art methods for constructing 3D alignments of tissue from consecutive ST slices[40,41], we compared the performance of Scube with STAligner and PASTE for aligning the simulated dataset under crop ratios from 0 to 0.25. Since the alignment results of STAligner and PASTE can be affected by their hyperparameters, we first performed a parameter optimization search to achieve the best alignment results of STAligner and PASTE (Supplementary Fig. 13), before it was compared with Scube. With a crop ratio equal to 0.25, the average SSIM/PCC values of the Scube-transformed slices were 0.96/0.97, whereas the values for the STAligner-transformed slices were 0.76/0.77 and for the PASTE-transformed slices were 0.72/0.75. (Fig. 5c). With crop ratios range = 0.10, 0.15, 0.20, and 0.25, the average SSIM values of the Scube-transformed slices were 0.98, 0.98, 0.97 and 0.96, higher than those by STAligner (= 0.79, 0.79, 0.72, and 0.76) or PASTE (= 0.82, 0.77, 0.74, and 0.72; Fig. 5d). Similarly, the average PCC values of the Scube-transformed slices were 0.99, 0.98, 0.98, and 0.97, which were also higher than those by STAligner (= 0.80, 0.81, 0.73, and 0.77) or PASTE (= 0.85, 0.79, 0.76, and 0.75; Fig. 5d), suggesting a better alignment in Scube for the simulated dataset.

To further compare the performance of Scube, STAligner and PASTE in constructing a 3D architecture on real ST dataset, we examined a mouse primary motor cortex (MOp) profile, which contains 33 consecutive 2D slices from the MERFISH experiment[13]. We first applied Splane and identified seven spatial domains from the 33 consecutive 2D slices, which exhibited the highest accuracy in spatial domain identification (average JI/median ARI = 0.44/0.43) compared to STACI (average JI/median ARI = 0.43/0.38), STAligner (average JI/median ARI = 0.33/0.25) and PRECAST (average JI/median ARI = 0.23/0.16; Supplementary Fig. 14). We then used Scube to align these slices by transforming the coordinate systems of each pair of adjacent slices (Supplementary Fig. 15a, b). Next, by stacking the transformed 2D slices together, we constructed a 3D architecture of MOp in Scube, enabling a 3D illustration of spatial domains and cell types (Fig. 5e).

In parallel, we applied STAligner and PASTE to perform an alignment and 3D construction from the same dataset (Supplementary Fig. 15). Since no ground truth is known in this case, we calculated the SSIM/PCC values between every two adjacent transformed slices to evaluate the alignment performance of the two methods, and found that the average SSIM/PCC values of the Scube-transformed slices were 0.71/0.76, significantly higher than those by STAligner (SSIM/PCC = 0.61/0.65) or PASTE (SSIM/PCC = 0.57/0.61; Fig. 5f). Notably,

when stacking the 2D transformed slices using either STAligner or PASTE, we obtained a twisted 3D architecture of MOp (Fig. 5e).

SPACEL also integrated a Gaussian process regression (GPR) model[5,59] in Scube, which predicts the expression level of a gene at any position in the 3D architecture, enabling a continuous illustration of transcript distribution along any direction in space (Fig. 5g, Supplementary Fig. 16a, b). By quantifying the Bayes Factor (BF, see Methods) of the GPR model, SPACEL is capable of identifying genes that vary significantly in any direction within the 3D space of tissue (Fig. 5h). SPACEL thereby allows users to explore the dynamics of transcript distributions from any direction, and hence reveals a real 3D structure of the spatial architecture of complex tissues or organs.

Accurate reconstruction of detailed 3D structures is crucial for unraveling biological phenomena in spatial omics data. In the mouse embryo Stereo-seq data[4], Scube achieved better performance compared to STAligner and PASTE in accurately reconstructing and depicting structures like brain, liver and paws (Supplementary Fig. 17a, b). Furthermore, Scube (SSIM/PCC = 0.62/0.66) exhibited higher consistency in the distribution of tissue regions between slices compared to STAligner (0.57/0.61) and PASTE (0.24/0.25) (Supplementary Fig. 17c). In summary, the Scube module of SPACEL outperformed STAligner and PASTE in alignment and 3D architecture construction in both the simulated (STARmap) and real (MERFISH and Stereo-seq) datasets.

## SPACEL as an integrated toolkit for ST data processing and analysis

To highlight the integrated nature of SPACEL, we applied a fully integrated workflow to analyze mouse whole brain ST data[60]. The dataset consisted of 75 consecutive slices generated by Spatial Transcriptomics technology[1] with a spot resolution of $100 \mu m$, covering a large portion of the mouse brain region. Using the original brain region annotations as the reference ground truth, we evaluated the performance of each module. In Spoint, the predicted distribution of major region-specific cell types matched well with the corresponding brain regions, such as excitatory neurons and hippocampus CA1 and CA3 cells predominantly predicted in the hippocampal region (Supplementary Fig. 18a, b). In Splane, we compared its performance against STAligner, PRECAST and STACI. The average JI/median ARI values of the spatial domains identified by Splane (0.42/0.56) were higher than those identified by STAligner (0.36/0.40), PRECAST (0.35/0.38) and STACI (0.31/0.40) across the 75 slices (Supplementary Fig. 18c, d). In Scube, the 3D tissue reconstruction achieved by Scube (SSIM/PCC = 0.83/0.85) exhibited high coherence and accuracy, surpassing the results obtained by PASTE (0.82/0.84) and STAligner (0.79/0.81) (Supplementary Fig. 18e, f). These results clearly demonstrate that SPACEL serves as an effective integrated toolkit for analyzing multiple ST slices.

## Discussion

In this study, we introduced a deep learning-based toolkit, SPACEL, comprising three modules: Spoint for cell type deconvolution, Splane for the identification of spatial domains across multiple ST slices, and Scube for the construction of 3D architecture from consecutive ST slices. We have demonstrated that the SPACEL modules outperform state-of-the-art methods for each of these tasks through analyses of 32 simulated and 11 real ST datasets acquired using five distinct ST technologies. While each module can be utilized separately, their synergistic interplay within the SPACEL platform offers an all-encompassing and streamlined solution for the best results of ST data interpretation, particularly accurate 3D tissue alignment, precise spatial domain identification, and effective batch effect removal.

Unlike the other state-of-the-art methods, the Splane module of SPACEL combines the cell type composition as input and adversarial training in the GCN model. Since the spatial distribution of genes, cells,

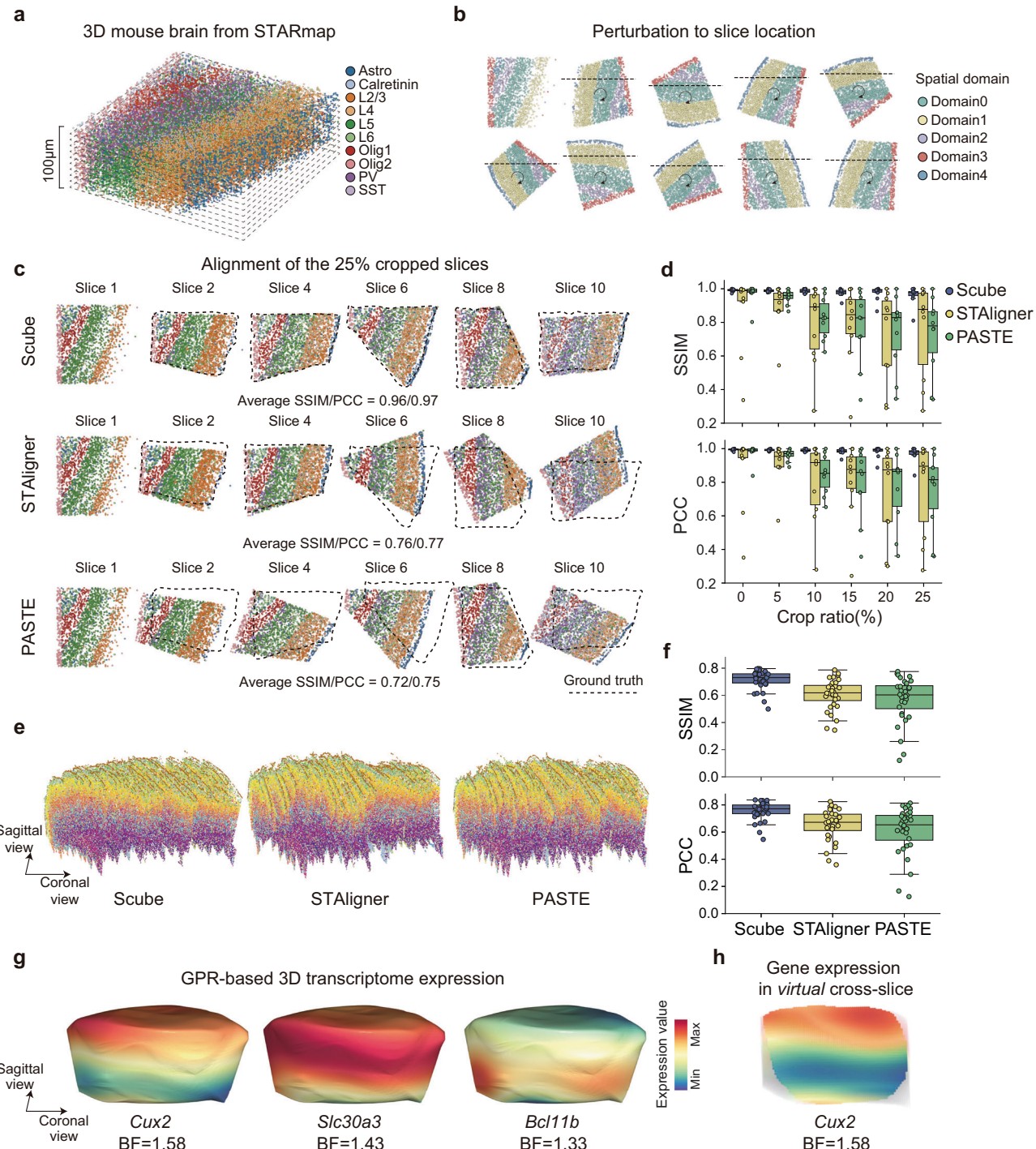

**Fig. 5 | 3D alignments of consecutive ST slices for the mouse brain. a** The 3D STARmap dataset of the mouse brain. **b** Simulating an ST dataset with consecutive 2D slices by dividing the 3D STARmap dataset into ten 2D slices and randomly rotating/flipping/cropping each slice. **c** Alignment of the simulated dataset with crop ratio=0.25, generated by Scube, STAligner, and PASTE. Dashed lines represent the positions of the ground truths. SSIM, structural similarity index measure; PCC, Pearson's correlation coefficient. **d** SSIM/PCC values of Scube's, STAligner's, and PASTE's results for simulated datasets with crop ratios from 0 to 0.25. Center line, median value; box limits, upper and lower quartiles; whiskers, 1.5× interquartile range; $n = 10$ slices. **e** Stacked 3D alignments constructed by Scube (left), STAligner

(middle), and PASTE (right) from 33 MERFISH slices of mouse primary motor cortex. **f** SSIM/PCC values of Scube's, STAligner's, and PASTE's alignment results for the MERFISH dataset. Center line, median value; box limits, upper and lower quartiles; whiskers, 1.5× interquartile range; $n = 33$ slices. **g** Continuous expression distribution of the genes *Cux2*, *Slc30a3*, and *Bcl11b* on the surface of the 3D architecture of the mouse brain after applying the GPR model in Scube. BF, Bayes factor. **h** Continuous expression distribution of *Cux2* in the coronal plane of the 3D architecture of the mouse brain. BF, Bayes factor. Source data are provided as a Source Data file.

and cell clusters from distinct ST experiments can be extremely heterogeneous, using cell type composition as input and introducing adversarial training to the GCN model significantly minimizes batch effects, leading to a more robust and efficient method for spatial domain identification. The joint analysis scheme that Splane incorporated is also particularly powerful to identify common features, and hence to accurately predict SVGs across multiple slices.

Spatial domains identified in Splane also set a foundation for Scube to achieve an accurate alignment of the consecutive slices and hence a precise 3D architecture construction. In contrast, the PASTE algorithm uses the raw expression matrix and spatial coordinates from the ST data as input for alignment, and a hyperparameter α to adjust the relative contributions of transcriptional dissimilarity and spatial distances among the aligned spots. This design makes PASTE's alignment results sensitive to hyperparameters in datasets with significant defects (e.g., slices with crop ratios ≥ 0.25). On the other hand, STAligner constructs MNN based on the raw expression matrix to select landmark spots/cells and relies on user-defined landmark domains shared across slices for alignment. However, this approach does not utilize complete information on each spot of the whole slice, which limits its ability to balance global structure similarity and alignment accuracy, particularly in the case of partially overlapped slices. In contrast, Scube adopts a global optimization strategy for 3D alignment, taking into account the correspondences between all spots in adjacent slices. This innovative method enables Scube to achieve more precise alignment, preserving the overall structural integrity in the process of spatial alignment.

To enhance the accessibility of the GPR model for 3D spatial transcriptomics, we have developed a user-friendly Python code by integrating the original MATLAB code and third-party GUI software. This integration enables researchers to easily access and utilize the model, fostering advancements in the field and promoting broader adoption of this technology within the scientific community. There are opportunities to improve the performance of SPACEL and extend its application. For example, Splane uses cell type composition information as input for the model; this means that to process seq-based ST datasets such as 10X Visium, one has to employ a cell type deconvolution algorithm such as Spoint to obtain the cell type composition information before Splane can be applied. Besides, the current version of Scube does not support nonlinear alignment as well as STAligner and PASTE. Although there are several tools that can nonlinearly align image data by setting anchor points manually, to our knowledge there is no method yet to do it automatically for ST data analysis. We acknowledge this limitation and plan to address this issue in a future version of Scube. Another issue for SPACEL is that when a new slice is added to the dataset, the entire deep learning model needs to be retrained from scratch, which may affect the computational efficiency of SPACEL to analyze large-scale datasets. We anticipate that transfer learning models and algorithms (e.g., ImageNet[61] and BERT[62]), which have been used in bioinformatics tasks such as cell type classification for large-scale single-cell data[63–66], can help us to overcome this limitation.

## Methods

### Construction of simulated ST data from scRNA-seq data

To establish training sets for Spoint and other deconvolution methods, we employed cells from a scRNA-seq dataset to simulate pseudo-spots, where the cell type composition of each pseudo-spot can be ascertained from its constituent cells. Specifically, assuming that the number of cells at each spot ($N_c$) follows a normal distribution $\mathcal{N}(\mu_c, \delta_c)$, and the number of cell types at each spot ($N_t$) follows a normal distribution $\mathcal{N}(\mu_t, \delta_t)$, we set the default parameters as $\mu_c = 10, \delta_c = 5, \mu_t = \mu_c/2, \delta_t = \delta_c/2$. Researchers can also detect $N_c$ from the image of an ST slice using stardist[67] or DeepCell[68], and use it as the input parameter. We first generated $N_t$ for each sampled spot, and

then sampled cells from the scRNA-seq data associated with each spot. Each cell was sampled based on its sampling probability which is determined by $P_t$. $P_t$ for each spot is defined as:

$$P_t = \begin{cases} f_t \ if \ 0 < r_c < \frac{1}{3} \\ \frac{1/f_t}{\sum_t (1/f_t)} \ if \ \frac{1}{3} \le r_c < \frac{2}{3} \\ \frac{\sqrt{f_t}}{\sum_t (\sqrt{f_t})} \ if \ \frac{2}{3} \le r_c < 1 \end{cases} \quad (1)$$

where $f_t$ is the composition of each cell type in the scRNA-seq data, and $r_c$ is a random value between 0 and 1. We then randomly sampled cells of each cell type from the scRNA-seq data, and combined the gene expression of $N_c$ sampled cells to generate an original spot. During the cell sampling process, we calculated the cell type numbers of these sampled cells. These cell type numbers serve as the cell type labels for the spot. We then estimated the mean value $\mu^L$ and standard deviation $\sigma^L$ of gene expression for each spot from a real ST data, and downsampled the original spot using the downsampleMatrix function of the Scuttle package[69] to obtain a simulated spot that has gene expression distribution similar to that of the real ST data.

### Deconvolution model of Spoint

Spoint contains three deep learning models. The first model is a variational autoencoder (VAE) model, consisting of an encoder layer, a decoder layer, and three hidden layers. The three hidden layers contain 128, 64, and 128 nodes, respectively. The input variables of the encoder and decoder layers are gene expression levels at each spot, $\vec{x}$, so the dimension of these two layers is equal to the number of detected genes. We inputted the expression matrices of the simulated ST data $\{\vec{x}'\}$ and the real ST data $\{\vec{x}\}$ into the VAE model, and used the latent variables of the middle hidden layer as the output results, $\vec{z}'$ and $\vec{z}$. Note that the simulated ST dataset should be constructed from a single-cell dataset obtained from the same type of tissue as the real ST dataset.

The second model $E$ consists of an input layer that accepts the latent variable $\vec{z}$ of the VAE model, three hidden layers each containing 512 nodes with ReLU activation function[70], and a output layer that outputs the proportion of each cell type in the spot with Softmax activation function. We adopted the maximum mean discrepancy (MMD) distance between the variable of model $E$'s last hidden layer for the simulated ST data and the real ST data as the objective function, as

$$\text{Loss}_M = \text{MMD}^2\left(E_2(\vec{z}), E_2(\vec{z}')\right) \quad (2)$$

$$\text{MMD}^2(U, V) = \frac{1}{n^2} \sum_i^n \sum_{i'}^n k(u_i, u'_i) - \frac{2}{nm} \sum_i^n \sum_j^m k(u_i, v_i)$$
$$+ \frac{1}{m^2} \sum_j^m \sum_{j'}^m k(v_i, v'_i) \quad (3)$$

$$k(u, v) = e^{\frac{-|u-v|^2}{\sigma}} \quad (4)$$

where $\vec{z}'$ and $\vec{z}$ are the latent variables of the simulated ST data and real ST data, respectively, generated by the VAE model, and $\sigma$ represents the dimension of the hidden layer. The third model $R$ consists of an input layer that accepts the output of model $E$, three hidden layers each containing 512 nodes with ReLU activation function[70], and a output layer is represents the recovery of latent variable $\vec{z}$ of the VAE model. Then, we used the following objective function to constrain the outputs of the model $E$ and the ground truth cell-type proportion of

simulated ST data:

$$\text{Loss}_E = \text{Cosine}\left(E\left(\vec{z}\right), \vec{p}\right) + \widehat{\text{KL}}\left(E\left(\vec{z}\right), \vec{p}\right) \tag{5}$$

and constrain the similarity between the outputs of the model $D$ and the latent variable $\vec{z}$ of the simulated ST data and the real ST dataset:

$$\text{Loss}_R = \text{Cosine}\left(R\left[E\left(\vec{z}\right)\right], \vec{z}\right) + \widehat{\text{KL}}\left(R\left[E\left(\vec{z}\right)\right], \vec{z}\right) \tag{6}$$

$$\text{Cosine}(a, a') = \frac{\sum a \times a'}{\sqrt{\sum a^2} \times \sqrt{\sum a'^2}} \tag{7}$$

$$\overline{\text{KL}}(a, a') = \sum a \times \log \frac{a}{a'} \underset{a}{\text{Min}} \, \text{Cosine}(a, a') + \text{KL}(a, a') \tag{8}$$

where $\vec{p}$ is the ground truth cell-type proportion of simulated ST data, the $R[E(\vec{z})]$ is the outputs of model $R$ for the real and simulated ST data, KL($a, a'$) is the Kullback-Leibler Divergence[71].

The weight of the model is initialized using the He initialization method[72]. The training of Spoint contains the following steps: 1) We calculated Loss$_E$ using $E(\vec{z})$ predicted from the simulated ST data and ground truth cell-type proportion $\vec{p}$, then adopted the Adam optimization algorithm[73] to obtain optimal parameters for model $E$. 2) We calculated Loss$_R$ using $R[E(\vec{z})]$ from the real and simulated ST data and latent variable $\vec{z}$, and Loss$_M$ using $E_2(\vec{z})$ and $E_2(\vec{z}')$ from the variable of model $E$'s last hidden layer for the simulated ST data and the real ST data, then updated the parameters of model $R$ and model $E$ using the Adam optimization algorithm. 3) We repeated steps 1) and 2) iteratively to update the parameters of models $E$ and $R$ until the Loss$_E$ of two adjacent training epochs reached a convergence value, i.e., $\Delta\text{Loss}_Q < 0.001$. After the training of Spoint, the values $E(\vec{z})$ at the final convergence epoch were used to predict the cell type composition of the real spatial data.

## Graph convolutional network of Splane

In Splane, we first used the spatial coordinates of spots/cells as inputs to construct an undirected graph. In the graph, each node represents a spot/cell, and an edge connects two spots/cells that are physically adjacent in space. The graph structure is stored in an $N \times N$ adjacency matrix $A$, in which $N$ is the number of spots/cells, and $A_{uv} = 1$ if spot/cell $v$ belongs to the k nearest neighbors of spot/cell $u$, otherwise $A_{uv} = 0$. For single-cell resolution ST data, such as MERFISH and STARmap, we set the default value of $k = 25$; but for 10X Visium data, we set the default value of $k = 6$, since each spot in the 10X Visium data is surrounded by 6 spots, and approximately 50–100 cells are contained in the 6 neighbor spots when $k = 6$.

Then, we used the cell type composition of spots/cells $Q$ and the adjacency matrix $A$ to construct a graph convolutional network (GCN) model $H$, as

$$H_{l+1}(Q) = f\left(H_l(Q), A\right) \tag{9}$$

where $l$ is the index of the layer, so $H_0$ is the input layer. We used the cell type composition of each spot/cell as the input layer for this model. According to Kipf and Welling's study[45], the propagation rule in GCN can be written as

$$f\left(H_l, A\right) = \text{ReLU}\left(\widehat{D}^{-\frac{1}{2}} \hat{A} \widehat{D}^{-\frac{1}{2}} H_l W_l\right) \tag{10}$$

$$\hat{A} = A + I \tag{11}$$

where $I$ is an identity matrix, $\hat{D}$ is the diagonal matrix of $A$, and $W_l$ is the weight matrix of the $l^{th}$ layer. We used a Chebyshev polynomial filter to estimate the convolution kernel of the GCN model[74]. In this process, a signal $x$ is filtered by $g_\theta$ as

$$y = g_\theta(L)x = g_\theta\left(U\Lambda U^T\right)x = Ug_\theta(\Lambda)U^Tx \tag{12}$$

where Laplacian matrix $L$ of the graph is given by $L = I - D^{-\frac{1}{2}}AD^{-\frac{1}{2}}$. The Laplacian matrix $L$ is real symmetric positive semidefinite matrix. It possesses a complete set of orthonormal eigenvectors $\{u_l\}_{l=0}^{n-1} \in R^n$, which are referred to as the graph Fourier modes. These eigenvectors are associated with ordered real nonnegative eigenvalues $\{\lambda_l\}_{l=0}^{n-1}$, and they correspond to the frequencies of the graph. The Fourier basis $U = [u_0, ..., u_{n-1}] \in R^{n \times n}$ diagonalizes the Laplacian matrix $L$, such that $L = U\Lambda U^T$, where $\Lambda = \text{diag}\left([\lambda_0, ..., \lambda_{n-1}]\right) \in R^{n \times n}$.

The Chebyshev polynomial filter can be expressed as

$$g_\theta(\Lambda) = \sum_{k=0}^{K-1} \theta_k T_k\left(\widetilde{\Lambda}\right) \tag{13}$$

where the parameter $\theta \in R^K$ is a vector of Chebyshev coefficients and $T_k(\widetilde{\Lambda}) \in R^{n \times n}$ is the Chebyshev polynomial of order $k$ evaluated at $\widetilde{\Lambda} = 2\Lambda/\lambda_{\max} - I$, a diagonal matrix of scaled eigenvalues that lie in $[-1, 1]$. Consequently, the filtering operation can be written as $y = g_\theta(L)x = \sum_{k=0}^{K-1} \theta_k T_k(\widetilde{L})x$, where $T_k(\widetilde{L}) \in R^{n \times n}$ is the Chebyshev polynomial of order k evaluated at the scaled Laplacian $\widetilde{L} = 2L/\lambda_{\max} - I$. Denoting $\bar{x}_k = T_k(\widetilde{L})x \in R^n$, we can use the recurrence relation to compute $\bar{x}_k = 2\widetilde{L}\bar{x}_{k-1} - \bar{x}_{k-2}$ iteratively with $\bar{x}_0 = x$ and $\bar{x}_1 = \widetilde{L}x$ as the initial values. The order of the polynomial $K$, controls the receptive field of the convolution kernel and is related to the parameter $K$ in the Chebyshev polynomial $T_k$. We set the default $K$ equal to 2 to ensure that the GCN model contains the information of the first order neighbors and second order neighbors (i.e., neighbors of neighbors) of each spot.

The GCN model of Splane consists of 5 layers, i.e. $H_0 \sim H_4$, where the first layer $H_0$ and the last layer $H_4$ both contain the cell type composition of each spot/cell, and the middle layer $H_2$ is used to cluster spots/cells and identify spatial domains. In Splane we embedded a cosine function Loss$_C$ to minimize the difference between $H_0$ and $H_4$, i.e.

$$\text{Loss}_C = \text{Cosine}\left[H_0(Q), H_4(Q)\right] \tag{14}$$

and we used another objective function Loss$_S$ to minimize the difference between the $H_2$ values of a spot and its neighbors, as

$$\text{Loss}_S = \left| H_2(q_i) - \frac{1}{n}\sum H_2\left(q_i^{\text{neigh}}\right) \right| \tag{15}$$

where $q_i$ and $q_i^{\text{neigh}}$ are the cell type composition of spot/cell $i$ and its neighbors respectively, and $n$ is the number of neighbors.

## Adversarial learning for multiple slices

In Splane, we used a discriminator model $D$ containing four layers to conclude uniform features from multiple slices. The input layer of model $D$ is the output latent variables of spot/cell $i$ from model $H$, i.e., $H_2(q_i)$; the output layer contains the probability $p_{i,s}$ of spot/cell $i$ belonging to slice $s$; and the two hidden layers are 64 nodes fully-connect layers. We defined the objective function of model $D$ as

$$\text{Loss}_D = \frac{1}{N_s}\sum_s y_{i,s} \log\left(p_{i,s}\right) \tag{16}$$

where $N_s$ is the number of slices and $y_{i,s}$ is the slice label of spot/cell $i$. The $Loss_D$ is minimized when the embeddings for different sections are maximally different, allowing the discriminator to correctly predict the labels of the slices. As shown in the next section, we trained GCN model with spots/cells from multiple ST slices, optimizing it to achieve a maximum $Loss_D$, ensuring that the latent variables of spots/cells from different slices would have the highest similarity.

## Clustering of spots/cells with latent features

The total objective function of Splane is a combination of the objective function of the GCN model and the objective function of the discriminator $D$, as

$$Loss = \alpha_C Loss_C + \alpha_S Loss_S - \alpha_D Loss_D \quad (17)$$

where $\alpha_C$, $\alpha_S$, and $\alpha_D$ are the weights of $Loss_C$, $Loss_S$, and $Loss_D$, and their default values are 1, 1, and 0.5, respectively. The weight of the model is initialized using the Xavier initialization method[75]. We used the RMSProp optimization algorithm to minimize the total objective function. Meanwhile, we clustered spots/cells using the output latent variables (termed shared latent features) of the GCN model (i.e. $H_2(Q)$) and the K-mean algorithm, and we evaluated the clarity of the clustering result using the Davies-Bouldin score (DBS) implemented by scikit-learn package[76,77]. We trained the entire model until both Loss and DBS within two adjacent training epochs reached convergent values, i.e., $\Delta Loss < 0.0001$ and $\Delta DBS < 0$. After the model converged, we adopted the spot/cell clusters obtained by Splane in the last epoch as spatial domains. As the shared latent features of the GCN model contain gene expression and spatial coordinate information of cells/spots, each spatial domain represents a batch of cells/spots that have common ground in both the transcriptome and space.

## Alignment of consecutive ST slices using Scube

In Scube, we first built a nearest neighbor graph between the cells/spots of two adjacent slices based on the coordinate of spots. As an initialization step for optimization algorithms, Scube aligns all slices by translating them so that their center points coincide with the coordinates (0,0). Then, we used the spatial domain information for each pair of nearest neighbor spots to construct an alignment objective function of two adjacent slices. We employed a differential evolution optimization to search for the optimal translation vectors and rotation angles, and transformed the coordinates of every slice accordingly. During each iteration of the global optimization process, Scube utilizes the adjusted coordinate information of spots to identify the nearest neighbors between two adjacent slices. By adjusting the coordinates of spots in the source slice, Scube aims to align the source slice to the target slice, ultimately maximizing the alignment objective function value. Finally, in Scube we performed all of the aforementioned steps for all adjacent slices, and used the transformed coordinates to stack these slices and construct the 3D alignment of the tissue.

Specifically, we used rigid body transformations (including mirroring, rotation, and translation) to transfer a slice $j$ as

$$S'_j = RMS_j + T \quad (18)$$

$$R = \begin{pmatrix} \cos\theta_j & -\sin\theta_j \\ \sin\theta_j & \cos\theta_j \end{pmatrix} \quad (19)$$

$$M = \begin{pmatrix} flip_j & 0 \\ 0 & 1 \end{pmatrix} \quad (20)$$

$$T = \begin{pmatrix} \Delta x_1 & \dots & \Delta x_{n_j} \\ \Delta y_1 & \dots & \Delta y_{n_j} \end{pmatrix} \quad (21)$$

where $S_j \overset{def}{=} \begin{pmatrix} x_1 & \dots & x_{n_j} \\ y_1 & \dots & y_{n_j} \end{pmatrix}$ is the coordinate matrix of slice $j$, $S'_j$ is the coordinate matrix after rigid body transformation, and $n_j$ is the number of spots in slice $j$; $R$ is the rotation matrix, $\theta_j$ is the rotation angle of slice $j$; $M$ is the mirroring matrix, $flip_j = -1$ represents the mirror operation of slice $j$, elsewise $flip_j = 1$; $T$ is the translation matrix. We defined $O_j$ as the set of overlapped spots/cells for the transformed slice $j$ between slice $j-1$ and the transformed slice $j$. This set includes spots/cells in the transformed slice $j$ whose Euclidean distance to their kth mutual nearest neighbor spots/cells in slice $j-1$ is less than a maximum distance. By default, the maximum distance defined as median Euclidean distance to the 2kth nearest neighbor of spots/cells in slice $j-1$. Then we defined an alignment objective function (AOF) to measure the alignment between the transformed slices $j$ and slice $j-1$ as follows:

$$AOF\left(S'_j | S_{j-1}\right) \overset{def}{=} \frac{1}{n_o} \sum_{l'_j \in O_j} \frac{1}{m} \sum_{l_{j-1} \in \langle l'_j \rangle_k^{j-1}} \delta\left[domain\left(l_{j-1}\right), domain\left(l'_j\right)\right] + f\left(\frac{n_o}{n_j}\right) \quad (22)$$

where $l'_j \in O_j$ are overlapped spots/cells in transformed slice $j$; $\langle l'_j \rangle_k^{j-1}$ is a set of spots/cells in slice $j-1$ which means the mutual $k$ nearest neighbors of spot/cell $l'_j$; $m$ means the elements number in set $\langle l'_j \rangle_k^{j-1}$; $\delta(x,y)$ is a Kronecker function. $f(x) = -(x-1)^p$ is the penalty term; $p$ is the exponent of the penalty, with a larger $p$ indicates stronger partial alignment capability; $n_o$ is the number of overlapped spots/cells in slice $j-1$; $n_j$ is the number of all spots/cells in the transformed slice $j$; Then we adopted the differential evolution algorithm to determine the best transformation matrices $\{R,M,T\}$ to maximize the AOF value between two adjacent slices. After determining transformation matrices for all slices, we can stack these transformed slices to construct a 3D alignment of the tissue.

## Gaussian progress regression model

We embedded a Gaussian process regression method[5,59] in SPACEL to obtain the continuous 3D distribution of a transcript from Scube transformed slices. First, we adopted the Alpha shape algorithm[78,79] to generate a 3D meshed manifold from the coordinates of the spots/cells in all slices, and smoothed this manifold using a subdivision algorithm[80]. We built a Gaussian process (GP) model for the expression level $y_i^{(j)}$ of gene $i$ in each spot/cell $j$, as

$$y_i^{(j)} | x^{(j)} = f_i(x^{(j)}) + \epsilon \quad (23)$$

where $x^{(j)}$ is the coordinate of spot/cell $j$, $\epsilon$ represents Gaussian additive noise, and the functional form of $f_i(\cdot)$ is unknown and may vary between genes, denoted as

$$f_i(x) \sim GP\left[\mu\left(\vec{x}\right), Cov\left(\vec{x}, \vec{x}'\right)\right] \quad (24)$$

$$Cov\left(\vec{x}, \vec{x}'\right) = \sum_{j,j'} \delta^2 \exp\left[-\frac{\left(x^{(j)} - x^{(j')}\right)^2}{2l^2}\right] \quad (25)$$

where $x^{(j)}$ is the coordinate of spot/cell $j$, $\mu(\vec{x})$ represents the mean function, $Cov(\vec{x}, \vec{x}')$ represents the covariance function, and $\delta$ and $l$ are the process-variance and length-scale hyperparameters, respectively. The objective function for training the GP model is the marginal

likelihood,

$$L(y_i|x,\theta) = \int P(y_i|f_i,x,\theta)P(f_i|x,\theta)df_i \qquad (26)$$

of gaussian process model. The hyper-parameter $\theta$ can be set by maximizing a posteriori estimation, defined as:

$$\theta^* \leftarrow \arg_\theta \max \int P(y_i|f_i,x,\theta)P(f_i|x,\theta)P(\theta)df_i \qquad (27)$$

We set the initial values of these hyperparameters as $\mu(\vec{\mathbf{x}}) = \text{mean}(y_i)$, $\delta = 4$, and $l = \lambda \cdot \text{var}(y_i)$, and we used a grid search strategy[81] to determine the value of $\lambda$ that maximizes the marginal likelihood. Then, we used the coordinates and gene expression information of spots/cells in all slices to train this GP model, and optimized the hyperparameters $\mu(\vec{\mathbf{x}})$, $\delta$, and $l$ using the Adam algorithm[73]. Finally, we uniformly sampled 500,000 points in the 3D meshed manifold, and used the trained GPR model to predict the expression level of a gene at each point. The gene expression information of these points constitutes the distribution of a transcript in 3D space. Based on the GPR model, we can use the Bayes Factor (BF) to quantify the spatial variation of a given gene in 3D space[5]:

$$BF = \frac{L(M_1)}{L(M_2)} \qquad (28)$$

where $L(M_1)$ denotes the marginal likelihood of the optimized GPR model and $L(M_2)$ denotes the marginal likelihood of a simplified GPR model in which $l$ was set to $\infty$. Larger BF values denote greater spatial variation in the gene expression level.

## Robustness test of SPACEL

To test the robustness of SPACEL, we evaluated its performance across various hyperparameters. Each hyperparameter was varied individually while keeping the other hyperparameter values constant. In Spoint, we assessed its performance by varying two user-defined hyperparameters (the number of simulated spots in the training set and the number of marker genes in each single-cell cluster) and three model hyperparameters (the dimensions of latent layers, hidden layers of predicted model, and probability output layers) in real single-cell resolution ST datasets[4,49]. In Splane, we compared its performance by varying five hyperparameters (the dropout rate of the model layers, the weight of the adversarial loss, the degree of neighbors in the constructed graph, the dimensions of latent layers, and the dimensions of hidden layers) using human DFPLC data[34]. In Scube, we tested its performance by varying the five hyperparameters used in Splane, one additional hyperparameter (the number of spatial domains), and two hyperparameters specific to Scube (the number of nearest neighbors in the MNN graph and the exponent of the penalty for the overlap ratio between adjacent slices) using 3D STARmap data[58] with a crop rate of 0.25.

## Jaccard index and shifting distance

To evaluate the prediction accuracy of the spatial domains identified by each method, we computed the Jaccard Index (JI) for each spatial domain with a manually annotated cortical layer. Specifically, we defined a set $P_d$ that contains all spots in the spatial domain $d$ and a set $P_l$ that contains all spots in the cortical layer $l$. We calculated $JI(d,l) = \frac{P_d \cap P_l}{P_d \cup P_l}$ between a spatial domain $d$ and all cortical layers, and defined the cortical layer with the largest JI value as the corresponding layer of domain $d$, i.e., layer $l_d$:

$$l_d = \text{argmax}_l[JI(d,l)] \qquad (29)$$

We then used $JI(d,l_d)$ to measure the prediction accuracy for the spatial domain $d$. We calculated the JI values for all spatial domains in a slice and used the average JI value for these domains to represent the prediction accuracy for the slice. In addition, for a spatial domain, we defined the distance from one of its spots to its corresponding cortical layer as the shifting distance (SD) of the spot and used the average SD of all spots in the spatial domain to evaluate the performance of each method. A lower SD value indicates a better consistency between the spatial domain and the corresponding cortical layer.

## Identification of spatial variable genes

We used two strategies to obtain the ground truth for spatial variable genes (SVGs). The first strategy was to use the differentially expressed genes of manually annotated cortical layers as the ground truth. Specifically, after normalizing the expression matrix of the ST data using Scanpy, we applied the two-sided Wilcoxon rank-sum test and the threshold log(fold-change) > 0.5, $P$-value < 0.01 to obtain differentially expressed genes (DEGs) for each cortical layer, which we used as the ground truth for SVGs. We used the same strategy to obtain DEGs for each spatial domain. Then, we calculated the sensitivity and specificity of the DEGs of spatial domains obtained by each method, and plotted receiver operating characteristic curves using the $P$-value ranking of DEGs.

## ASW of slices from different experimental batches

To assess the performance of Splane in reducing the batch effect when jointly analyzing multiple slices, we calculated the Average Silhouette Width (ASW) using the equation:

$$ASW = \frac{1}{K}\sum_{k=1}^{K}\frac{1}{C_k}\sum_{i=1}^{C_k} 1 - |\text{Silhouette}(i)| \qquad (30)$$

$$\text{Silhouette} = \sum_{i}^{N}\frac{b(i) - a(i)}{\max[a(i),b(i)]} \qquad (31)$$

$$a(i) = \frac{1}{n_s - 1}\sum_{j \in s}^{n_s} \text{distance}(i,j) \qquad (32)$$

$$b(i) = \min_{s'}\left[\frac{1}{n_{s'} - 1}\sum_{j \in s'}^{n_{s'}} \text{distance}(i,j)\right] \qquad (33)$$

where spot/cell $i$ is from dataset $s$, spot/cell $j$ is from dataset $s$ (for $a$) or $s'$ (for $b$), $C_k$ is the number of cells with the domain/cluster $k$, $K$ is the number of domains/clusters, $n_s$ is the spot/cell number in dataset $s$, and $N$ is the total number of spots/cells in all datasets. We calculated the Euclidean distance between spots/cells $i$ and $j$ from the raw expression matrix or from the latent features of spots/cells $i$ and $j$. We used the scib[82] implementation to compute the ASW scores.

## Copy number variation estimated by inferCNV

We employed inferCNV[57] to estimate copy number variations (CNVs) for the breast cancer ST slices. We used the default setting cut-off = 0.1 for inferCNV, i.e., only genes with an average count > 0.1 were considered. We used spots containing <10% cancer cells as reference samples and other spots as malignant samples. To compare the CNV values inferred from different slices and different experimental batches, we scaled the inferred CNV values to −1 - 1, and we calculated the average inferred CNV at each genomic locus for all spots in a spatial domain, to represent the mutation information of the spatial domain.

## Validation of immune domains

To validate that D3, D5, and D9 are immune domains, we calculated the percentage of overlap (PO) between the experimental annotation of immune spots and the spatial domains identified by Splane, as

$$\text{PO}_d = \frac{P_{\text{lym}} \cap P_d}{P_d} \tag{34}$$

where $P_{lym}$ is the set of immune spots annotated by experiments, and $P_d$ is the set of spots in domain d. For slices S1, S2, S5, and S6, we used the H&E staining information from the original study as the experimental annotation of immune spots[35]. For slice S10, we used the anti-CD3 immunofluorescence images of the slice to annotate immune spots[30]. Specifically, we first used DeepCell[68] and DAPI channel information to filter cells with sizes between 150 and 1000 pixels. Then, we scaled the CD3 fluorescence intensity of each pixel to 0−1, and used the average of the scaled CD3 intensity of pixels within a cell to represent the CD3 intensity of the cell. Cells with CD3 intensity > 0.37 were defined as CD3$^+$ cells, and spots containing > 30% CD3$^+$ cells were used as experimentally annotated lymphocyte spots.

## Metrics for benchmarking deconvolution methods

We employed the five metrics from a previous benchmarking study of deconvolution methods[50] to compare the performance of Splane with other state-of-the-art methods in processing the 32 simulated ST datasets:

1. Pearson correlation coefficient (PCC):

$$\text{PCC} = \frac{E\left[\left(\widetilde{\mathbf{x}}_i - \widetilde{u}_i\right)\left(\mathbf{x}_i - u_i\right)\right]}{\widetilde{s}_i s_i} \tag{35}$$

where $\mathbf{x}_i$ is the cell type composition of cell type $i$ in the ground truth, $u_i$ is the average cell type composition of cell type $i$ in the ground truth, $\sigma_i$ is the standard deviation of the cell type composition of cell type $i$ in the ground truth, and $\widetilde{x}_i$, $\widetilde{u}_i$, and $\widetilde{\sigma}_i$ are the corresponding values in the predicted result. For a cell type, a higher PCC value indicates higher deconvolution accuracy.

2. Structural similarity (SSIM):

$$\text{SSIM} = \frac{\left(2\widetilde{u}_i u_i + C_1^2\right)\left(2\text{cov}(\mathbf{x'}_i, \widetilde{\mathbf{x}}'_i) + C_2^2\right)}{\left(\widetilde{u}_i^2 + u_i^2 + C_1^2\right)\left(\widetilde{s}_i^2 + s_i^2 + C_2^2\right)} \tag{36}$$

where $\text{cov}(\mathbf{x}_i, \widetilde{\mathbf{x}}_i)$ is the covariance between the ground truth and the predicted result, and $C_1$ and $C_2$ are 0.01 and 0.03, respectively. For a cell type, a higher SSIM value indicates higher deconvolution accuracy.

3. Root mean square error (RMSE):

$$\text{RMSE} = \sqrt{\frac{1}{M}\sum_{j=1}^{M}\left(\widetilde{x}_{ij} - x_{ij}\right)^2} \tag{37}$$

For a cell type, a lower RMSE value indicates higher deconvolution accuracy.

4. Jensen-Shannon divergence (JS):

$$\text{JS} = \frac{1}{2}\text{KL}\left(\widetilde{\mathbf{P}}_i \middle| \frac{\widetilde{\mathbf{P}}_i + \mathbf{P}_i}{2}\right) + \frac{1}{2}\text{KL}\left(\mathbf{P}_i \middle| \frac{\widetilde{\mathbf{P}}_i + \mathbf{P}_i}{2}\right) \tag{38}$$

$$\text{KL}(\boldsymbol{a}_i || \boldsymbol{b}_i) = \sum_{j=0}^{M}\left(a_{ij} * \log\frac{a_{ij}}{b_{ij}}\right) \tag{39}$$

where $\mathbf{P}_i$ and $\widetilde{\mathbf{P}}_i$ are the spatial distribution of cell type $i$ in the ground truth and the predicted result, respectively. For a cell type, a lower JS value indicates higher deconvolution accuracy.

5. Accuracy score (AS): We computed the average PCC, SSIM, RMSE, and JS of all cell types predicted by each method, and sorted the average PCC/SSIM and RMSE/JS of all deconvolution methods in ascending and descending order, respectively, to obtain RANK$_{\text{PCC}}$, RANK$_{\text{SSIM}}$, RANK$_{\text{RMSE}}$, and RANK$_{\text{JS}}$. Then, we calculated the average of the four rankings to obtain the AS value of each method as follows:

$$\text{AS} = \frac{1}{4}\left(\text{RANK}_{\text{PCC}} + \text{RANK}_{\text{SSIM}} + \text{RANK}_{\text{RMSE}} + \text{RANK}_{\text{JS}}\right) \tag{40}$$

For a dataset, the method with the highest AS value has the best performance among all deconvolution methods.

For the DPLFC dataset, the cell type composition of each spot in the ST slices has not been validated by any experiments. To benchmark deconvolution methods in processing this real ST dataset, we adopted the cortical layers annotated by the original study as ground truth. We defined the accuracy score for the deconvolution results of the DPLFC datasets as the predicted proportion of one cell type in its corresponding cortical layer. Additionally, we used the Wilcoxon Rank Sum test to calculate P-values for the difference in the average proportion of each excitatory neuron cell type between the corresponding layer and the other layers across all samples. The method with the highest accuracy score and the lowest P-value has the highest deconvolution accuracy.

## Evaluation of alignment on continuous slices

We employed a 3D STARmap dataset of the mouse brain from Wang et al. study to build a ground truth to compare the performance of Scube, PASTE, and STAligner. Specifically, we divided the 1400 μm × 1700 μm × 100 μm mouse brain sample into ten 10 μm thick slices, $\{\hat{S}_i\}$, and used these slices as a ground truth. To simulate changes in the shape and position of slices during tissue sectioning, we randomly flipped, cropped, translated, and rotated each ground truth slice, to generate an ST dataset comprising consecutive 2D slices, $\{S_i\}$. Specifically, each slice (except the first slice) was randomly flipped, rotated by a random degree of 0°–360°, and translated by a random distance of 300–2000 μm along the x-axis and/or y-axis. Moreover, each slice was cropped from its edges with a crop ratio of 0.05–0.25. Then, we used Scube, PASTE, and STAligner to align each pair of adjacent simulated slices, $(S_i, S_{i+1})$, and obtain the aligned pair of slices, $(S'_i, S'_{i+1})$.

To compare the alignment results of Scube, PASTE, and STAligner with the ground truth, we first calculated the transformation matrix $M_i$ from an aligned slice $S'_i$ and its corresponding ground truth slice $\hat{S}_i$ as

$$M_i = S_i'^{(3)}\begin{pmatrix} \hat{S}_i^{(3)} \\ \mathbf{1}_{1\times 3} \end{pmatrix}^{-1} \tag{41}$$

where $S_i'^{(3)}$ and $\hat{S}_i^{(3)}$ are the first three columns of $S'_i$ and $\hat{S}_i$, respectively. We used the transformation matrix $M_i$ to transform the aligned slice $S'_{i+1}$ adjacent to $S'_i$, as

$$S_{i+1}^* = M_i\begin{pmatrix} S'_{i+1} \\ \mathbf{1}_{1\times n_{i+1}} \end{pmatrix} \tag{42}$$

Then, we divided the transformed aligned slice $S_{i+1}^*$ and its corresponding ground truth slice $\hat{S}_{i+1}$ into 10 × 10 grids, and calculated the SSIM/PCC values between the cell type compositions of the grids in $S_{i+1}^*$ and $\hat{S}_{i+1}$, to measure the accuracy of Scube, PASTE, and STAligner.

In addition, we employed mouse brain MERFISH, mouse embryo Stereo-seq, and mouse brain Spatial Transcriptomics[1] data to assess the performance of Scube, PASTE, and STAligner. As there are no ground truths for these datasets, we divided the common area of each pair of adjacent slices into 10 × 10 grids, and calculated the SSIM/PCC values between the cell type composition of the grids in the two adjacent slices aligned by Scube, PASTE and/or STAligner to measure the performance of the two methods.

## Hyperparameter settings for benchmarking methods

The selection of the hyperparameters for each method was followed by the official tutorials and codes provided by their respective authors:

- STAligner: https://staligner.readthedocs.io/en/latest/Tutorial_DLPFC.html
- PRECAST: https://feiyoung.github.io/PRECAST/articles/PRECAST.BreastCancer.html
- STACI:
  STACI GitHub repository for Visium and Spatial Transcriptomes data, and cells with gene number less 10 were filtered:
  https://github.com/uhlerlab/STACI/blob/master/train_gae_visium_10xADFFPE.ipynb
  for MERFISH data:
  https://github.com/uhlerlab/STACI/blob/master/train_gae_starmap_multisamples.ipynb
- STAGATE: https://stagate.readthedocs.io/en/latest/T1_DLPFC.html
- SpaGCN: https://github.com/jianhuupenn/SpaGCN/blob/master/tutorial/tutorial.ipynb
- BayesSpace: https://edward130603.github.io/BayesSpace/articles/maynard_DLPFC.html
- stLearn: https://stlearn.readthedocs.io/en/latest/tutorials/stSME_clustering.html#Human-Brain-dorsolateral-prefrontal-cortex-(DLPFC)
- CARD: https://yingma0107.github.io/CARD/documentation/04_CARD_Example.html
- Cell2location: https://cell2location.readthedocs.io/en/latest/notebooks/cell2location_tutorial.html
- DestVI: https://docs.scvi-tools.org/en/stable/tutorials/notebooks/spatial/DestVI_tutorial.html
- DSTG: https://github.com/Su-informatics-lab/DSTG
- RCTD: https://raw.githack.com/dmcable/spacexr/master/vignettes/spatial-transcriptomics.html
- Seurat: https://satijalab.org/seurat/archive/v3.2/integration.html
- SpatialDWLS: https://drieslab.github.io/Giotto_site_suite/articles/analyses_deconvolution_Oct2021.html
- SPOTlight: https://marcelosua.github.io/SPOTlight
- Stereoscope: https://docs.scvi-tools.org/en/stable/user_guide/models/stereoscope.html
- STRIDE: https://stridespatial.readthedocs.io/en/latest/tutorials/Mouse_embryo.html
- Tangram: https://github.com/broadinstitute/Tangram

The hyperparameters used for each method were reported in Supplementary Data 2.

## Statistics & reproducibility

No statistical method was used to predetermine sample size. In Fig. 2a, b, the CARD was excluded from the benchmark on the scRNA-seq data simulated datasets due to lack of spatial information for these datasets. In Supplementary Fig. 4e, the STRIDE was not shown in Stereo-seq mouse embryo brain dataset due to a runtime error for this dataset. In Supplementary Fig. 14a, two slices were excluded due to the lack of L6b cells, which prevented the definition of ground truth. The experiments were not randomized. The Investigators were not blinded to allocation during experiments and outcome assessment.

## Reporting summary

Further information on research design is available in the Nature Portfolio Reporting Summary linked to this article.

## Data availability

All relevant data supporting the key findings of this study are available within the article and its Supplementary Information files. The simulation datasets used for the evaluation of Spoint and other deconvolution methods are available in a GitHub repository at https://github.com/QuKunLab/SpatialBenchmarking. The raw data of 11 ST datasets and five paired single-cell/nucleus RNA sequence datasets are available from the following studies: (1) 12 slices of human DLPFC 10X Visium data at http://research.libd.org/spatialLIBD/[34]; (2) six slices of human breast cancer 10X Visium data at https://doi.org/10.5281/zenodo.4739739[35]; (3) four slices of human breast cancer 10X Visium data: Parent_Visium_Human_BreastCancer, V1_Breast_Cancer_Block_A_Section_1, V1_Breast_Cancer_Block_A_Section_2 and Visium_FFPE_Human_Breast_Cancer at https://support.10xgenomics.com/spatial-gene-expression/datasets[14]; (4) one slice of human breast cancer 10X Visium data: Invasive Ductal Carcinoma Stained With Fluorescent CD3 Antibody at https://support.10xgenomics.com/spatial-gene-expression/datasets[30]; (5) Mouse brain STARmap data at https://www.starmapresources.com/data[58]; (6) 33 slices of Mouse MOp MERFISH data at https://doi.braininagelibrary.org/doi/10.35077/g.21[13]; (7) one slice of mouse E16.5 embryo brain Stereo-seq data, one slice of mouse brain Stereo-seq data, and 13 slices of mouse E16.5 whole embryo Stereo-seq data at https://db.cngb.org/stomics/mosta/download/[4]; (8) ten slice of human brain MERFISH data at https://datadryad.org/stash/dataset/doi:10.5061/dryad.x3ffbg7mw[49]; (9) 75 slice of mouse whole brain Spatial Transcriptomics data are available in the GEO database under accession number GSE147747[60]; (10) single-nucleus transcriptomics data across multiple human cortical areas at https://portal.brain-map.org/atlases-and-data/rnaseq/human-multiple-cortical-areas-smart-seq[51]; (11) single-cell transcriptomics data of human breast cancer data at https://singlecell.broadinstitute.org/single_cell/study/SCP1039[35]; (12) single-cell transcriptomics data of mouse embryo brain at http://mousebrain.org/development/downloads.html[83]; (13) single-cell transcriptomics data of mouse whole cortex and hippocampus at https://portal.brain-map.org/atlases-and-data/rnaseq/mouse-whole-cortex-and-hippocampus-10x[52]; (14) single-cell transcriptomics data of mouse whole brain at mousebrain.org/adolescent/downloads.html[84]. All the aforementioned data used in this study are available in a public Zenodo repository at https://doi.org/10.5281/zenodo.8316334. Source data for figures are provided with this paper. Source data are provided with this paper.

## Code availability

The open-source package of SPACEL is available at a GitHub repository: https://github.com/QuKunLab/SPACEL[85]. We uploaded all codes and scripts used for the analyses and figure plotting in this study to a public Zenodo repository (https://doi.org/10.5281/zenodo.8316334). The documentation, usage instructions and other relevant information related to SPACEL can be accessed through (https://spacel.readthedocs.org).

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

## Acknowledgements

This work was supported by the National Key R&D Program of China (2020YFA0112200 and 2022YFA1303200 to K.Q.), the National Natural Science Foundation of China grants (T2125012 and 91940306 to K.Q.; 32170668 to B.L.), CAS Project for Young Scientists in Basic Research YSBR-005 (to K.Q.), Anhui Province Science and Technology Key Program (202003a07020021 to K.Q.) and the Fundamental Research Funds for the Central Universities (YD2070002019, WK9110000141, and WK2070000158 to K.Q.). We thank the USTC Supercomputing Center and the School of Life Science Bioinformatics Center for providing computing resources for this project.

## Author contributions

K.Q., J.L., and B.L. conceived the project. H.X., J.L., and S.Wang designed the framework and performed data analysis with the help of M.F., S.L., C.C., S.Wan, R.W., M.T., and T.X.; B.L., K.Q., H.X., and J.L. wrote the manuscript with input from all authors. K.Q. and J.L. supervised the entire project. All authors read and approved the final manuscript.

## Competing interests

The authors declare no competing interests.
