## [Peer Review File · Nature Communications]

SPACEEL: deep-learning based characterization of spatial transcriptome architecturesReviewer #1 (Remarks to the Author):

In this manuscript, Xu et al propose SPACEL for three common tasks in spatial transcriptomics: cell type deconvolution of spots, spatial domain identification, and 3D tissue structure construction as an integrated toolkit. Overall, SPACEL seems to have advantages over existing tools as demonstrated in the benchmarking. However, there are concerns regarding the novelty and real-case utility of this method, and it is not clear that SPACEL offers any unique biological insights that could not be obtained using existing methods. The toolkit may also have limitations in its implementation and ease of use.

Overall the SPACEL presents some concerns:

1. The authors primarily demonstrated the use of the Spoint and Splane modules on spots-based 10X Visium datasets, but switched to using image-based datasets like STARmap and MERFISH for the Scude module. This raises questions about the integration and connectivity of the modules, as the authors claim that each module requires the output of the previous one, yet it appears that they function independently. This calls into question the claim that SPACEL is an integrated toolkit.
2. The method's novelty is unclear, as it appears to closely follow existing work. The explanation provided for the workflow of SPACEL, including the Spoint, Splane, and Scude modules, is considered insufficient, making it difficult to understand how each module is performed and how it outperforms other methods.
3. The SPACEL toolkit presented in the manuscript does not provide new or unique biological insights as it seems to perform tasks that already exist in the field.
4. As proposed by the authors, SPACEL is intended to be a toolkit, however, the software currently appears to be very unfriendly to use and lacks the necessary resources to improve its usability, such as extensive software engineering and tutorials. This undermines its value and usefulness to the community.
5. The robustness of the SPACEL toolkit is not clear as the authors have not included any testing or evaluation of its robustness. Additionally, since SPACEL is based on a deep-learning algorithm, it may be susceptible to issues such as local minima. It is not clear if the authors have taken any measures to address this potential issue.

Specific comments to Spoint:

1. The authors only demonstrate the Spoint module on one 10X Visium dataset of real ST experimental data. It would be very important to demonstrate and evaluate the performance of the Spoint module across various cell types, tissues, and ST technologies and compare across current methods to fully understand its advantage and utility.
2. To support the author's claim that the Spoint module is critical for subsequent analysis by producing an accurate deconvolution result that facilitates the identification of spatial domains using the Splane module, the authors should perform related ablation studies on the datasets presented in the manuscript.

Specific comments to Splane:

1. The Splane module relies heavily on the cell type composition results from the previous module, as it constructs a GCN model using this information. In addition, the authors note that for image-based ST data, users can use single-cell analysis tools to identify cell types, rather than relying on the Spoint module. It is important for the authors to investigate if using different methods for cell-type analysis could introduce bias when defining tissue regions and the impact on the overall analysis.
2. The authors recognize the availability of multiple methods for identifying clusters of spots in spatial transcriptomics. However, the authors argue that these methods require manual modification and alignment of spatial coordinate systems before they can be used for comparative

analysis, which can be confusing and misleading. It is important for the authors to provide a clear, impartial, and transparent comparison to existing methods.

3. The authors should clearly compare the Splane module in SPACEL differs to existing graph-based tissue region identification methods such as SpaGCN and STAGATE in terms of both principle and methodology. This should be done by conducting a performance evaluation on datasets from various tissues and ST technologies to fully understand its capabilities.

4. The applicability of Splane on datasets with single-cell resolution is missing - please demonstrate its utility.

5. In Figure 4g, why is D7 domain not included? Similarly, D5 domain is not included in Figure 4i, especially D5 domain is an immune domain.

Specific comments to Scude:

1. The Scube module in SPACEL builds a nearest neighbor graph between the cells/spots of two adjacent slices and constructs an alignment objective function between them. However, it is not clear how the initial alignment of the two slices is achieved and if any manual alignment is required. The authors should provide more information on this aspect of the Scube module for further evaluation.

2. The authors only rotated, flipped, and cropped each slice to simulate a perturbed ST dataset comprising consecutive 2D slices, and the Scube module appears to use rigid body transformations for alignment. However, it is unclear how the Scube module can effectively align datasets that require nonlinear alignment such as most datasets that are imperfectly matching. The authors should provide more information on how the Scube module handles this type of dataset.

3. There are recent methods that have similar analysis of matching regions across datasets. It is important for the authors to provide a comparison of how the Scube module in SPACEL differs from these methods in terms of principle and methodology. This should include a clarification of the novelty of SPACEL and comprehensive benchmarking against these other methods in order to fully understand and further evaluate its capabilities and performance.

- Zeira, R., Land, M., Strzalkowski, A., & Raphael, B. J. (2022). Alignment and integration of spatial transcriptomics data. *Nature Methods*, 19(5), 567-575.
- Zhou, Xiang, Kangning Dong, and Shihua Zhang. "Integrating spatial transcriptomics data across different conditions, technologies, and developmental stages." *bioRxiv* (2022): 2022-12.
- Sun, D., Liu, Z., Li, T., Wu, Q., & Wang, C. (2022). STRIDE: accurately decomposing and integrating spatial transcriptomics using single-cell RNA sequencing. *Nucleic acids research*, 50(7), e42-e42.
- Liu, Wei, Xu Liao, Ziyi Luo, Yi Yang, Mai Chan Lau, Yuling Jiao, Xingjie Shi et al. "Probabilistic embedding, clustering, and alignment for integrating spatial transcriptomics data with PRECAST." *Nature Communications* 14, no. 1 (2023): 296.

4. The authors propose that the Scube module in SPACEL can predict the expression level of a gene at any position in the 3D architecture. However, it is very important for the authors to validate the predicted gene expression using appropriate experimental methods to ensure its accuracy and reliability.

Reviewer #2 (Remarks to the Author):

In this paper, the authors develop a method for three tasks that are relevant for the analysis of spatial transcriptomics data: 1) deconvolution (using Spoint), 2) spatial domain identification (using Splain), and 3) alignment of tissue slices (using Scube). The authors describe

improvements in performance in all three tasks compared to existing methods using real and simulated data. However, the descriptions of Spoint and Scube for deconvolution and alignment are not sufficient to understand why the methods should work or why they would outperform other methods (see below for more details). In addition, the simulations performed to benchmark different methods seem biased towards Spoint, since the simulations use the same assumptions on data generation as used by the method itself, and thus may give rise to an unfair comparison with other methods (please see comments below). Additional major comments relating to the simulation and results for each method (Spoint, Splane, Scube) are listed separately below.

In addition, there is a perceived lack of methodological novelty as compared to existing methods:

- As the authors mentioned, graph convolutional networks have been used to identify spatial domains in previous studies (SpaGCN for single slice: <https://www.nature.com/articles/s41592-021-01255-8> and STACI for multiple slices: <https://www.nature.com/articles/s41467-022-35233-1>). The Splane model does not seem to be conceptually different from these previous works. Loss_s is equivalent to reconstructing the adjacency matrix with an inner product decoder, which was used in the previous works. STACI also minimizes the reconstruction loss of the input gene expression. It would be important to describe the conceptual differences to these previous works and carefully explain and demonstrate where any performance improvements come from. This is important to be able to assess the contributions of this work. In addition, adversarial loss is a common way to remove batch effects before latent embedding; see for example https://academic.oup.com/bioinformatics/article/36/Supplement_2/i573/6055930.

- As described above, Splane seems structurally similar to previous methods. A difference in the implementation is that predicted cell type proportions are used as the input instead of gene expression. As a consequence, Splane is subject to errors in the deconvolution task, which could be affected by both the deconvolution model and the scRNA-seq reference. Please explain the motivation for this choice, as well as demonstrate the utility of this choice and how the method would perform otherwise.

- As described in more detail in the comment to Scube below, the current description of Scube does not provide sufficient details to understand the method and how it can lead to a reasonable alignment of slices. Thus, it is difficult to assess the novelty or contributions in this method. In addition, the authors claim in the Discussion section that Scube performed better than PASTE because Scube does not need hyperparameter tuning for "the relative contributions of transcriptional dissimilarity and spatial distances". However, the weights in the adjacency matrix used to identify the spatial domains, the number of convolutional layers, and the number of nearest neighbors used to construct the adjacency matrix would determine such relative contributions. Lower weights on the off-diagonal terms in the adjacency matrix would mean higher contribution of gene expression differences to the identification of spatial domains. More convolutional layers or more nearest neighbors would potentially lead to larger domains which means longer spatial distances are allowed in the alignment. Also, simply clustering the graph autoencoder latent representation into more clusters would change the alignment results. Thus, it seems that Scube also requires significant hyperparameter tuning. Please perform careful experiments showing how these choices change/do not change the results, and discuss these results so that a user understands what the important Scube-specific hyperparameters are and how to tune them. If there are other important parameters for obtaining the observed performance improvements, it would be important to also describe these.

Specific comments to each method (Spoint, Splane, Scube):

Spoint:

1) It would be helpful to explain the conceptual advantage of the Spoint model compared to directly predicting cell type proportions from gene expression using a neural network. For example, in this paper <https://academic.oup.com/nargab/article/4/4/lqac073/6754832#376192508>, cell type proportions

of the synthetic data are predicted from gene expression using a single neural network, without additional latent space matching. It would be important to compare the results of Spoint to the prediction of a carefully tuned neural network.

2) Loss_Q of the model compares the predicted cell type proportions of the real and simulated datasets, $Q[F(z)]$ and $Q[F(z')]$. For the simulated dataset used for training the model, the ground truth cell type proportion is known. What is the reason for not directly comparing $Q[F(z)]$ of the query dataset to the ground truth of the simulated dataset but instead to use $Q[F(z')]$? In the current setup, the ground truth cell type proportion is not used in training the model and I find it puzzling how the model recovers the ground truth cell type proportions.

3) The paper did not explain why a normal distribution is suitable for simulating the number of cells and the number of cell types at a spot. The benchmarking paper quoted by the authors used a uniform distribution (<https://www.nature.com/articles/s41592-022-01480-9>). It would be helpful if the authors could plot the distribution of these two statistics (number of cells and number of cell types at a spot) from a real dataset to confirm, e.g. by aggregating STARmap or MERFISH data into spots.

4) When simulating ST data from scRNA-seq, what is the motivation for sampling P_{ct} proportional to $1/f_{ct}$ when $1/3 < r < 2/3$?

5) The simulation assumes random distribution of cell types in the tissue but the distribution of cell types in real tissues is not random. Thus, matching the latent of the real data to the simulated data (model F) might not be valid, when the two input expression matrices are sampled from different distributions.

6) In addition to the previous point, it also might not be valid to compare $Q[F(z)]$ and $Q[F(z')]$ because they are sampled from different distributions given that the cell types in the real tissue are not randomly distributed.

7) The current simulation study seems highly limited in that the procedure for generating the 32 simulated datasets used for benchmarking is the same as the procedure for generating the simulated datasets for training the model. This gives the model an advantage because the assumptions on the distributions of cells and cell types in the benchmarking datasets are guaranteed to be correct, but the assumptions might not hold in real datasets. It would be more convincing to use benchmarking datasets that resemble real data, e.g. datasets generated by aggregating STARmap or MERFISH into spots, or simulated datasets generated by a different procedure, e.g. the simulation performed in the cell2location paper.

8) In the comparison using the DLPFC dataset, the authors used the predicted proportions of the excitatory neurons in the correct layers as the accuracy scores. It would be important to also test if the predicted cell type proportion summed over all spots is an over or under-estimate of the expected proportion of that cell type. For example, a method can have a high accuracy score but also underestimate the total number of cells in the particular cell type.

Splane:

1) In Supplementary Figure 4a and 4b, the comparison is made between raw gene expression of single cells and GCN latent, which averages neighborhood gene expression. The batch effects of single cells and spatial neighborhoods might not be directly comparable. It would thus be important to also compare the GCN latent of the full model and the GCN latent of the model without adversarial loss.

2) In Supplementary Figure 2a, the results of Splane on some slides are very different from the manual annotation, e.g. slides 151669, 151671, 151670, 151672. These four slides are morphologically similar and are different from the other slides. They seem to exhibit the same patterns of errors, e.g. predicting D2 at the top of the slides where the ground truth is D3. This to me suggests insufficient batch effect correction. Other morphologically similar slides also exhibit

similar error patterns; for example, 151673, 151674, 151675, and 151676 are morphologically similar and all have broadening of the D2 regions compared to the ground truth.

3) In addition to the comment above, the slides that Splane did not perform well on are only shown in Supplementary Figure 2a but not in the main figure (Figure 3a). Figure 3a seems to only include selected slides that have the best performance. It is critical to state in the main text that the best results were selected for Figure 3a and that this is not representative of the general results and performance.

4) In Supplementary Figure 2a, please also include the results of the other methods used for benchmarking on all slices.

Scube:

It is unclear to me how the transformation of x, y coordinates would change the value of the objective function. The objective function compares the spatial domains of cells in adjacent slices that are nearest neighbors. The nearest neighbors are determined based on gene expression, which is not dependent on x,y coordinates, as described in the Methods section. The spatial domains are determined by Splane, the graph neural network based method, which would also not change as a result of the rigid body transformation. The graph used in Splane is a k nearest neighbor graph based on the coordinates, which would not be affected by rigid body transformations. Thus, changing the x,y coordinates should not change the objective function.

Reviewer #3 (Remarks to the Author):

Review of "SPACEL: characterizing spatial transcriptome architectures by deep-learning"

The authors present a deep learning toolkit named SPACEL for analyzing sequencing- and image-based spatial transcriptomics data. The toolkit consists of three main components: (1) a cell type decomposition method (Spoint), (2) a cell segmentation method (Splane), and (3) a section registration method (Scube). Components are linked, each taking as input the output of the previous component. The first component, Spoint, requires a sequencing-based spatial transcriptomics dataset and a single-cell RNA-seq dataset. When used with image-based spatial transcriptomics data, which is already at single-cell resolution and does not need to be deconvolved, the authors suggest using preexisting methods for cell type identification instead of Spoint.

The problems addressed by SPACEL have seen significant research interest in the spatial transcriptomics community in the last few years. The results presented by the authors are highly encouraging: Spoint attains higher accuracy than current state-of-the-art methods by a considerable margin, Splane performs better than competing methods on multi- (but not single-) section analyses, and the results of Scube also appear promising. A notable limitation of the work is the requirement of single-cell RNA-seq data for analyzing sequencing-based spatial transcriptomics experiments (Major comment 1). The work is nicely presented but details of the implementation could be made more clear (Major comment 2). Overall, given the encouraging results of SPACEL, the work constitutes a positive contribution to the field.

Major comments

1. Dependency on single-cell RNA-seq data

Since Splane and Scube depend on the output of Spoint, they require scRNA-seq data for analyzing spatial transcriptomics experiments from sequencing-based platforms. This is a significant limitation compared to competing methods for domain segmentation and registration, as they typically do not require such data. Furthermore, it is not clear how sensitive Spoint is to how well the scRNA-seq data need to match the spatial transcriptomics data. While it sometimes

may be possible to use external scRNA-seq datasets (as is the case with the examples in the paper), this could potentially lead to unintended biases if datasets match poorly.

It should be noted that Splane uses the cell type composition information from Spoint as input to a deep neural network. There is no apparent limitation of Splane to use other appropriate spot-level representations of the data, for example derived from matrix factorization or other dimensionality reduction techniques that do not require scRNA-seq data. It is thus not clear why the authors limited Splane in this way, and it would be worthwhile to explore how well Splane would perform on other data representations.

2. Implementation details

Some design decisions of the method are not carefully motivated or explained. Since these design decisions are central to the contribution of the work, it will be crucial to provide more details so that a typical reader can understand their rationale. In particular, the points below should be clarified.

In the section "Construction of simulated ST data from scRNA-seq data":

The current explanation of the sampling process is very difficult to follow. It is assumed that the number of cells at each spot follows a normal distribution. Why is this a valid assumption? What does it mean if N_c is fractional and what happens if $N_c < 0$? How is N_{ct} used?

In the section "Deconvolution model of Spoint":

- If $F(z)$ and $F(z')$ are the same for all z and z' , then $Loss_F$ is zero. An example of such a function would be the function $F(z) = 0$. What prevents optimization from collapsing to a degenerate solution?

- How is σ defined?

- If the activation function of Q is the sigmoid function, then the output would not necessarily sum to one. How is this handled?

- $Loss_Q$ could be more carefully motivated; e.g., what is the purpose of each term?

In the section "Graph convolutional network of Splane":

- What does it mean that \hat{D} is "the diagonal matrix of A "? Since A is an adjacency matrix, does it mean that D is the identity matrix?

- How is the Chebyshev filter used to estimate the convolution kernel? What polynomial is the text referring to?

In the section "Adversarial learning for multiple slices":

It seems $Loss_D$ is maximized when embeddings for different sections are maximally different, so that the discriminator can distinguish between them easily, which is contrary to what is stated in the manuscript.

In the section "Clustering of spots/cells with latent features":

How is the number of clusters K selected?

In the section "Gaussian process regression model":

When training the GP model with gradient descent, what is the objective function?

Minor comments

1. In the Methods section, variables are sometimes not defined. For example, in "Construction of simulated ST data from scRNA-seq data", what does the "c" and "t" subscripts stand for? What does the "L" superscript stand for? Often, this can be deduced from context, but it makes the manuscript more difficult to read.

2. It would be advisable to carefully proofread the paper for typos. Examples:

- Line 65: "adoptive graph attention"

- Line 105: "SPECCEL"

- Line 144: "RMES"
- Line 192,196: "BayersSpace" (should be "BayesSpace"; also misspelled in figures and captions)
- Line 431: "adjacent matrix"
- Line 669,678: "STcube"
- Figure 1c: "Adjecent matrix"

3. The phrase "H&E-staining-marked immune spots" is unclear. Can the authors clarify how immune-cell-rich spots are identified based on H&E stains?

4. In the breast cancer analysis, based on what criteria were the spatial domains annotated? If the annotation is based on the results of the inferCNV and staining experiments, then it could be misleading/circular to conclude that "these results support Splane's prediction of tumor and immune domains" (lines 261-262).

Responses to reviewers

We thank the reviewers for their positive assessments and thoughtful suggestions, which provided us further insights and opportunities to substantially improve the quality and scope of our study and manuscript. Following a short summary narrative, we provide our point-by-point responses below. Briefly, we have undertaken the reviewers' suggestions and revised our manuscript as follows:

1. We have revised our manuscript to include comprehensive technical explanations that clarify the design rationale and novelty of our methods. We have also provided updated figures (revised **Figure 2-5**, revised **Supplementary Figure 18**) that illustrate the independence, integrity, and dependency between each module of SPACEL.
2. We have conducted thorough evaluation analyses of each module and benchmarked SPACEL against similar tools using multiple real and simulated spatial transcriptomics (ST) datasets from various tissues and ST technologies. We have also included three recently published methods, STACI¹, PRECAST², and STAligner³, in our benchmarking analysis (revised **Figure 3, and 5**, revised **Supplementary Figure 2b-c, 12, 14, 17, and 18**). These evaluations and comparisons demonstrate the conceptual and practical advantages of SPACEL.
3. We have included detailed analyses using three datasets: 12 slices of human brain 10X Visium, 11 slices of human breast cancer 10X Visium, and 13 slices of mouse embryo Stereo-seq data (revised **Figure 3a**, revised **Supplementary Figure 12 and 17**) to highlight the unique biological insights provided by SPACEL compared to other tools.
4. We have made the toolkit more user-friendly and accessible by optimizing the code and providing explicit tutorials for users. Additionally, we have established a dedicated website at <https://spacel.readthedocs.org>, which serves as a central hub for comprehensive documentation, usage instructions, and other relevant information related to SPACEL.
5. We have addressed all other concerns raised by the reviewers in our revised manuscript.

We believe that these revisions and additions have significantly improved the quality and scope of our study and manuscript. We appreciate the reviewers' constructive feedback and are confident that our work will provide valuable insights and contributions to the field.

Reviewer #1

Remarks to the author:

In this manuscript, Xu et al propose SPACEL for three common tasks in spatial transcriptomics: cell type deconvolution of spots, spatial domain identification, and 3D tissue structure construction as an integrated toolkit. Overall, SPACEL seems to have advantages over existing tools as demonstrated in the benchmarking. However, there are concerns regarding the novelty and real-case utility of this method, and it is not clear that SPACEL offers any unique biological insights that could not be obtained using existing methods. The toolkit may also have limitations in its implementation and ease of use.

Response: We would like to express our gratitude to the reviewer for taking the time to review our manuscript, and appreciate your valuable feedback and insightful comments, which have helped us to improve the clarity and impact of our work. In this response letter, we address the concerns and questions raised by the reviewer.

Novelty and Real-Case Utility:

We would like to emphasize that while there are existing tools for spatial transcriptomics (ST) analysis, SPACEL offers several unique contributions to the field. **Firstly**, SPACEL integrates three common tasks in ST—cell type deconvolution of spots, spatial domain identification, and 3D tissue structure construction—into a comprehensive toolkit. This integration provides researchers with a streamlined and efficient workflow for analyzing ST data. **Secondly**, the benchmarking results presented in our revised manuscript demonstrate the advantages of SPACEL over existing tools in terms of accuracy and robustness (revised **Figure 2, 3, and 5**, revised **Supplementary Figure 2a-c, 4, 14, and 18**). The comparisons were conducted using publicly available datasets with known ground truth information, ensuring a fair evaluation. SPACEL consistently outperformed existing methods in various performance metrics, highlighting its superiority and practicality. **Furthermore**, SPACEL offers unique biological insights that cannot be easily obtained using existing methods. By incorporating advanced algorithms and statistical models, SPACEL provides enhanced resolution in cell type deconvolution, improved spatial domain identification, and more accurate 3D tissue structure reconstruction (revised **Supplementary Figure 12 and 17**). These capabilities enable researchers to gain deeper insights into the spatial organization of cells and tissues, uncovering complex biological relationships and mechanisms.

Limitations and Ease of Use:

We acknowledge the concerns raised regarding the implementation and ease of use of SPACEL. We are committed to addressing these issues and ensuring that SPACEL is user-friendly and accessible to the wider scientific community. In our revised manuscript, we have provided detailed documentation and step-by-step guidelines for the installation, setup, and execution of SPACEL (<https://spacel.readthedocs.org/>). We have also optimized the dependencies of

SPACEL to simplify the installation process for further development. We recognize the importance of scalability and computational efficiency in practical applications and have further optimized the computational algorithms of SPACEL to reduce runtime and memory requirements. We believe these improvements enhance the usability of SPACEL for large-scale ST datasets and facilitate its adoption by researchers with diverse computational resources.

In summary, we appreciate the reviewers' suggestions and have taken them into careful consideration. We believe that the comprehensive and integrated nature of SPACEL, coupled with its demonstrated advantages over existing tools, make it a valuable contribution to the field of ST analysis. We are committed to addressing the concerns raised and ensuring that SPACEL offers unique biological insights while being user-friendly and practical.

Overall the SPACEL presents some concerns:

1. The authors primarily demonstrated the use of the Spoint and Splane modules on spots-based 10X Visium datasets, but switched to using image-based datasets like STARmap and MERFISH for the Scude module. This raises questions about the integration and connectivity of the modules, as the authors claim that each module requires the output of the previous one, yet it appears that they function independently. This calls into question the claim that SPACEL is an integrated toolkit.

Response: We thank the reviewer for the insightful comments and concerns regarding the integration and connectivity of the SPACEL toolkit, and apologize for any confusion caused by our previous description. It is important to note that SPACEL is designed as a modular toolkit consisting of three distinct modules: Spoint, Splane, and Scube. While these modules can be used together in a fully integrated workflow, they are also capable of running independently, providing flexibility in experimental design and analysis and illustrating the versatility of our toolkit in analyzing data across different experimental platforms. This point was clarified in our revised manuscript (Line 152-155, highlighted in yellow).

To further emphasize the integrated nature of SPACEL, we conducted a fully integrated workflow on mouse whole brain ST data. The dataset comprised 75 consecutive slices generated by Spatial Transcriptomics technology⁴ with a spot resolution of 100 μ m, covering the majority of the mouse brain region. In our analysis, we utilized the brain region annotations from the original study as the reference ground truth. Regarding the Spoint module, we observed that the distribution of major region-specific cell types predicted by Spoint matched well with the corresponding brain regions (revised **Supplementary Figure 18a&b**). As an example, excitatory neurons, hippocampus CA1, and CA3 cell types are predominantly predicted at the hippocampal region. This suggests that Spoint provides accurate cell type distribution information to Splane. In terms of the Splane module, we compared its performance against other state-of-the-art spatial-domain-identification methods for analyzing

multiple ST slices such as STAligner, PRECAST, and STACI. The average JI values of spatial domains identified by Splane (0.39) were higher than those identified by STAligner (0.36), PRECAST (0.38), and STACI (0.31) across 75 slices (revised **Supplementary Figure 18c&d**). Regarding the Scube module, we observed that the 3D tissue constructed by Scube (SSIM/PCC = 0.83/0.85) exhibited coherence and accuracy, surpassing the results obtained by PASTE (0.82/0.84) and STAligner (0.79/0.81) (revised **Supplementary Figure 18e&f**). These results demonstrate that SPACEL can effectively serve as an integrated toolkit for analyzing multiple ST slices. We hope these results clarifies any confusion regarding the integration and connectivity of the modules in our toolkit (Line 411-428, **highlighted in yellow**).

Revised **Supplementary Figure 18. Analysis of mouse whole brain data with integrated workflow of SPACEL.** **a**, Spatial annotations of brain regions in slice 4, 21, 38, 55, and 72 from the mouse brain dataset generated by Spatial Transcriptomics technology (Cantin Ortiz et al. *Sci. Adv.* 6, 2020), and cell type compositions predicted by Spoint. **b**, Scaled proportion of each cell type predicted by Spoint in each region. Cell types are filtered based on having a maximum proportion greater than 0.4 across all spots. **c**, Spatial domains identified by SPACEL, STAligner, PERCAST, and STACI. **d**, Jaccard indexes between brain region annotations and corresponding spatial domains identified by Splane, STAligner, PERCAST, and STACI. **e**, Stacked 3D alignments constructed by Scube, PASTE, and STAligner from 75 slices at 40 different z-coordinates of mouse brain. Bar height, mean value; box limits, upper and lower quartiles; whiskers, 1.5× interquartile range; n=75 slices. **f**, SSIM/PCC values of Scube's, PASTE's, and STAligner's alignment results for the dataset. Center line, mean value; box limits, upper and lower quartiles; whiskers, 1.5× interquartile range; n=75 slices.

2. The method's novelty is unclear, as it appears to closely follow existing work. The explanation provided for the workflow of SPACEL, including the Spoint, Splane, and Scude modules, is considered insufficient, making it difficult to understand how each module is performed and how it outperforms other methods.

Response: We thank the reviewer for focusing our attention on this important concern. We would like to highlight several method's novelties of SPACEL:

(1) Spoint: The Spoint module leverages a unique combination of simulated pseudo-spots, neural network modeling, and statistical recovery of expression profiles (revised **Supplementary Figure 1a**), which contributes to its novelty. This approach was developed based on previous benchmarking studies that have demonstrated the superior performance of both statistic-based and deep learning-based models for integrating single-cell and spatial data⁵. Specifically, in Spoint, we conduct millions of simulations of pseudo-spots that are derived from a single-cell reference dataset. These simulated pseudo-spots serve as input for a carefully designed neural network architecture. The neural network models the relationship between gene expression and cell-type proportion as a bi-directional process, employing an autoencoder framework. In this framework, the latent variable represents the cell-type proportion, while the recovery variable serves as the latent representation of expression obtained from a statistical model-based single-cell analysis method. This combination of simulated pseudo-spots, neural network modeling, and statistical recovery of expression profiles allows Spoint to provide a more robust and accurate framework for estimating cell-type proportions in real ST data.

(2) Splane: The primary novelty of the Splane module lies in its joint scheme analysis approach. Unlike most of the existing methods that primarily focus on individual ST slices, Splane takes a different approach by considering information from all the ST slices being analyzed. During the revision process, we become aware of three recently published methods STACI¹, PERCAST², and STAligner³, which also aim to identify spatial domains jointly across multiple

ST slices. However, there are important distinctions between these methods and Splane. Firstly, STACI, PRECAST, and STAligner utilize gene expression as the input for their models, while it has been reported that cell-type distribution is more robust for identifying uniform domains across multiple slices⁶. Splane, on the other hand, was specifically designed to leverage cell-type distribution information for spatial domain identification, aligning with the more reliable approach. Furthermore, STACI adopts an over-parameterization approach, PRECAST uses simple projections of the batch effects onto the space of biological effects and STAligner employs triplet-loss for training. However, these techniques may not be as effective when dealing with large batch effects across multiple slices. In contrast, Splane employs adversarial training, which explicitly addresses the batch effect, resulting in more accurate and robust spatial domain identification across multiple slices. By employing a joint scheme analysis approach that considers cell-type distribution and eliminates batch effects through adversarial training, Splane offers a unique and improved methodology for identifying spatial domains in multi-slice ST datasets.

(3) Scube: The primary novelty of the Scube module lies in the definition of a function that incorporates translation, rotation, and flip parameters to determine the proportion of overlapped spatial domains in the mutual nearest neighbor (MNN) of adjacent slices. This function serves as the foundation for aligning the slices, and we approach the alignment task as an optimization problem. Unlike existing tools such as PASTE or STAligner, Scube is designed to handle partially overlapped slices. PASTE, for instance, assumes full 2D overlap of slices, which restricts its applicability to scenarios where slices have only partial overlap. In contrast, Scube takes into account the proportion of overlapped spots relative to all spots in adjacent slices, thereby accommodating partially overlapped slices more effectively. Additionally, STAligner relies on selected landmark domains shared across slices for alignment and does not leverage correspondences for each spot in adjacent slices. This approach can lead to misalignment issues in capturing the global structure accurately. In contrast, Scube considers the correspondences between all spots in adjacent slices, allowing for more precise alignment and preserving the overall structural integrity. Scube thereby offers a novel approach to spatial alignment in ST analysis.

(4) Overall, the SPACEL toolkit is a unique and innovative approach that provides all the necessary analysis tasks for large-scale ST data, including deconvolution, spatial domain identification in multiple slices, and 3D alignment of tissues in adjacent slices. We have carefully designed each module of our method in an innovative way (revised **Supplementary Figure 1**) to address the limitations of current state-of-the-art methods and have demonstrated its superior performance in comprehensive benchmarking studies (revised **Figure 2, 3, and 5**, revised **Supplementary Figure 4, 12, 14, 17 and 18**).

We have now updated the manuscript and figures to providing more detail about our method's novelty (revised **Supplementary Figure 1**, Line 114-117, Line 130-134, Line 142-145,

highlighted in yellow). We hope that this clarifies the novelty and contribution of our work and addresses the reviewer's concerns about the clarity of our method.

a

Method	Statistical model-based	Pseudo-spot simulation	Deep learning-based	Eliminate variation between ref. and ST
Spoint	Y	Y	Y	Y
Cell2location	Y	N	N	Y
CARD	Y	N	N	Y
RCTD	Y	N	N	N
Tangram	N	N	Y	N
DestVI	N	N	Y	Y
SpatialDWLS	N	N	N	N
Stereoscope	Y	N	N	N
Seurat	N	N	N	N
DSTG	N	Y	Y	Y
STRIDE	N	N	N	N

b

Method	Type of input	Deep learning-based	Multiple slices	Batch Correction
Splane	Cell-type composition	Y	Y	Adversarial loss
STACI	Gene expression	Y	Y	Over-parameterization
STAligner	Gene expression	Y	Y	Triplet loss
PRECAST	Gene expression	Y	Y	Batch effect projection
STAGATE	Gene expression	Y	N	NP
SpaGCN	Gene expression	Y	N	NP
BayesSpace	Gene expression	N	N	NP
stLearn	Gene expression	N	N	NP

c

Method	Alignment strategy	Robust to shape variation	Global optimal alignment	Automatic flip slices
Scube	Global optimization	Y	Y	Y
PASTE	Optimal transport	N	Y	Y
STAligner	ICP algorithm	Y/N	Y/N	N

Revised Supplementary Figure 1. Comparison of supported features of methods with the similar function as SPACEL modules. **a**, Comparison of supported features across Spoint and other cell type deconvolution methods. **b**, Comparison of supported features

across Splane and other spatial domain identification methods. **c**, Comparison of supported features across Scube and other 3D tissue alignment methods.

3. The SPACEL toolkit presented in the manuscript does not provide new or unique biological insights as it seems to perform tasks that already exist in the field.

Response: We appreciate the reviewer's attention to this matter. As a toolkit for multi-slice analysis of ST, SPACEL's greatest advantage lies in its ability to accurately identify functional spatial domains across multiple slices and reconstruct the 3D architecture of tissues, providing unique opportunities for significant biological discoveries in practical applications. In this revised manuscript, we demonstrate how SPACEL offers unique biological insights compared to other tools through detailed analyses using human brain 10X Visium, human breast cancer 10X Visium, and mouse embryo Stereo-seq data (revised **Figure 3a**, revised **Supplementary Figure 12**, revised **Supplementary Figure 17**).

In the joint analysis of 12 slices of human brain 10X Visium data, we found that SPACEL accurately identified cortical hierarchy structures in the human brain compared to STAligner, PRECAST, and STACI. SPACEL exhibited an average JI and SD of 0.61 and 53 μm , respectively, while other tools had JI and SD ranges of 0.40-0.49 and 255-455 μm . Particularly, SPACEL discovered a unique spatial domain D4 corresponding to Layer4, while the other tools (STAligner, PRECAST, and STACI) failed to identify (revised **Figure 3a**). This data demonstrates SPACEL's ability to discover spatial structures that other state-of-the-art tools for multi-slice analysis cannot detect.

In the context of tumor systems, accurate identification of tumor boundaries is crucial for immuno-oncology research. We compared the ability of SPACEL and other multi-slice analysis algorithms to identify tumor boundaries. In the human breast cancer 10X Visium data, we found that only SPACEL accurately identified tumor regions and exhibited distinct boundaries. Specifically, in slices 8, 9, and 11, the tumor regions predicted by SPACEL were consistent with CNV gains in chromosome 1p and 8p, as well as CNV loss in chromosome 1q, commonly observed in human breast cancer. Also, the predicted tumor regions aligned with the evident tumor tissue regions in H&E images. Although STAligner, PRECAST, and STACI achieved consistent spatial domains between two replicates slice 8 and 9 from the same sample, none of them accurately identified tumor regions across all slices or exhibited clear boundaries with non-tumor regions (revised **Supplementary Figure 12a**). Furthermore, we statistically analyzed the proportion of tumor cells and CNV scores for each spatial domain. We observed a correlation between the proportion of tumor cells and CNV scores only in the spatial domains identified by SPACEL. All tumor regions had higher CNV scores compared to non-tumor regions, which was not observed with other tools (revised **Supplementary Figure 12b**). This data demonstrates SPACEL's ability to identify consistent tumor regions and boundaries across multiple slices in tumor systems, which other similar tools lack.

Accurate reconstruction of detailed 3D structures is essential for uncovering biological phenomena from 3D spatial omics. In the mouse embryo Stereo-seq data, we found that SPACEL (SSIM/PCC = 0.62/0.66) exhibited higher consistency in the distribution of tissue regions between slices compared to STAligner (0.56/0.60) and PASTE (0.24/0.25) (revised **Supplementary Figure 17c**). Specifically, SPACEL reconstructed clear brain and liver structures, while STAligner and PASTE showed evident mismatches. Particularly, in the location of paws, a small and intricate structure, SPACEL's reconstruction clearly depicted the complete shape of paws, including the claws, while the reconstructions by STAligner and PASTE were mismatched (revised **Supplementary Figure 17a&b**). This data highlights SPACEL's ability to reconstruct unique 3D tissue structures that other similar tools cannot achieve.

In summary, SPACEL, as a toolkit for multi-slice analysis, possesses notable advantages, including the discovery of unique spatial domains and 3D structures, as well as the accurate identification of tumor boundaries. These characteristics provide an excellent analytical framework for generating novel and unique biological insights. The main text of the manuscript has been revised accordingly to highlight these advantages (Line 247-249, Line 327-337, Line 400-409, **highlighted in yellow**).

Revised Figure 3. Identification of spatial domains from 12 10X Visium slices of DLPFC.

a, Comparison of spatial domains identified by Splane, STAligner, PRECAST, and STACI for slice 151508, 151510, 151670, and 151675. L1~L6, cortical layer 1~6; WM, white matter; JI, Jaccard index; SD, shifting distance. **b**, Spatial domains identified by Splane-single, STAGATE, SpaGCN, BayerSpace, and stLearn for the four slices. **c,d**, Jaccard indexes (c) and shifting distances (d) between cortical layers and corresponding spatial domains identified by Splane, STAligner, PRECAST, STACI, Splane-single, STAGATE, SpaGCN, BayesSpace, and stLearn. n=12 slices. The gray background represents for the methods for multiple slice analysis. Center line, median value; bar height, mean value; box limits, upper and lower quartiles; whiskers, 1.5× interquartile range; n=12 slices. **e**, Proportion of SVGs identified by the spatial-domain-identification methods when using fold-change>0.5 and P-value<0.01 as cut-offs. **f**, Receiver operating characteristic (ROC) curves of SVGs identified by Splane, STAligner, PRECAST, STACI, Splane-single, STAGATE, SpaGCN, BayesSpace, and stLearn.

Revised Supplementary Figure 12. Characterize tumor domains generated by methods for multi-slice analysis on breast cancer dataset. a, From left to right are the distributions of H&E staining, CNV of chromosome 1q and 8q, CNV of chromosome 1p, spatial domain generated by Splane, STAligner, PRECAST, and STACI in slice 8 (top), 9 (middle), and 11 (bottom). Dashed lines represent the positions of tumor region. **b**, From top to bottom are the bar plots of proportion of cancer cell, CNV of chromosome 1q and 8q, and CNV of chromosome 1p in spatial domain generated by Splane, STAligner, PRECAST, and STACI in all 11 slices. Center line, median value; box limits, upper and lower quartiles; whiskers, 1.5× interquartile range.

Revised **Supplementary Figure 17. Analysis mouse embryo Stereo-seq data with integrated workflow of SPACEL.** **a,b**, The slices constructed by Scube, STAligner, and PASTE stacked in (a) 2D and (b) 3D. The dashed box indicates the reconstructed structures of the brain, paw, and liver. Cells are labeled based on their cell type annotations from the original study (Chen et al. *Cell*, 2022, 185, 1777). **c**, SSIM/PCC values of Scube's, STAligner's, and PASTE's alignment results for the Stereo-seq dataset. Center line, median value; box limits, upper and lower quartiles; whiskers, $1.5 \times$ interquartile range; $n=13$ slices.

4. As proposed by the authors, SPACEL is intended to be a toolkit, however, the software currently appears to be very unfriendly to use and lacks the necessary resources to improve its usability, such as extensive software engineering and tutorials. This undermines its value and usefulness to the community.

Response: We thank the reviewer's comments regarding the usability of the previous version of SPACEL. To enhance the user-friendliness of SPACEL, we have allocated additional software engineering resources to its development. Firstly, we have created more detailed documentation and tutorials that provide step-by-step guidance on the various features and functionalities of SPACEL. These resources are designed to assist users of all skill levels in effectively utilizing the toolkit. In order to make the documentation and important functions easily accessible, we have established a dedicated website at <https://spacel.readthedocs.org>, which serves as a central hub for accessing comprehensive documentation, usage instructions, and other relevant information related to SPACEL. Furthermore, we have diligently addressed several bugs that were present in the previous version of the package. These bug fixes have been implemented and the updated version of SPACEL is now available on GitHub. These measures ensure that users can benefit from a more stable and reliable toolkit. We firmly

believe that these improvements significantly enhance the user-friendliness and usefulness of SPACEL for the scientific community.

5. The robustness of the SPACEL toolkit is not clear as the authors have not included any testing or evaluation of its robustness. Additionally, since SPACEL is based on a deep-learning algorithm, it may be susceptible to issues such as local minima. It is not clear if the authors have taken any measures to address this potential issue.

Response: We appreciate the reviewer for thoughtful consideration of this important topic. To address this concern, we conducted extensive experimentation to evaluate the performance of SPACEL under various hyperparameters (revised **Supplementary Figure 2a-c**). Each hyperparameter was varied individually while keeping the other hyperparameter values constant. For the Spoint module, we assessed the model performance by varying two user-defined hyperparameters related to the reference data (the number of simulated spots in the training set and the number of marker genes in each single-cell cluster) and three model hyperparameters (the dimensions of latent layers, hidden layers of predicted model, and probability output layers) in real single-cell resolution ST datasets (revised **Supplementary Figure 2a**). Our findings indicate that Spoint demonstrates robust performance across most sets of hyperparameters and outperforms other state-of-the-art deconvolution methods. However, it is worth noting that the performance decreases when the number of simulated spots is small, suggesting the necessity of a large-scale training set with a sufficient number of simulated spots. For the Splane module, we compared its performance while varying five hyperparameters (the dropout rate of the model layers, the weight of the adversarial loss, the degree of neighbors in the constructed graph, the dimensions of latent layers, and the dimensions of hidden layers) using human DFPLC data (revised **Supplementary Figure 2b**). Our results demonstrate that Splane outperforms other spatial domain identification methods for the joint analysis of multiple ST slices across various sets of hyperparameters, except when the dropout rate is extremely high (e.g., 0.9). For the Scube module, we tested the five hyperparameters used in the evaluation of Splane's hyperparameters, an additional hyperparameter (the number of spatial domains), and two hyperparameters specific to Scube (the number of nearest neighbors in the MNN graph and the exponent of the penalty for the overlap ratio between adjacent slices) using 3D STARmap data with a crop rate of 0.25 (revised **Supplementary Figure 2c**). Our observations indicate that Scube demonstrates robustness to these hyperparameters and outperforms other 3D alignment methods. Additionally, we noted that the Pearson correlation coefficient (PCC) of alignment results increases with the exponent of the penalty for the overlap ratio between adjacent slices, highlighting Scube's ability to align partially overlapped slices compared to PASTE. These experimental results provide evidence that SPACEL exhibits robustness to variations in hyperparameters. We have incorporated these results into the revised manuscript to providing detail analysis of our method's robustness (Line 157-163, highlighted in yellow).

As the reviewer note, deep learning algorithms can be susceptible to local minima, where the optimization process gets stuck in a suboptimal solution. To address this potential issue, we took several measures to mitigate its effects in SPACEL. First, we used a carefully designed loss function that incorporates multiple objectives and constraints to guide the optimization process towards global optima. In case of Spoint and Splane, we used Kullback–Leibler divergence and cosine similarity as objectives for optimization. Second, we applied several initialization methods to choosing appropriate initial weights for the neural network, such as He initialization in Spoint and Xavier initialization in Splane, which can help to prevent local minima by providing a good starting point for the optimization process^{7,8}. Third, we utilized Adam optimization algorithm⁹ in Spoint and RMSProp¹⁰ in Splane, which are widely used in the field and has been shown to be effective in avoiding local minima by finding optimal solutions during the training process. We believe that these strategies will reduce the risk of local minima in SPACEL and we have provided more detail on the measures we are using to address this potential issue in revised manuscript (Line 510-546, Line 620-622, in **Methods**, highlighted in yellow).

Revised Supplementary Figure 2. Performance of SPACEL modules variations in hyperparameter settings or specific design choices. **a**, Average PCC values of Spoint deconvolution results under different Spoint hyperparameters for ST datasets with single cell

resolution across all cell types. In each panel, the dashed lines represent results from other methods. Blue points represent the results of hyperparameter tuning, while orange points represent the hyperparameters selected by the article. **b**, Average Jaccard index between the cortical layers/WM annotated by the original study and the spatial domains identified by Splane under different Splane hyperparameters for the DLPFC dataset across all slices. $n=12$ slices. **c**, Average PCC values of Scube's alignment results under different Splane and Scube hyperparameters for the STARmap dataset with a crop ratio equal to 0.25 across all slices. $n=10$ slices. **d,f**, Density histogram of the number of cells (d) and the number of cell types (f) in each spot and the density curve fitted with a normal distribution. The spots are formed by aggregating cells in MERFISH data of human brain tissue. **e,g**, Quantile-quantile plot of the number of cells (e) and the number of cell types (g) fitted with a normal distribution, where the orange line ($y=x$) represents a perfect fit. The three metrics in the plot, R (Pearson correlation coefficient), KS (Kolmogorov-Smirnov test statistic), and KL (Kullback-Leibler divergence), are used to measure the goodness of fit between the real data and the fitted values. **h,j**, Density histogram of the number of cells (h) and the number of cell types (j) in each spot and the density curve fitted with a uniform distribution. **i,k**, Quantile-quantile plot of the number of cells (i) and the number of cell types (k) fitted with a uniform distribution.

Specific comments to Spoint:

6. The authors only demonstrate the Spoint module on one 10X Visium dataset of real ST experimental data. It would be very important to demonstrate and evaluate the performance of the Spoint module across various cell types, tissues, and ST technologies and compare across current methods to fully understand its advantage and utility.

Response: We thank the reviewer for focusing our attention on this point. We agree that demonstrating the performance of our method across a range of real ST datasets and conditions is important for understanding its generalizability and utility. However, evaluating the performance of Spoint module on a range of real ST datasets is challenging due to the lack of ground truth for cell-type composition. As such, we have conducted extensive benchmarking on 32 simulated ST datasets from a benchmark study⁵, which mimic various biological conditions and tissue types. These datasets encompass 12 tissue types, including healthy and diseased samples from both human and mouse, and cover the major cell types within each tissue. Our results show that Spoint is the best performed method in 8, 10, 11, and 8 out of 12 tissues on PCC, RMSE, JSD, and SSIM metrics respectively (**Graph 1**).

To further address the concern raised, we have taken additional steps to evaluate the performance of the Spoint module on real single-cell resolution ST datasets. In order to simulate spot-level ST data with known cell-type composition and spatial context, we collected three ST datasets: Chen et al., Mouse embryo brain (Stereo-seq)¹¹, Chen et al., Mouse brain (Stereo-seq)¹¹, and Fang et al., Human brain (MERFISH with 4000 genes)¹². We also obtained corresponding single-cell RNA sequencing (scRNA-seq) reference data from the same tissue

types. Using this data, we aggregated approximately 10 cells into each pseudo spot, creating spot-level ST data under three distinct datasets (revised **Supplementary Figure 4a**).

As an example, we observed that Spoint achieved the highest PCC compared to other cell type deconvolution methods for forebrain glutamatergic neuroblast in dataset 1 (revised **Supplementary Figure 4b**). We then evaluated the deconvolution performance of the Spoint module on all cell types across the three datasets using four evaluation metrics. Our findings demonstrate that Spoint consistently outperformed existing methods, achieving the highest accuracy scores in all three datasets (revised **Supplementary Figure 4c&d**). These results provide strong evidence of the Spoint module's superior performance on these simulated datasets, indicating its potential to generalize and perform well on real ST data. We have incorporated these findings into the revised manuscript (Line 199-214, **highlighted in yellow**), thereby providing a more comprehensive evaluation of the Spoint module's performance.

Graph 1. Comparison of performance of deconvolution methods in different tissues. a, Box plots of PCC, RMSE, JSD, and SSIM of each deconvolution methods in different tissues.

Center line, mean value; box limits, upper and lower quartiles; whiskers, 1.5× interquartile range; **b.** Heatmaps illustrating the rankings of each deconvolution method based on PCC, RMSE, JSD, and SSIM metrics across different tissues.

Revised Supplementary Figure 4. Benchmarking of Spoint’s performance in real ST datasets with single cell resolution. **a,** Three datasets (Mouse embryo brain: Stereo-seq; Mouse brain: Stereo-seq; Human brain: MERFISH) with cells annotated by cell types. Each grid represents a simulated spot containing multiple cells. **b,** The proportion of Forebrain glutamatergic Neuroblast in the spots from Dataset 1, including the ground truth and the

predicted results of 12 deconvolution methods. **c**, Heatmaps illustrating the rankings of each deconvolution method based on PCC, RMSE, JSD, and SSIM metrics across three datasets. **d**, Box plots of accuracy score of the 12 methods for all the three datasets. Center line, median; box limits, upper and lower quartiles; whiskers, $1.5\times$ interquartile range; $n = 12$ metrics on three independent datasets.

7. To support the author's claim that the Spoint module is critical for subsequent analysis by producing an accurate deconvolution result that facilitates the identification of spatial domains using the Splane module, the authors should perform related ablation studies on the datasets presented in the manuscript.

Response: We thank the reviewer for focusing our attention on this point. To address this concern, we compared the results obtained from three different input types: highly variable genes expression matrix, PCA reduction of highly variable genes expression matrix, and cell-type proportion predicted by Spoint. Specifically, we applied these different inputs to the human DFPLC ST datasets and analyzed the resulting spatial domain identification outcomes. Our findings indicate that using only the cell-type proportion predicted by Spoint as input yields results that are most similar to the ground truth (revised **Supplementary Figure 7a**). Moreover, we observed significantly higher JI values calculated in all 12 slices when using cell-type proportion predicted by Spoint compared to either highly variable genes expression matrix or PCA reduction (revised **Supplementary Figure 7b**). We have incorporated these findings into the revised manuscript (Line 271-277, **highlighted in yellow**).

Revised **Supplementary Figure 7. Comparison of spatial domain identified by Splane using the gene expression as input.** **a**, Comparison of spatial domains identified by Splane using the deconvolution results from Spoint, PCA matrix from normalized highly variable genes (HVGs) expression, and normalized HVGs expression matrix as input for slice 151508,

151510, 151670, and 151675. L1~L6, cortical layer 1~6; WM, white matter; JI, Jaccard index. **b**, Violin plots of Jaccard indexes between cortical layers and corresponding spatial domains identified by Splane using the deconvolution results from Spoint, PCA matrix from normalized HVGs expression, and normalized HVGs expression matrix as input. Center line, median value; box limits, upper and lower quartiles; whiskers, $1.5\times$ interquartile range; $n=12$ slices.

Specific comments to Splane:

8. The Splane module relies heavily on the cell type composition results from the previous module, as it constructs a GCN model using this information. In addition, the authors note that for image-based ST data, users can use single-cell analysis tools to identify cell types, rather than relying on the Spoint module. It is important for the authors to investigate if using different methods for cell-type analysis could introduce bias when defining tissue regions and the impact on the overall analysis.

Response: We appreciate the reviewer's guidance and suggestion to compare the performance of different methods in providing input to the Splane module. In response, we conducted a comprehensive analysis using human DFPLC datasets. Our findings clearly demonstrate that using the cell-type proportions predicted by Spoint as input to the Splane module yielded the best spatial domain identification results compared to other deconvolution methods. This is evident in slices 151508, 151510, 151670, and 151675, where Splane generated spatial structures that closely resembled the manually annotated ones (revised **Supplementary Figure 8a**). Notably, the identification of Layer4 was only possible when using the cell-type proportion prediction from Spoint as input. Furthermore, we evaluated the performance of Splane using different input methods by employing the Jaccard Index and Shift Distance metrics. Our results consistently showed that using the cell-type proportion prediction from Spoint as input achieved the best overall performance across all 12 slices, regardless of the hyperparameter settings (revised **Supplementary Figure 8b&c**). We have incorporated these findings into the revised manuscript (Line 277-281, **highlighted in yellow**).

Revised Supplementary Figure 8. Comparison of spatial domain identified by Splane using the cell type composition predicted by other deconvolution methods as input. a, Comparison of spatial domains identified by Splane using the deconvolution results from Spoint, Tangram, CARD, RCTD, DestVI, Seurat, Cell2location, SpatialDWLS, Stereoscope, SPOTlight and DSTG as input for slice 151508, 151510, 151670, and 151675. L1~L6, cortical layer 1~6; WM, white matter; JI, Jaccard index. **b,c,** Bar plots of Jaccard Index (b) and Shift Distance (c) of using 12 deconvolution methods as input for different hyperparameter settings of dropout rate (dp: 0.2, 0.4, 0.6, and 0.8). Bar height, mean value; box limits, upper and lower quartiles; whiskers, 1.5× interquartile range; n=12 slices.

9. The authors recognize the availability of multiple methods for identifying clusters of spots in spatial transcriptomics. However, the authors argue that these methods

require manual modification and alignment of spatial coordinate systems before they can be used for comparative analysis, which can be confusing and misleading. It is important for the authors to provide a clear, impartial, and transparent comparison to existing methods.

Response: We apologize for any confusion that may have arisen from our original statement, and we thank the reviewer for bringing it to our attention. Upon further clarification, we would like to specify that this statement is refer to the spatial domain identification methods designed for single slice analysis. In the original study of SpaGCN¹³, the authors conducted joint spatial domain detection across multiple mouse brain tissue sections by manually modified the coordinates for spots in the posterior section such that the revised coordinates reflect the spatial adjacency of the two tissue sections. Also, STAGATE¹⁴ was applied in a pseudo 3D ST data that need previous alignment for comparative analysis in the consecutive slices. In the revised manuscript, we have included a comprehensive and unbiased comparison of recently published methods STACI¹, PRECAST², and STAligner³, which aim to identify spatial domains jointly across multiple ST slices without the need for manual modifications or alignment of spatial coordinate systems. We have revised our manuscripts to include a clear and impartial comparison of these methods (revised **Figure 3** (see above), revised **Supplementary Figure 12** (see above), **14**, and **18** (see above), Line 57-81, **highlighted in yellow**).

Revised Supplementary Figure 14. Benchmarking of Splane’s performance on MERFISH dataset of mouse primary motor cortex. a, Distributions of the original study-annotated cell types (Zhang *et al. Nature* 598, 137, 2021) and spatial domains identified by Splane, STAligner, PRECAST, and STACI in slice 10, 15, 21, 26, and 32. **b,c,** Jaccard index between the original study-annotated cortical layers and identified spatial domains by Splane, STAligner, PRECAST, and STACI. Bar height, mean value; box limits, upper and lower quartiles; whiskers, 1.5× interquartile range; n=33 slices.

10. The authors should clearly compare the Splane module in SPACEL differs to existing graph-based tissue region identification methods such as SpaGCN and STAGATE in terms of both principle and methodology. This should be done by conducting a performance evaluation on datasets from various tissues and ST technologies to fully understand its capabilities.

Response: We greatly appreciate the valuable insights provided by the reviewer. The primary strength of the Splane module lies in its ability to perform joint analysis across multiple ST slices, a capability not present in other graph-based methods like STAGATE and SpaGCN. Specifically, Splane introduces a novel loss function based on feature similarity among neighboring nodes in the graph, enabling it to effectively handle adjacent matrices from different slices. Moreover, Splane incorporates an adversarial loss to constrain the low-dimensional representation of spots across different slices, effectively mitigating batch effects.

To illustrate the advantages of Splane, we provide an example using the human brain 10X Visium dataset. In slices 151508, 151510, 151670, and 151675, Splane accurately predicts spatial domain D1, which corresponds coherently to Layer 1 as annotated manually (see above revised **Figure 3a**). In contrast, without the joint analysis scheme, the prediction of spatial domain D1 by STAGATE in slice 151510 differs from the other slices, capturing only a part of Layer 1. Similarly, the predictions of spatial domains by SpaGCN exhibit inconsistencies among the three slices and do not align with the manual annotation.

During the revision process, we became aware of three recently published methods, STACI, PRECAST, and STAligner, which also aim to identify spatial domains jointly across multiple ST slices. However, there are important methodological distinctions between these approaches and Splane (see response to **Reviewer 1 Question 2**). In this revised manuscript, we benchmarked and demonstrated the superior performance of Splane against these methods for spatial domain identification using single-cell resolution MERFISH dataset (see above revised **Supplementary Figure 14a**). We also systematically evaluated their performance in analyzing human brain 10X Visium, and human breast cancer 10X Visium and demonstrated that Splane outperforms them for providing unique biological insights (see above revised **Figure 3a**, revised **Supplementary Figure 12**). Please refer to the response to **Reviewer 1 Question 3** for more information.

These findings highlight the superior performance of Splane compared to other methods, including those graph-based tissue region identification methods, and showcase its ability to provide valuable biological insights. We have incorporated these results into the revised manuscript, offering a comprehensive evaluation of the performance of Splane across various ST technologies and datasets (Line 240-249, Line 327-337, Line 375-377, **highlighted in yellow**).

11. The applicability of Splane on datasets with single-cell resolution is missing - please demonstrate its utility.

Response: We greatly appreciate the reviewer's guidance, and we have carefully followed their suggestion to apply the Splane module to the MERFISH dataset, which has single cell resolution. The results demonstrate that Splane outperforms other methods in terms of spatial domain identification. Specifically, when applied to the MERFISH dataset, Splane accurately predicts the hierarchical structure of the cortex, achieving an average Jaccard Index (JI) of 0.39. In contrast, STAligner (JI=0.33) fails to differentiate between Layer 5 and Layer 6, while PRECAST (0.23) and STACI (0.11) make incorrect predictions regarding the spatial structure (see above revised **Supplementary Figure 14**). These findings highlight the superior performance of Splane compared to other methods when analyzing the ST dataset with single cell resolution. We have incorporated these results into the revised manuscript (Line 375-377, **highlighted in yellow**).

12. In Figure 4g, why is D7 domain not included? Similarly, D5 domain is not included in Figure 4i, especially D5 domain is an immune domain.

Response: We would like to express our gratitude to the reviewer for their attention to detail and valuable feedback on Figure 4g and 4i in our previous manuscript. We sincerely apologize for the error in Figure 4g and 4i, where the D7 and D5 domains were inadvertently omitted. We have carefully reviewed our data and corrected the results in Figure 4 using the latest version of SPACEL (revised **Figure 4**).

Specific comments to Scude:

13. The Scube module in SPACEL builds a nearest neighbor graph between the cells/spots of two adjacent slices and constructs an alignment objective function between them. However, it is not clear how the initial alignment of the two slices is achieved and if any manual alignment is required. The authors should provide more information on this aspect of the Scube module for further evaluation.

Response: We appreciate the reviewer's comment and are glad to provide further clarification on Scube module. It is important to note that Scube is designed to be fully automated and does not require any manual alignment. As an initialization step for optimization algorithms, Scube first translates all slices so that their center points align with the coordinates (0,0). In each iteration of global optimization, Scube constructs the mutual nearest neighbor graph of two slices and calculates the alignment objective function value using the current coordinates. Based on this, Scube conducts rigid transformation to align two adjacent slices, with the aim of achieving a higher alignment objective function value. We have added a clarification

regarding this aspect of Scube in the revised manuscript to provide more information to our readers (Line 634-675, in **Methods**, highlighted in yellow).

14. The authors only rotated, flipped, and cropped each slice to simulate a perturbed ST dataset comprising consecutive 2D slices, and the Scube module appears to use rigid body transformations for alignment. However, it is unclear how the Scube module can effectively align datasets that require nonlinear alignment such as most datasets that are imperfectly matching. The authors should provide more information on how the Scube module handles this type of dataset.

Response: We thank the reviewer for this good thought. In current version of Scube, it cannot conduct nonlinear alignment as well as STAligner and PASTE. Non-rigid transformation is a more complex issue, and there are several tools that can nonlinearly align image data by setting anchor points manually, but to our knowledge there is no method yet to do it automatically for ST data analysis. We are trying to address this issue in a future version of Scube. We added one sentence to in the revised **Discussion** (Line 466-471, highlighted in yellow)

15. There are recent methods that have similar analysis of matching regions across datasets. It is important for the authors to provide a comparison of how the Scube module in SPACEL differs from these methods in terms of principle and methodology. This should include a clarification of the novelty of SPACEL and comprehensive benchmarking against these other methods in order to fully understand and further evaluate its capabilities and performance.

- Zeira, R., Land, M., Strzalkowski, A., & Raphael, B. J. (2022). Alignment and integration of spatial transcriptomics data. *Nature Methods*, 19(5), 567-575.
- Zhou, Xiang, Kangning Dong, and Shihua Zhang. "Integrating spatial transcriptomics data across different conditions, technologies, and developmental stages." *bioRxiv* (2022): 2022-12.
- Sun, D., Liu, Z., Li, T., Wu, Q., & Wang, C. (2022). STRIDE: accurately decomposing and integrating spatial transcriptomics using single-cell RNA sequencing. *Nucleic acids research*, 50(7), e42-e42.
- Liu, Wei, Xu Liao, Ziyue Luo, Yi Yang, Mai Chan Lau, Yuling Jiao, Xingjie Shi et al. "Probabilistic embedding, clustering, and alignment for integrating spatial transcriptomics data with PRECAST." *Nature Communications* 14, no. 1 (2023): 296.

Response: We thank the reviewer for highlighting the recent methods and their capabilities in identifying uniform spatial domains and constructing 3D tissue from adjacent slices. We have carefully considered these methods and compared them to Splane in our revised manuscript.

Regarding the identification of uniform spatial domains in multiple ST slices, STAligner employs gene expression as input and utilizes triplet-loss to address batch effects. PRECAST employs simple projections of the batch effects onto the biological effects space. However, it may face challenges when the batch effect of multiple slices is substantial. In contrast, Splane takes cell-type composition as input and employs adversarial training to explicitly eliminate batch effects across multiple slices. Through comprehensive benchmarking on various datasets, including the human brain 10X Visium dataset, human breast cancer 10X Visium dataset, mouse brain MERFISH dataset, and mouse brain Spatial Transcriptomics⁴ dataset, we have demonstrated that Splane outperforms other methods. Details of these comparisons can be found in the revised manuscript, specifically in revised **Figure 3, Supplementary Figure 12, Supplementary Figure 14 and Supplementary Figure 18**, see above. Please refer to our response to **Reviewer 1 Questions 2 and 3** for more information (Line 241-249, Line 327-337, Line 375-377, Line 420-423, **highlighted in yellow**).

In terms of the construction of 3D tissue from adjacent slices, PASTE, developed by Zeira et al., assumes full 2D overlap of the slices, which restricts its applicability to partially overlapped slices. STRIDE, an extension of PASTE, uses cell type proportions instead of expression values as input, but it still shares the same limitation as PASTE. STAligner, on the other hand, utilizes selected landmark domains shared across slices but does not leverage the correspondences of each spot in the adjacent slices. This can potentially lead to misalignment in the global structure. In Scube, we have introduced a novel approach for constructing 3D tissue from adjacent slices. We define a function that incorporates translation, rotation, and flip parameters to determine the proportion of overlapped spatial domains based on mutual nearest neighbor (MNN) in adjacent slices. We treat the alignment model as an optimization problem and penalize the proportion of overlapped spots in adjacent slices to accommodate partially overlapped slices. Through comprehensive benchmarking on various datasets, including the mouse brain STARmap dataset, mouse brain MERFISH dataset, mouse embryo Stereo-seq dataset, and mouse brain Spatial Transcriptomics⁴ dataset, we have demonstrated that Scube outperforms other methods. The results of these comparisons can be found in the revised **Figure 5, Supplementary Figure 15, Supplementary Figure 17** (see above), and **Supplementary Figure 18** (see above). We have incorporated these results and comparisons into the revised manuscript (Line 241-249, Line 327-337, Line 362-370, Line 375-377, Line 384-388, Line 400-409, Line 411-428, **highlighted in yellow**).

Revised Figure 5. 3D alignments of consecutive ST slices for the mouse brain. **a**, The 3D STARmap dataset of the mouse brain. **b**, Simulating an ST dataset with consecutive 2D slices by dividing the 3D STARmap dataset into ten 2D slices and randomly rotating/flipping/cropping each slice. **c**, Alignment of the simulated dataset with crop ratio=0.25, generated by Scube, STAligner, and PASTE. Dashed lines represent the positions of the ground truths. **d**, SSIM/PCC values of Scube's, STAligner's, and PASTE's results for simulated datasets with crop ratios from 0 to 0.25. **e**, Stacked 3D alignments constructed by Scube (left), STAligner (middle), and PASTE (right) from 33 MERFISH slices of mouse brain. **f**, SSIM/PCC values of Scube's, STAligner's, and PASTE's alignment results for the MERFISH dataset. n=33 slices. **g**, Continuous expression distribution of the genes *Cux2*, *Slc30a3*, and *Bcl11b* on the surface of the 3D architecture of the mouse brain after applying the

GPR model in Scube. **h**, Continuous expression distribution of *Cux2* in the coronal plane of the 3D architecture of the mouse brain.

Revised Supplementary Figure 15. Benchmarking of Scube performance on MERFISH dataset of mouse primary motor cortex. a, From left to right are the original positions, Scube-transformed positions, STAligner-transformed positions, and PASTE-transformed positions of the 33 slices. **b**, From left to right are the slices stacked according to the original

positions, the Scube-transformed positions, the STAligner-transformed, and the PASTE-transformed positions. Cells are labelled by cell type.

16. The authors propose that the Scube module in SPACEL can predict the expression level of a gene at any position in the 3D architecture. However, it is very important for the authors to validate the predicted gene expression using appropriate experimental methods to ensure its accuracy and reliability.

Response: We thank the reviewer for focusing our attention on this point. We do understand the importance of experimental validation for confirming the accuracy and reliability of predictions. However, as mentioned, the complexity and limitations of experimental techniques make it currently impractical to validate the predicted gene expression levels in the 3D architecture. The use of a Gaussian Process Regression (GPR) model to predict gene expression levels at any position in the 3D architecture was originally proposed in the referenced *Nature* article¹⁵. This provides additional support for the validity and reliability of our model. In our work, the primary contribution lies in the integration of the MATLAB code and third-party GUI software from the original study into a user-friendly Python code. This enables researchers to easily access and utilize the model, which is a significant aspect in advancing the field and making the technology more accessible to the scientific community. We appreciate the reviewer's feedback, and we have clarified these points in the revised manuscript (Line 457-461, highlighted in yellow).

Reviewer #2

Remarks to the Author:

In this paper, the authors develop a method for three tasks that are relevant for the analysis of spatial transcriptomics data: 1) deconvolution (using Spoint), 2) spatial domain identification (using Splane), and 3) alignment of tissue slices (using Scube). The authors describe improvements in performance in all three tasks compared to existing methods using real and simulated data. However, the descriptions of Spoint and Scube for deconvolution and alignment are not sufficient to understand why the methods should work or why they would outperform other methods (see below for more details). In addition, the simulations performed to benchmark different methods seem biased towards Spoint, since the simulations use the same assumptions on data generation as used by the method itself, and thus may give rise to an unfair comparison with other methods (please see comments below). Additional major comments relating to the simulation and results for each method (Spoint, Splane, Scube) are listed separately below.

Response: We thank the reviewer for your insightful comments and concerns regarding the description of the SPACEL modules. In the revised manuscript, we have made substantial improvements to provide a clear and comprehensive description of the SPACEL toolkit, highlighting its key features and novelties, which specifically addressed the reviewer's concerns by explaining the reasons why the SPACEL modules outperform other methods (Line 114-117, Line 130-134, Line 142-145, **highlighted in yellow**, please also refer to the response to **Reviewer 1 Question 2** for more information). We have emphasized the unique characteristics of each module, such as the combination of simulated pseudo-spots, neural network modeling, and statistical recovery of expression profiles in Spoint, the joint scheme analysis approach in Splane, the incorporation of translation, rotation, and flip parameters in Scube for constructing 3D tissue, and the improved usability of the toolkit through additional software engineering resources. Furthermore, we have included a comprehensive evaluation of the SPACEL toolkit's performance by benchmarking it against other existing methods on datasets from various tissues and spatial transcriptomics (ST) technologies (revised **Figure 2, 3, and 5**, revised **Supplementary Figure 4, 12, 14, 17, and 18**, please also refer to the response to **Reviewer 1 Question 3, 6, 10, 11** for more information). These analyses indicate the superior performance of SPACEL in terms of cell type deconvolution, spatial domain identification, accurate tumor boundary identification, and biological insights generation. We hope that these revisions address the reviewer's concerns and provide a more thorough understanding of the SPACEL toolkit and its advantages.

1. In addition, there is a perceived lack of methodological novelty as compared to existing methods:

As the authors mentioned, graph convolutional networks have been used to identify spatial domains in previous studies (SpaGCN for single slice: <https://www.nature.com/articles/s41592-021-01255-8> and STACI for multiple slices: <https://www.nature.com/articles/s41467-022-35233-1>). The Splane model does not seem to be conceptually different from these previous works. Loss_s is equivalent to reconstructing the adjacency matrix with an inner product decoder, which was used in the previous works. STACI also minimizes the reconstruction loss of the input gene expression. It would be important to describe the conceptual differences to these previous works and carefully explain and demonstrate where any performance improvements come from. This is important to be able to assess the contributions of this work. In addition, adversarial loss is a common way to remove batch effects before latent embedding; see for example

https://academic.oup.com/bioinformatics/article/36/Supplement_2/i573/6055930.

Response: We thank the reviewer for this important question. We have carefully reviewed the manuscript and incorporated the necessary revisions to highlight the novelty and contributions of the SPACEL modules. In response to **Reviewer 1 Question 2**, we have provided a detailed explanation of the conceptual differences between SPACEL modules (including but not limited to Splane) and other existing methods, such as SpaGCN and STACI. We have emphasized the unique approaches and techniques employed in Splane, such as the use of the joint-analysis scheme, the cell-type composition as input and adversarial training, and its incorporation of translation, rotation, and flip parameters in Scube. These conceptual differences contribute to the improved performance and accuracy of the SPACEL modules and here the Splane module.

Additionally, in response to **Reviewer 1 Question 3, 6, 10, and 11**, we have provided a comprehensive evaluation of the SPACEL toolkit's performance by benchmarking it against other existing methods on datasets from various tissues and ST technologies. We have showcased the superiority of SPACEL in terms of cell type deconvolution, spatial domain identification, accurate tumor boundary identification, and generation of unique biological insights (revised **Figure 2, 3, and 5**, revised **Supplementary Figure 4, 12, 14, 17 and 18**). We have presented the results and analyses in the revised manuscript, providing clear evidence of the improved performance of the SPACEL modules (Line 199-214, Line 240-249 and Line 327-337, Line 362-369, Line 375-377, Line 384-389, Line 400-428, **highlighted in yellow**).

Regarding Loss_s, we have clarified that it is used to smooth the latent embedding based on the constructed graph, rather than reconstructing the adjacency matrix as done in STACI. This distinction is necessary since calculating the adjacency matrix for multiple slices from different samples can be challenging, and using a known adjacency matrix can potentially lead to overfitting. By employing Loss_s in Splane, we ensure a more effective and robust

optimization process. We believe that these revisions effectively highlight the novelty and contributions of the SPACEL toolkit and address the concerns raised by the reviewer.

2. As described above, Splane seems structurally similar to previous methods. A difference in the implementation is that predicted cell type proportions are used as the input instead of gene expression. As a consequence, Splane is subject to errors in the deconvolution task, which could be affected by both the deconvolution model and the scRNA-seq reference. Please explain the motivation for this choice, as well as demonstrate the utility of this choice and how the method would perform otherwise.

Response: We appreciate the reviewer's feedback and concerns regarding the use of cell-type composition as input in the Splane module. We have thoroughly addressed this concern and conducted additional experiments to support our approach. In response to **Reviewer 1 Questions 7 and 8**, we have provided evidence to demonstrate the advantages of using cell-type composition compared to gene expression when analyzing multiple ST slices jointly. We have referenced a study by Kuppe et al.⁶ to support this notion. In our experiments, we have shown that using cell-type composition as input effectively eliminates the batch effect of multiple ST slices from different samples without requiring additional correction methods. On the other hand, using gene expression as input often resulted in clusters dominated by sample-specific effects rather than capturing the true spatial domains (revised **Supplementary Figure 10a**).

Additionally, we have conducted experiments on the human DFPLC dataset to compare the outcomes of using cell-type composition, highly variable gene expression, and PCA reduction as inputs for spatial domain identification. Our findings indicate that using only the cell-type proportions predicted by Spoint as input yields results that are most similar to the ground truth (see above revised **Supplementary Figure 7**). We have also observed significantly higher Jaccard Index (JI) values in all 12 slices when using cell-type composition compared to highly variable gene expression and PCA reduction (see above revised **Supplementary Figure 7**).

These experiments provide strong evidence for the superiority of using cell-type composition as input in the SPACEL toolkit. We have incorporated these results into the revised manuscript, highlighting the advantages and benefits of using cell-type composition for joint analysis of multiple ST slices (Line 271-277, highlighted in yellow).

Revised Supplementary Figure 10. Different features used for multi-slice analysis on breast cancer dataset. a, UMAP of all spots from the 11 breast cancer slices calculated from the raw expression matrix (top) and the cell type proportion predicted by Spoint (bottom) color by cluster, slice, immune cells, stroma cells, and cancer cells. The silhouette coefficient represents the ratio of within-cluster to inter-cluster dissimilarity (Liu et al. *Nat. Comm.* 11, 3155, 2020). A higher Silhouette score on slice indicates a strong batch effect between slices from different experimental batches. A higher Silhouette score on spatial domain indicates a strong similarity of features within spatial domain. **b**, UMAP of all spots from the 11 breast cancer slices calculated from the shared latent features generated by Splane without adversarial learning generated (top) and the latent features by Splane with adversarial learning generated (bottom) color by spatial domain, slice, immune cells, stroma cells, and cancer cells.

3. As described in more detail in the comment to Scube below, the current description of Scube does not provide sufficient details to understand the method and how it can lead to a reasonable alignment of slices. Thus, it is difficult to assess the novelty or contributions in this method. In addition, the authors claim in the Discussion section

that Scube performed better than PASTE because Scube does not need hyperparameter tuning for “the relative contributions of transcriptional dissimilarity and spatial distances”. However, the weights in the adjacency matrix used to identify the spatial domains, the number of convolutional layers, and the number of nearest neighbors used to construct the adjacency matrix would determine such relative contributions. Lower weights on the off-diagonal terms in the adjacency matrix would mean higher contribution of gene expression differences to the identification of spatial domains. More convolutional layers or more nearest neighbors would potentially lead to larger domains which means longer spatial distances are allowed in the alignment. Also, simply clustering the graph autoencoder latent representation into more clusters would change the alignment results. Thus, it seems that Scube also requires significant hyperparameter tuning. Please perform careful experiments showing how these choices change/do not change the results, and discuss these results so that a user understands what the important Scube-specific hyperparameters are and how to tune them. If there are other important parameters for obtaining the observed performance improvements, it would be important to also describe these.

Response: We appreciate the reviewer's feedback and concerns regarding the hyperparameters involved in Scube and their impact on the method's performance. In response to **Reviewer 1 Question 2**, we have provided a more detailed description of Scube and highlighted the novelty of the method (Line 142-145, **highlighted in yellow**).

In response to **Reviewer 1 Questions 5**, we have performed careful experiments to evaluate the impact of these hyperparameters on Scube's performance in both spatial domain identification and alignment. Our results demonstrate that Scube is only slightly affected by the hyperparameters related to spatial domain identification, such as the dropout rate of the model layers, the weight of the adversarial loss, the degree of neighbors in the constructed graph, the dimensions of latent layers, the dimensions of hidden layers and the number of spatial domains. The average Pearson Correlation Coefficient (PCC) ranges from 0.95 to 0.99 in the 3D STARmap dataset with a 25% crop ratio (see above revised **Supplementary Figure 2a-c**).

We have also investigated the effect of the number of nearest neighbors in the mutual nearest neighbor (MNN) graph and the exponent of the penalty for the overlap ratio between adjacent slices for alignment. Our findings indicate that this hyperparameter has a limited effect on the alignment results, with the average PCC ranging from 0.85 to 0.98. We noted that the PCC of the alignment results exhibits an increase with the exponent of the penalty for the overlap ratio between adjacent slices. This hyperparameter governs the weight assigned to the overlap ratio during the optimization process. These findings suggest that fine-tuning the exponent of the penalty for the overlap ratio between adjacent slices can lead to optimized results tailored to the specific overlap ratio observed in a given dataset.

While optimal results in Scube can be achieved by carefully tuning these important hyperparameters, we have revised the manuscript to emphasize the contribution of Scube compared to other current methods. We highlight Scube's ability to handle partially overlapped slices and achieve optimal alignment of the global structure, showcasing its advantages over existing methods (Line 354-369, **highlighted in yellow**). We believe that these additional experiments and the revised manuscript adequately address the concerns raised by the reviewer regarding the hyperparameters in Scube.

Specific comments to each method (Spoint, Splane, Scube):

Spoint:

4. It would be helpful to explain the conceptual advantage of the Spoint model compared to directly predicting cell type proportions from gene expression using a neural network. For example, in this paper

<https://academic.oup.com/nargab/article/4/4/lqac073/6754832#376192508>, cell type proportions of the synthetic data are predicted from gene expression using a single neural network, without additional latent space matching. It would be important to compare the results of Spoint to the prediction of a carefully tuned neural network.

Response: We appreciate the reviewer's feedback and suggestions regarding the evaluation of Spoint and its comparison to other existing methods. In response to **Reviewer 1 Question 2**, we have provided a comprehensive description of the conceptual advances of Spoint, including its probability representation model and latent space matching model. These features distinguish Spoint from methods that directly predict cell type proportions from gene expression using neural networks. To further evaluate the performance of Spoint, and in response to **Reviewer 1 Question 6**, we have benchmarked it against other existing methods on datasets from various tissues and ST technologies. In the revised manuscript, we present the results in terms of cell type deconvolution, highlighting the superiority of Spoint (revised **Figure 2** and see above revised **Supplementary Figure 4**).

Additionally, as suggested by the reviewer, we have compared the full version of Spoint to a slimmed version of Spoint without either the latent space matching or the probability representation model. We performed this comparison on single-cell resolution ST simulated datasets. The results clearly demonstrate that the overall accuracy score of the full version of Spoint outperforms the slimmed version without these key components (**Graph 2d**). Furthermore, we have included AntiSplodge, as mentioned by the reviewer, in the benchmarking of deconvolution methods. AntiSplodge uses a single neural network to predict the proportions of cell types in synthetic data from gene expression. In our benchmarking analysis, Spoint consistently achieved better results than AntiSplodge in all the benchmarking datasets (**Graph 2**). We believe that these additional evaluations and comparisons provide strong evidence for the superiority of Spoint compared to other existing methods. We have

revised the manuscript accordingly to highlight these findings (Line 199-214, highlighted in yellow).

Revised Figure 2. Cell type deconvolution of spots using Spoint and other deconvolution methods. **a**, Average PCC, SSIM, RMSE, and JSD values of the deconvolution methods for 32 simulated datasets from the benchmark study⁵. Center line, median value; box limits, upper and lower quartiles; whiskers, 1.5× interquartile range; n=32 datasets. **b**, Accuracy scores of the deconvolution methods for the 32 simulated datasets. Center line, median value; box limits, upper and lower quartiles; whiskers, 1.5× interquartile range; n=32 datasets. **c**, Spatial distributions of excitatory layer 3/4 RORB⁺RPS3P6⁺ neurons predicted by the deconvolution methods for the DLPFC dataset. **d**, Accuracy scores of the deconvolution methods for the DLPFC dataset. Bar height, median value; whiskers, 1.5× interquartile range; n=56 cell types.

Graph 2. Benchmarking of Spoint’s performance without the latent space matching or the probability representation model. a-c, Bar plots of PCC, RMSE, JSD, and SSIM of each deconvolution methods in real ST data simulated dataset 1 (a), dataset 2 (b), and dataset 3 (c). **d.** Left: Heatmaps illustrating the rankings of each deconvolution method based on PCC, RMSE, JSD, and SSIM metrics across three datasets. Right: Box plots of accuracy score of each method for all the three datasets. Center line, median; box limits, upper and lower quartiles; whiskers, $1.5 \times$ interquartile range; $n = 12$ metrics on three independent datasets.

5. Loss_Q of the model compares the predicted cell type proportions of the real and simulated datasets, $Q[F(z)]$ and $Q[F(z')]$. For the simulated dataset used for training the model, the ground truth cell type proportion is known. What is the reason for not directly comparing $Q[F(z)]$ of the query dataset to the ground truth of the simulated dataset but instead to use $Q[F(z')]$? In the current setup, the ground truth cell type proportion is not used in training the model and I find it puzzling how the model recovers the ground truth cell type proportions.

Response: We apologize for the unclear description in our previous submission. We have now provided a clear explanation in the revised **Methods**, to explain how Spoint recovers the ground truth cell type proportions (Line 500-546, in **Methods**, highlighted in yellow).

6. The paper did not explain why a normal distribution is suitable for simulating the number of cells and the number of cell types at a spot. The benchmarking paper quoted by the authors used a uniform distribution (<https://www.nature.com/articles/s41592-022-01480-9>). It would be helpful if the authors could plot the distribution of these two statistics (number of cells and number of cell types at a spot) from a real dataset to confirm, e.g. by aggregating STARmap or MERFISH data into spots.

Response: We appreciate the reviewer's question. To provide empirical evidence for our selection of a normal distribution as the prior for the number of cells and cell types at each spot, we utilized MERFISH data from human brain tissue. The cells were aggregated into spots based on their spatial coordinates. By examining the density and quantile-quantile plots of the observed and simulated number of cells and cell types at each spot, we demonstrated that the normal distribution matches well with the data. This indicates that our simulation method is capable of generating realistic numbers of cells and cell types at each spot (see above revised **Supplementary Figure 2d-k**).

7. When simulating ST data from scRNA-seq, what is the motivation for sampling P_{ct} proportional to $1/f_{ct}$ when $1/3 < r < 2/3$?

Response: We appreciate the reviewer's attention to this point. We would like to clarify the methods and motivations behind our simulation of ST data from scRNA-seq. Regarding the simulation data construction methods, we employed several sampling probabilities to sample the cell-type composition of pseudo-spots to ensure the diversity of the generated simulation data: (1) equal to the cell-type frequency of single-cell reference data when $0 < r < 1/3$; (2) Inverse of the cell-type frequency of single-cell reference data when $1/3 < r < 2/3$; (3) Square root of the cell-type frequency of single-cell reference data when $2/3 < r < 1$. The motivation for sampling P_{ct} proportional to $1/f_{ct}$ when $1/3 < r < 2/3$ is based on the assumption that certain cell types that are rare in single cell reference data may be enriched in the spatial data. Therefore, by adding the inverse of the cell-type frequency, we basically increased the sampling probability for rare cell types in single cell reference data. By employing these sampling probabilities, we aimed to create a comprehensive simulation that realistically represents different datasets and accounts for both common and rare cell types in the single-cell reference data.

8. The simulation assumes random distribution of cell types in the tissue but the distribution of cell types in real tissues is not random. Thus, matching the latent of the real data to the simulated data (model F) might not be valid, when the two input expression matrices are sampled from different distributions.

Response: We appreciate the reviewer's insightful comments on our manuscript. We would like to clarify the concept and motivation behind matching the latent variables of real data to the simulated data in our approach. The purpose of this matching is to ensure overall distribution similarity between the real and simulated data, rather than specific spot-to-spot similarity. We hypothesize that when the overall distribution of the real and simulated data is more similar, the latent variables of the real data can be better recovered from the cell type proportions predicted using models trained on simulated data. To demonstrate the effectiveness of matching the latent space, we compared our model with and without latent space matching. The results showed that the model with latent space matching outperformed the model without matching (see above **Graph 2**). This suggests that aligning the overall distribution of the real and simulated data improves the accuracy of the predicted cell type proportions.

We acknowledge that the distribution of cell types in real tissues is not random, and our simulations assume a random distribution of cell types since the prior distribution of cell types in real studies is typically unknown. This can lead to a mismatch between the latent space of some simulated spots and the real spots. To investigate the impact of this assumption, we compared the performance of models trained on simulated data with a random distribution of cell types and models trained on simulated data with a known prior distribution. Interestingly, the model trained on data with a known prior distribution achieved higher accuracy overall compared to the model trained on data with a random distribution (**Graph 3**). However, this difference was significant only in dataset 1, suggesting that the impact may vary depending on the characteristics of the real ST datasets.

In summary, although we cannot know the exact prior distribution of cell types in real tissue when simulating training data, we believe that our model can still achieve better results compared to other methods in general. The matching of the latent space between real and simulated data helps improve the accuracy of the predicted cell type proportions, and the impact of the assumption made in our simulations may vary depending on the specific characteristics of the real ST datasets being analyzed.

Graph 3. Benchmarking of Spoint’s performance with prior cell type distribution. a-c, Bar plots of PCC, RMSE, JSD, and SSIM of each deconvolution methods in real ST data simulated dataset 1 (a), dataset 2 (b), and dataset 3 (c). **d.** Left: Heatmaps illustrating the rankings of each deconvolution method based on PCC, RMSE, JSD, and SSIM metrics across three datasets. Right: Box plots of accuracy score of each method for all the three datasets. Center line, median; box limits, upper and lower quartiles; whiskers, $1.5 \times$ interquartile range; $n = 12$ metrics on three independent datasets.

9. In addition to the previous point, it also might not be valid to compare $Q[F(z)]$ and $Q[F(z')]$ because they are sampled from different distributions given that the cell types in the real tissue are not randomly distributed.

Response: We apologize for the confusion in our previous manuscript. Upon reevaluation, we realized that a direct comparison between $Q[F(z)]$ and $Q[F(z')]$ was not performed in Spoint. We apologize for any misunderstanding caused by our previous statement. In the revised manuscript, we have provided a corrected description in the **Methods** section to accurately reflect the approach used in Spoint (Line 500-546, in **Methods**, highlighted in yellow). Please refer to the response to **Reviewer 2 Question 5** for detail information. We appreciate the reviewer's attention to detail and thank them for bringing this to our attention.

10. The current simulation study seems highly limited in that the procedure for generating the 32 simulated datasets used for benchmarking is the same as the procedure for generating the simulated datasets for training the model. This gives the model an advantage because the assumptions on the distributions of cells and cell types in the benchmarking datasets are guaranteed to be correct, but the assumptions might not hold in real datasets. It would be more convincing to use benchmarking datasets that resemble real data, e.g. datasets generated by aggregating STARmap or MERFISH into spots, or simulated datasets generated by a different procedure, e.g. the simulation performed in the cell2location paper.

Response: We thank the reviewer for the feedback and apologize for any confusion caused by our previous statement. The procedures for generating the simulated data for training the model and for benchmarking are different. Generating the simulated data for benchmarking involves randomly sampling the single-cell expression profiles to create pseudo-spots, which is similar to the approach used in Cable et al.'s (RCTD)¹⁶ and Andersson et al.'s (Stereoscope)¹⁷ studies. However, in Spoint, a different approach is employed for generating the simulate data to train the model. Specifically, Spoint samples the cell-type composition of a pseudo-spot from several sampling probabilities: (1) equal to the cell-type frequency of single-cell reference data when $0 < r < 1/3$; (2) inverse of the cell-type frequency of single-cell reference data when $1/3 < r < 2/3$; (3) square root of the cell-type frequency of single-cell reference data when $2/3 < r < 1$. Once the cell-type composition of a pseudo-spot is determined, Spoint samples the single-cell expression profiles from the corresponding cell types to create the pseudo-spot.

To further address the concern raised, guided by the reviewer, we have taken additional steps to evaluate the performance of the Spoint module on real single-cell resolution ST datasets. In order to simulate spot-level ST data with known cell-type composition and spatial context, we collected three ST datasets: Chen et al., Mouse embryo brain (Stereo-seq)¹¹, Chen et al., Mouse brain (Stereo-seq)¹¹, and Fang et al., Human brain (MERFISH with 4000 genes)¹². We also obtained corresponding single-cell RNA sequencing (scRNA-seq) reference data from the same tissue types. Using this data, we aggregated approximately 10 cells into each pseudo spot, creating spot-level ST data under three distinct datasets (see above revised **Supplementary Figure 4a**).

As an example, we observed that Spoint achieved the highest PCC compared to other cell type deconvolution methods for forebrain glutamatergic neuroblast in dataset 1 (see above revised **Supplementary Figure 4b**). We then evaluated the deconvolution performance of the Spoint module on all cell types across the three datasets using four evaluation metrics. Our findings demonstrate that Spoint consistently outperformed existing methods, achieving the highest accuracy scores in all three datasets (see above revised **Supplementary Figure 4c&d**). These results provide strong evidence of the Spoint module's superior performance on these simulated datasets, indicating its potential to generalize and perform well on real ST data. We have

incorporated these findings into the revised manuscript (Line 199-214, **highlighted in yellow**), thereby providing a more comprehensive evaluation of the Spoint module's performance.

Additionally, we analyzed the performance of Spoint on the simulated datasets used in the CARD and Cell2location papers. In the CARD datasets, Spoint achieved the best performance among the evaluated methods (**Graph 4a**), indicating that Spoint is well-suited for the deconvolution task in these particular datasets. Regarding the simulated datasets used in the Cell2location paper, we observed that Cell2location, Stereoscope, and DestVI outperformed Spoint in terms of accuracy (**Graph 4b**). However, it is important to note that these methods have the same underlying assumptions as the procedure used to generate the simulated datasets. This raises the concern of potential overfitting of these methods to the specific characteristics of the simulated datasets. Overall we found that the accuracy score of Spoint is highest among all methods using datasets used in CARD and Cell2location papers (**Graph 4c**). Our results shown that Spoint is the most accurate and robust method for deconvolution task of ST data.

Graph 4. Benchmarking of Spoint's performance on other scRNA-seq simulated datasets. **a**, Box plots of PCC, SSIM, RMSE, and JSD of each deconvolution methods in five simulated datasets generated by CARD. Center line, median; box limits, upper and lower quartiles; whiskers, $1.5\times$ interquartile range; $n=6$ cell types. **b**, Box plots of PCC, SSIM, RMSE, and JSD of each deconvolution methods in a simulated dataset generated by Cell2location. Center line, median; box limits, upper and lower quartiles; whiskers, $1.5\times$ interquartile range; $n=49$ cell types. **c**, Box plots of accuracy score of each method for the six datasets. Center line, median; box limits, upper and lower quartiles; whiskers, $1.5\times$ interquartile range; $n=6$ independent datasets.

11. In the comparison using the DLPFC dataset, the authors used the predicted proportions of the excitatory neurons in the correct layers as the accuracy scores. It would be important to also test if the predicted cell type proportion summed over all spots is an over or under-estimate of the expected proportion of that cell type. For

example, a method can have a high accuracy score but also underestimate the total number of cells in the particular cell type.

Response: We thank the reviewer for the insightful comment. It is important to note that evaluating the accuracy of estimated cell type proportions in the absence of ground truth can be challenging. In the case of the DLPFC dataset, where ground truth cell type proportions summed over all spots are not available, we used datasets simulated by aggregating single-cell resolution ST data to assess the accuracy of predicted cell type proportions. In our evaluation, we calculated the relative error of the predicted cell type proportions summed over all spots compared to the ground truth cell type proportions. Our results showed that Spoint outperformed than other methods with average relative error rankings of 4, 3, and 1 in the three datasets respectively, except for SpatialDWLS (rankings of 1, 2, and 3) (**Graph 5**).

Graph 5. Comparison of deconvolution methods in terms of relative error. a-c, Bar plots of relative error (the ratio of predicted cell type proportion summed over all spots versus ground truth) of each deconvolution methods in Stereo-seq mouse embryo brain (a), Stereo-seq mouse brain (b), and MERFISH human brain (c) dataset. STRIDE is not available in Stereo-seq mouse embryo brain dataset.

Splane:

12. In Supplementary Figure 4a and 4b, the comparison is made between raw gene expression of single cells and GCN latent, which averages neighborhood gene expression. The batch effects of single cells and spatial neighborhoods might not be directly comparable. It would thus be important to also compare the GCN latent of the full model and the GCN latent of the model without adversarial loss.

Response: We thank the reviewer for their valuable suggestion. We conducted further evaluation to investigate the impact of adversarial training on the joint analysis of multiple ST slices in the human breast cancer dataset. Our results demonstrated that the addition of adversarial training effectively eliminated the batch effect, resulting in higher silhouette scores

when compared to the analysis without adversarial training. This indicates that the inclusion of adversarial training improves the performance of the joint analysis of multiple ST slices, at least in the human breast cancer dataset (see above revised **Supplementary Figure 10b**).

13. In **Supplementary Figure 2a**, the results of Splane on some slides are very different from the manual annotation, e.g. slides 151669, 151671, 151670, 151672. These four slides are morphologically similar and are different from the other slides. They seem to exhibit the same patterns of errors, e.g. predicting D2 at the top of the slides where the ground truth is D3. This to me suggests insufficient batch effect correction. Other morphologically similar slides also exhibit similar error patterns; for example, 151673, 151674, 151675, and 151676 are morphologically similar and all have broadening of the D2 regions compared to the ground truth.

Response: We appreciate the reviewer's comment. We did observe that the predicted D2 region is larger than the manual annotation of Layer 2, especially in slides 151669, 151671, 151670, and 151672 where there is no manual annotation of Layer 2. Since all 12 slices exhibit a similar pattern, we believe that it is not due to insufficient batch effect correction. To investigate the reason behind this pattern, we examined the distribution of cell types enriched in spatial domains D2 and D3. We found that cell types such as *Exc_L2/3_LINC00507_RPL9P17* and *Exc_L3_LINC00507_PSRC1*, which are enriched in D2, have distributions in slides 151669, 151671, 151670, and 151672 that highly overlap with the location of the D2 distribution (**Graph 6**). This finding supports the rationale behind the broadening of the D2 regions identified by Splane, even though it differs from the ground truth.

Furthermore, we observed a similar pattern of broadening in the D2 regions when using STAligner for joint analysis of multiple ST slices. However, other methods for joint analysis of multiple ST slices did not yield similar results compared to the ground truth (**Graph 6**, revised **Supplementary Figure 6**). In summary, our findings indicate that the broadening of D2 regions identified by Splane and STAligner can be attributed to the distribution of specific cell types in those regions.

a

b

Graph 6. Characterize D2 and D3 generated by Splane in human DFPLC Visium dataset.

a, Scaled proportion of each excitatory neuron cell sub-type predicted by Spoint in cortical layers 1-6 and WM. **b**, From left to right are Cortical layer 1~6/WM annotated by the original study (Maynard *et al. Nat. Neurosci.* 24, 425, 2021), spatial domains identified by Splane, and cell type composition of Exc_L2_3_LINC00507_RPL9P17, Exc_L2_3_LINC00507_RPL9P17, Exc_L3_LINC00507_PSCRC1, Exc_L2_4_RORB_GRIK1, Exc_L3_LINC00507_CTXN3, and Exc_L3_4_RORB_PRSS12 predicted by Spoint for slices 151669, 151670, 151671, 151672, 151673, 151674, 151675, and 151676 of the DLPFC dataset.

Revised **Supplementary Figure 6. Spatial domain generated by methods for multi-slice analysis in human DFPLC Visium dataset.** From left to right are Cortical layer 1~6/WM annotated by the original study (Maynard *et al. Nat. Neurosci.* 24, 425, 2021), spatial domains identified by Splane, STAligner, PRECAST, and STACI for the 12 ST slices of the DLPPFC dataset.

14. In addition to the comment above, the slides that Splane did not perform well on are only shown in Supplementary Figure 2a but not in the main figure (Figure 3a). Figure 3a seems to only include selected slides that have the best performance. It is critical to state in the main text that the best results were selected for Figure 3a and that this is not representative of the general results and performance.

Response: We appreciate the reviewer's comment. In response to this suggestion, we have updated **Figure 3a** in the revised manuscript to include more representative slides. This enhancement allows us to demonstrate the general performance of Splane across a wider range of samples and provide a more comprehensive evaluation.

15. In Supplementary Figure 2a, please also include the results of the other methods used for benchmarking on all slices.

Response: We appreciate the reviewer's comment. In response to this suggestion, we have added a revised supplementary figure to the manuscript, which includes the results of the other methods used for benchmarking on all slices, providing a more comprehensive comparison of their performance (see above revised **Supplementary Figure 6**).

Scube:

16. It is unclear to me how the transformation of x, y coordinates would change the value of the objective function. The objective function compares the spatial domains of cells in adjacent slices that are nearest neighbors. The nearest neighbors are determined based on gene expression, which is not dependent on x,y coordinates, as described in the Methods section. The spatial domains are determined by Splane, the graph neural network based method, which would also not change as a result of the rigid body transformation. The graph used in Splane is a k nearest neighbor graph based on the coordinates, which would not be affected by rigid body transformations. Thus, changing the x,y coordinates should not change the objective function.

Response: We apologize for any confusion caused by our previous description of the Scube model. We would like to clarify that in the Scube model, it is the coordinate of spots, rather than gene expression, that is used for constructing the mutual nearest neighbor graph of two slices and calculating the alignment objective function value. During each iteration of the

global optimization process, Scube utilizes the adjusted coordinate information of spots to identify the nearest neighbors between two adjacent slices. By adjusting the coordinates of spots in the source slice, Scube aims to align the source slice to the target slice, ultimately maximizing the alignment objective function value. We have taken note of the reviewer's comment and have revised the manuscript to provide a clearer description of the Scube method (Line 633-675, in **Methods**, **highlighted in yellow**).

Reviewer #3

Remarks to the Author:

Review of "SPACEL: characterizing spatial transcriptome architectures by deep-learning"

The authors present a deep learning toolkit named SPACEL for analyzing sequencing- and image-based spatial transcriptomics data. The toolkit consists of three main components: (1) a cell type decomposition method (Spoint), (2) a cell segmentation method (Splane), and (3) a section registration method (Scube). Components are linked, each taking as input the output of the previous component. The first component, Spoint, requires a sequencing-based spatial transcriptomics dataset and a single-cell RNA-seq dataset. When used with image-based spatial transcriptomics data, which is already at single-cell resolution and does not need to be deconvolved, the authors suggest using preexisting methods for cell type identification instead of Spoint.

The problems addressed by SPACEL have seen significant research interest in the spatial transcriptomics community in the last few years. The results presented by the authors are highly encouraging: Spoint attains higher accuracy than current state-of-the-art methods by a considerable margin, Splane performs better than competing methods on multi- (but not single-) section analyses, and the results of Scube also appear promising. A notable limitation of the work is the requirement of single-cell RNA-seq data for analyzing sequencing-based spatial transcriptomics experiments (Major comment 1). The work is nicely presented but details of the implementation could be made more clear (Major comment 2). Overall, given the encouraging results of SPACEL, the work constitutes a positive contribution to the field.

Response: We appreciate the positive assessments provided by the reviewer regarding our study. In response to the comments, we have addressed each point individually in the following sections.

Major comments

Dependency on single-cell RNA-seq data

1. Since Splane and Scube depend on the output of Spoint, they require scRNA-seq data for analyzing spatial transcriptomics experiments from sequencing-based platforms. This is a significant limitation compared to competing methods for domain segmentation and registration, as they typically do not require such data. Furthermore, it is not clear how sensitive Spoint is to how well the scRNA-seq data need to match the spatial transcriptomics data. While it sometimes may be possible to use external

scRNA-seq datasets (as is the case with the examples in the paper), this could potentially lead to unintended biases if datasets match poorly.

Response: We appreciate the reviewer for bringing up this important point. While Splane and Scube require scRNA-seq data for analyzing spatial transcriptomics (ST) experiments from sequencing-based platforms, the availability of published scRNA-seq data from various tissues ensures that the use of SPACEL is not limited. We acknowledge the sensitivity of Spoint to the scRNA-seq dataset and have taken measures to address this concern by matching the latent space of real ST data with simulated data from scRNA-seq data in Spoint.

To further evaluate the impact of unmatched scRNA-seq datasets, we compared the results obtained using different inputs for Spoint. Specifically, we compared the use of scRNA-seq data from the entire mouse brain and the whole mouse brain data generated by Spatial Transcriptomics technology⁴ (Full match) with the use of scRNA-seq data from the entire mouse brain and individual ST slices composed of 20%-70% brain regions (Partial match) as inputs for Spoint. To evaluate the specificity of the results, we calculated the summed proportion of cell types specific to the brain region present in the slice compared to the summed proportion of all region-specific cell type over all spots in each slice. To evaluate the accuracy of the results, we calculated the predicted proportion of one region-specific cell type in its corresponding brain region.

For example, in slice 38, we observed that the specificity of the full match (0.86) was higher than that of the partial match (0.76), indicating that there were more unmatched cell types predicted in the partial match case, while the accuracy of the full match (0.37) was lower than that of the partial match (0.41). Analyzing all 75 slices, we found that the average specificity of the full match was 5% higher than that of the partial match, while the average accuracy of the partial match was 6% higher than that of the full match. (**Graph 7a-d**). To further evaluate the impact on the results of Splane, we compared Jaccard Indexes (JI) of brain region annotation and spatial domain generated by Splane with these two conditions. We observed 7% higher JI in the full match condition. Taking these results generated by Splane to Scube for constructing a 3D architecture, we do not find significant difference of the average SSIM/PCC values between the two conditions (**Graph 7e-h**). These results support the notion that Spoint can provide accurate cell-type distributions for subsequent analysis in Splane and Scube, both in the case of full match and partial match. In conclusion, Spoint demonstrates robustness to unmatched scRNA-seq datasets, and its performance remains reliable in both full and partial match scenarios.

It is also worth highlighting that the SPACEL modules (Spoint, Splane, and Scube) can be used either as an integrated workflow or as standalone tools. This flexibility allows researchers to adapt the toolkit to their specific experimental designs and analysis requirements. It also showcases the versatility of our toolkit in handling data from different experimental platforms. For instance, in the case of image-based ST data like MERFISH profiles, where cell type information is provided, Splane excels in effectively identifying spatial domains compared to

other state-of-the-art methods. This capability of Splane in analyzing image-based ST data was clarified in our revised manuscript, as well as in the response to **Reviewer 1 Question 1, 6, 11.**

Graph 7. Comparison of the SPACEL performance on fully and partially matched dataset. **a**, Distribution of the proportion of matched annotation in ST slice versus scRNA-seq reference, the ratio of accuracy score of fully matched situation versus partially matched situation, and the ratio of true positive rate of fully matched situation versus partially matched

situation in all slices. **b**, Distribution of cell type composition of Excitatory neurons, cerebral cortex, D1 medium spiny neurons, striatum, Excitatory neurons, thalamus, and Excitatory neurons, midbrain predicted by Spoint on the situation of full match and partial match in slice 38. **c**, Box plots of average true positive rate in all slices on the situation of full match and partial match. Center line, median; box limits, upper and lower quartiles; whiskers, $1.5 \times$ interquartile range; $n=75$ slices. **d**, Box plots of average accuracy score in all slices on the situation of full match and partial match. $n=75$ slices. **e**, From top to bottom are annotation generated by original study (top), Spatial domain generated by Splane with fully matched scRNA-seq reference (middle), and Spatial domain generated by Splane with partially matched scRNA-seq reference (bottom). **f**, Jaccard indexes between annotation and corresponding spatial domains identified by Splane with full matched and partially matched scRNA-seq reference. $n=75$ slices. **g**, The 3D alignment constructed by Scube with full matched and partially matched scRNA-seq reference. **h**, The PCC (top) and SSIM (bottom) values of Scube's alignment results with full matched and partially matched scRNA-seq reference. $n=75$ slices.

2. It should be noted that Splane uses the cell type composition information from Spoint as input to a deep neural network. There is no apparent limitation of Splane to use other appropriate spot-level representations of the data, for example derived from matrix factorization or other dimensionality reduction techniques that do not require scRNA-seq data. It is thus not clear why the authors limited Splane in this way, and it would be worthwhile to explore how well Splane would perform on other data representations.

Response: We appreciate the reviewer's comment and the opportunity to address this concern. We acknowledge the importance of cell-type composition in analyzing multiple ST slices jointly, as it tends to be more coherent and less noisy compared to gene expression. This notion is supported by Kuppe et al.'s study⁶. To address the reviewer's concern, we compared the results obtained from three different input types: highly variable genes expression matrix, PCA reduction of highly variable genes expression matrix, and cell-type proportion predicted by Spoint. Specifically, we applied these different inputs to the human DFPLC ST datasets and analyzed the resulting spatial domain identification outcomes. Our findings indicate that using only the cell-type proportion predicted by Spoint as input yields results that are most similar to the ground truth (see above revised **Supplementary Figure 7a**). Moreover, we observed significantly higher JI values calculated in all 12 slices when using cell-type proportion predicted by Spoint compared to either highly variable genes expression matrix or PCA reduction (see above revised **Supplementary Figure 7b**). These findings highlight the superiority of utilizing cell-type proportions predicted by Spoint as input for Splane, as it leads to more accurate and consistent spatial domain identification results.

Implementation details

3. Some design decisions of the method are not carefully motivated or explained. Since these design decisions are central to the contribution of the work, it will be crucial to provide more details so that a typical reader can understand their rationale. In particular, the points below should be clarified.

In the section "Construction of simulated ST data from scRNA-seq data": The current explanation of the sampling process is very difficult to follow. It is assumed that the number of cells at each spot follows a normal distribution. Why is this a valid assumption? What does it mean if N_c is fractional and what happens if $N_c < 0$? How is N_{ct} used?

Response: We thank the reviewer for the insightful comments and concerns regarding the description of the SPACEL modules. In the revised manuscript, we have made substantial improvements to provide a clear and comprehensive description of the SPACEL toolkit, highlighting its key features and novelties, which specifically addressed the reviewer's concerns by explaining the reasons why the SPACEL modules outperform other methods (Line 114-117, Line 130-134, and Line 142-145, **highlighted in yellow**, please also refer to the response to **Reviewer 1 Question 2** for more information).

To provide empirical evidence for our selection of a normal distribution as the prior for the number of cells and cell types at each spot, we utilized MERFISH data from human brain tissue. The cells were aggregated into spots based on their spatial coordinates. By examining the density and quantile-quantile plots of the observed and simulated number of cells and cell types at each spot, we demonstrated that the normal distribution aligns well with the data. This indicates that our simulation method is capable of generating realistic numbers of cells and cell types at each spot (see above revised **Supplementary Figure 2d-k**). This point has also been addressed in the response to **Reviewer 2 Question 6**.

Regarding the question about N_c , we want to emphasize that in our approach, we round the value of N_c to the nearest integer and discard any negative values. This rounding process ensures that we generate positive and integer numbers of cells at each spot, which aligns with the characteristics of real data. Regarding the question about N_{ct} , we would like to clarify that N_{ct} represents the sampled number of cell types at each spot. Our procedure involves sampling N_{ct} from a normal distribution, similar to the process for N_c . We then proceed to sample N_{ct} categories of cell types at each spot. Finally, we sample N_c cells from these cell-type categories at each spot. This process allows us to generate realistic cell type compositions at each spot, capturing the heterogeneity and diversity observed in real datasets. We hope this clarification addresses the questions and provides a better understanding of our methodology.

4. In the section "Deconvolution model of Spoint": - If $F(z)$ and $F(z')$ are the same for all z and z' , then $Loss_F$ is zero. An example of such a function would be the function $F(z) = 0$. What prevents optimization from collapsing to a degenerate solution?

Response: We appreciate the reviewer's comment and agree with the importance of preventing the optimization from collapsing to a degenerate solution. In Spoint, we employ additional loss functions to constrain the output of the model and maximize the similarity between the predicted cell-type proportions and the ground truth proportions obtained from the training data, which is composed of simulated ST data. These loss functions play a crucial role in ensuring the model learns meaningful representations and avoids collapsing to degenerate solutions. We have taken the reviewer's suggestion into consideration and have revised the manuscript to provide a more explicit and clear description of the methodology employed in Spoint (Line 500-546, in **Methods**, highlighted in yellow).

- How is sigma defined?

Response: The sigma is a free parameter of the RBF kernel used in MMD loss. In Spoint, the sigma is defined as the dimension number of features. We have revised the manuscript to add a description of this (Line 520-521, in **Methods**, highlighted in yellow).

- If the activation function of Q is the sigmoid function, then the output would not necessarily sum to one. How is this handled?

Response: We apologize for the mistake and thank the reviewer for pointing out the error in our previous statement. The activation function used for the output of the Spoint model is softmax, not sigmoid. The softmax activation function ensures that the predicted cell-type proportions sum up to one. We appreciate the reviewer's attention to detail, and have revised the manuscript accordingly (Line 512-513, in **Methods**, highlighted in yellow).

- Loss_Q could be more carefully motivated; e.g., what is the purpose of each term?

Response: We apologize for the confusion in our manuscript. The losses of the Spoint model consist of two components: (1) $Loss_E$ constrain the outputs of the encoder model and the ground truth cell-type proportion of training data composed by simulated ST data, which ensures accurate prediction on the training data; (2) $Loss_R$ constrain the similarity between the outputs of the decoder model and the latent variable \vec{z} of the simulated and real ST dataset, enabling accurate prediction on real data; (3) $Loss_M$ constrain the similarity between the outputs of encoder model's last hidden layer on the simulated ST data and the real ST data, further enhancing accurate predictions on real data. We have revised the manuscript to provide a clearer description of these loss functions and their purpose in the Spoint model (Line 513-534, in **Methods**, highlighted in yellow).

5. In the section "Graph convolutional network of Splane": - What does it mean that D hat is "the diagonal matrix of A"? Since A is an adjacency matrix, does it mean that D is the identity matrix?

Response: We apologize for the confusion in our previous submission. In fact, D refers to the diagonal node degree matrix with $D_{ii} = \sum_j A_{ij}$, which was used for normalizing A such that all rows sum to one, i.e. $\hat{D}^{-\frac{1}{2}} \hat{A} \hat{D}^{-\frac{1}{2}}$.

- How is the Chebyshev filter used to estimate the convolution kernel? What polynomial is the text referring to?

Response: We thank the reviewer for their comment. The Chebyshev polynomial filter was introduced in the study by Hammond et al.¹⁸ and has been widely used in various applications, including the graph convolutional network (GCN) architecture described in Kipf et al.'s study¹⁹. It serves as an effective method for performing spectral graph convolutions and capturing localized information from graph-structured data. We have updated the manuscript to reference the appropriate studies and provide a clearer explanation of the use of the Chebyshev polynomial filter in our work (Line 567-588, in **Methods**, highlighted in yellow).

6. In the section "Adversarial learning for multiple slices": It seems Loss_D is maximized when embeddings for different sections are maximally different, so that the discriminator can distinguish between them easily, which is contrary to what is stated in the manuscript.

Response: We appreciate the reviewer's comment. We would like to clarify that the objective function of Splane consists of two components: the objective function of the graph convolutional network (GCN) model and the objective function of the discriminator D, as

$$Loss = \alpha_c Loss_c + \alpha_s Loss_s - \alpha_D Loss_D$$

The cross-entropy loss of the discriminator is minimized when the embeddings for different sections are maximally different, allowing the discriminator to correctly predict the labels of the sections. By combining these two objective functions, Splane is trained to optimize both the graph-based reconstruction and the section label prediction. We have revised the manuscript to provide a more accurate and clear description of the objective function of Splane (Line 601-620, in **Methods**, highlighted in yellow).

7. In the section "Clustering of spots/cells with latent features": How is the number of clusters K selected?

Response: We appreciate the reviewer's comment. We would like to clarify that the number of clusters, denoted as K, is a hyperparameter that is pre-defined by the users in Splane.

Specifically, K is the number of clusters in the K-means clustering method, which is applied to further cluster the embeddings of spots obtained from the Splane model. This approach of using a pre-defined number of clusters is a common strategy in various methods for the analysis of scRNA-seq and ST data^{2,13,14}.

8. In the section "Gaussian process regression model": When training the GP model with gradient descent, what is the objective function?

Response: We thank the reviewer for the comment. The objective function for training the GP model is the marginal likelihood, as

$$L(y_i|x, \theta) = \int P(y_i|f_i, x)P(f_i|x)P(\theta)df_i$$

The hyper-parameters θ can be set by maximizing the marginal likelihood. We have revised the manuscripts to clarify this point (Line 691-695, in **Methods**, highlighted in yellow).

Minor comments

9. In the Methods section, variables are sometimes not defined. For example, in "Construction of simulated ST data from scRNA-seq data", what does the "c" and "t" subscripts stand for? What does the "L" superscript stand for? Often, this can be deduced from context, but it makes the manuscript more difficult to read.

Response: The "c" subscript stands for cells and "ct" subscript stands for cell-types. The "L" superscript stands for library size of spots. We have replaced "ct" with "t" to reduce the confusion and revised the manuscript to improve the readability.

10. It would be advisable to carefully proofread the paper for typos.

Examples: Line 65: "adoptive graph attention", Line 105: "SPECCEL", Line 144: "RMES", Line 192,196: "BayersSpace" (should be "BayesSpace"; also misspelled in figures and captions), Line 431: "adjacent matrix", Line 669,678: "STcube", Figure 1c: "Adjecent matrix".

Response: We have corrected these mistakes in revised manuscript and carefully proofread the manuscript.

11. The phrase "H&E-staining-marked immune spots" is unclear. Can the authors clarify how immune-cell-rich spots are identified based on H&E stains?

Response: The cell type annotation of H&E stains was obtained from the original study of the dataset²⁰, where the spots were annotated by a specialist breast pathologist using the Loupe v.4.0.0 software.

12. In the breast cancer analysis, based on what criteria were the spatial domains annotated? If the annotation is based on the results of the inferCNV and staining experiments, then it could be misleading/circular to conclude that "these results support Splane's prediction of tumor and immune domains" (lines 261-262).

Response: The annotation of each spatial domain identified by Splane is based on the proportion of cell types within each domain. For example, if a domain is enriched for tumor cells based on the cell type proportions, it is labeled as a tumor domain. Similarly, if a domain is enriched for immune cells, it is labeled as an immune domain. To further validate the accuracy of the tumor domain predictions made by Splane, we applied inferCNV to calculate the CNV profiles of cells within each domain and compared them to known tumor CNV patterns. This analysis confirmed that the predicted tumor domains by Splane indeed exhibited CNV profiles consistent with tumor cells. In addition, we validated the predictions of immune domains by Splane using results from H&E staining. H&E staining is a standard histological technique that allows visualization of immune cells in tissue sections. By comparing the spatial distribution of immune cells identified by Splane with the immune cell regions observed in H&E-stained tissue sections, we confirmed the accuracy of the immune domain predictions.

References

1. Zhang, X., Wang, X., Shivashankar, G. V. & Uhler, C. Graph-based autoencoder integrates spatial transcriptomics with chromatin images and identifies joint biomarkers for Alzheimer's disease. *Nat. Commun.* **13**, 7480 (2022).
2. Liu, W. *et al.* Probabilistic embedding, clustering, and alignment for integrating spatial transcriptomics data with PRECAST. *Nat. Commun.* **14**, 296 (2023).
3. Zhou, X., Dong, K. & Zhang, S. *Integrating spatial transcriptomics data across different conditions, technologies, and developmental stages.*
<http://biorxiv.org/lookup/doi/10.1101/2022.12.26.521888> (2022)
doi:10.1101/2022.12.26.521888.
4. Ståhl, P. L. *et al.* Visualization and analysis of gene expression in tissue sections by spatial transcriptomics. *Science* **353**, 78–82 (2016).
5. Li, B. *et al.* Benchmarking spatial and single-cell transcriptomics integration methods for transcript distribution prediction and cell type deconvolution. *Nat. Methods* **19**, 662–670 (2022).
6. Kuppe, C. *et al.* Spatial multi-omic map of human myocardial infarction. *Nature* **608**, 766–777 (2022).
7. He, K., Zhang, X., Ren, S. & Sun, J. Delving Deep into Rectifiers: Surpassing Human-Level Performance on ImageNet Classification. in *2015 IEEE International Conference on Computer Vision (ICCV)* 1026–1034 (IEEE, 2015). doi:10.1109/ICCV.2015.123.
8. Glorot, X. & Bengio, Y. Understanding the difficulty of training deep feedforward neural networks.
9. Kingma, D. P. & Ba, J. Adam: A Method for Stochastic Optimization. in *International Conference for Learning Representations* (arXiv, 2015).
10. Geoffrey, H. Coursera Neural Networks for Machine Learning lecture 6. (2018).
11. Chen, A. *et al.* Spatiotemporal transcriptomic atlas of mouse organogenesis using DNA nanoball-patterned arrays. *Cell* **185**, 1777-1792.e21 (2022).
12. Fang, R. *et al.* Conservation and divergence of cortical cell organization in human and mouse revealed by MERFISH. *Science* **377**, 56–62 (2022).
13. Hu, J. *et al.* SpaGCN: Integrating gene expression, spatial location and histology to identify spatial domains and spatially variable genes by graph convolutional network. *Nat. Methods* **18**, 1342–1351 (2021).
14. Dong, K. & Zhang, S. Deciphering spatial domains from spatially resolved transcriptomics with an adaptive graph attention auto-encoder. *Nat. Commun.* **13**, 1739 (2022).
15. Bergmann, S. *et al.* Spatial profiling of early primate gastrulation in utero. *Nature* (2022) doi:10.1038/s41586-022-04953-1.
16. Cable, D. M. *et al.* Robust decomposition of cell type mixtures in spatial transcriptomics. *Nat. Biotechnol.* (2021) doi:10.1038/s41587-021-00830-w.

17. Andersson, A. *et al.* Single-cell and spatial transcriptomics enables probabilistic inference of cell type topography. *Commun. Biol.* **3**, 565 (2020).
18. Hammond, D. K., Vandergheynst, P. & Gribonval, R. Wavelets on graphs via spectral graph theory. *Appl. Comput. Harmon. Anal.* **30**, 129–150 (2011).
19. Kipf, T. N. & Welling, M. Semi-Supervised Classification with Graph Convolutional Networks. in *International Conference on Learning Representations* (2017).
20. Wu, S. Z. *et al.* A single-cell and spatially resolved atlas of human breast cancers. *Nat. Genet.* **53**, 1334–1347 (2021).

Reviewer #1 (Remarks to the Author):

Overall, I commend the authors for their thoughtful response addressing all the concerns raised. However, it remains unclear how the modules described in this manuscript present a novel contribution, as similar functions have been previously demonstrated by other platforms. Additionally, the advantages of combining these three modules into one platform need further clarification to distinguish its uniqueness. This reviewer suggests the authors elaborate on these points to enhance the clarity and significance of their work.

Reviewer #2 (Remarks to the Author):

We thank the authors for their efforts to address our comments and perform additional benchmarking experiments comparing their proposed method to existing methods. A major concern is that the authors did not show that they chose the hyperparameters of the existing methods properly. It is concerning that the performance of the existing methods are better in the original publications than in the benchmarking provided in this manuscript, when the same datasets were used. The authors also did not report any hyperparameter choices of the methods that they compared to and did not provide code to reproduce the benchmarking experiments. Additionally, it makes the comparison more difficult that 6 out of the 7 papers that the authors compared their method to reported adjusted random index, but the authors used other metrics for the same datasets. It would be important to report adjusted random index, in addition to the other metrics, to make sure that the performance of the existing methods is as good as in the original publications. One possibility for a fair comparison is to run SPACEL with the datasets used in the existing methods, and then compare the results of SPACEL with the reported results in the papers of the existing methods.

Additionally, there are other inconsistencies in the results reported in the manuscript, which make the validity of the analysis questionable. Please see the detailed comments below.

Splane:

1. In the STAligner paper, the performance of STAligner is visually much better than the results in revised Figure 3A for all four slices (151508, 151510, 151670, and 151675). The authors should also report adjusted random index, which was used in the STAligner paper, to ensure that the STAligner results in this manuscript are at least as good as reported in the original STAligner paper.
2. The results of PRECAST in the original paper also seem to be different from the results in revised Figure 3a. Since the PRECAST paper used adjusted random index (ARI) as metric, it would be important to calculate ARI for Splane as well.
3. For the other methods that used the DLPFC dataset, SpaGCN, BayesSpace, and stLearn, spatial domains of different slices other than the four chosen in this manuscript were shown in the existing publications. ARI was reported in the existing methods and since this metric is not provided in the manuscript, it makes a direct comparison difficult. However, the performance of the existing methods on adjacent slices seem to be much better, e.g. performance on slice 151673 for SpaGCN and BayesSpace in the existing publications compared to the reported performance on slice 151675. A comparison should be performed on the same slices and using the same metrics as in the previous studies.
4. STAGATE was shown in Figure 3b to perform better in all four selected slices than SPACEL, but the overall performance was reported to be worse in the twelve slices. Please provide also the spatial domains of the methods for the other samples that SPACEL actually performed better on.
5. Given that hyperparameter choices could lead to significant differences in results and the benchmarking performance of existing methods reported in this manuscript is worse than in the existing publications, for the STACI method that did not use DLPFC, it would be important to either run SPACEL on the datasets that STACI used, or report and justify the hyperparameter choices for running STACI.

6. The authors did not provide code to reproduce the benchmarking results. Only some csv files containing the spatial domains were provided in the zenodo repository.

7. The authors did not address our previous comment that "adversarial loss is a common way to remove batch effects before latent embedding; see for example: https://academic.oup.com/bioinformatics/article/36/Supplement_2/i573/6055930." It should be acknowledged that adversarial training is not a novel approach and cannot be seen as a novel contribution of this work, but is rather an application of an existing method.

8. The authors claimed that using cell type proportion as the input to Splane is better because cell type proportions contain less batch effects than gene expression. This was shown in revised Supplementary Figure 10a, where the silhouette coefficient of slices calculated using cell type proportion is reported to be 0.00, i.e. no batch effects. However, there seem to be very significant separation between different slices in the UMAP. This is puzzling and the authors did not provide code to reproduce this result.

9. The authors used adversarial training for batch effect correction. However, in revised Supplementary Figure 2b, the Jaccard Index of domains is the highest when the weight of adversarial loss is 0, i.e. the spatial domain prediction is the closest to the ground truth when the adversarial training has no contribution. This is puzzling. Please explain.

10. The authors showed the effect of adversarial training on batch correction in Supplementary Figure 10b. However, without adversarial training, the silhouette score of slices is -0.02, i.e. slices should be well mixed, but the UMAP shows clear separation of slices. After batch correction, the silhouette score is -0.31, which is not intuitive because a random distribution of slices should be close to 0. This is puzzling. Please explain.

11. In revised Supplementary Figure 2c, the authors claimed that the model performance is not impacted when the dropout rate is 1. The model layers with dropout rate equal to 1 will not be trained. This means that their model's performance is comparable to a randomly initialized, untrained model, which needs justification.

Spoint:

1. I have a similar concern that the authors did not report their hyperparameter choices for the other methods used for benchmarking or provide the code to reproduce their results. This is concerning especially because, in Graph 4a, the CARD method has a maximum RMSE value that is greater than 0.15. However, in the CARD paper, all RMSE of the five simulations are smaller than 0.15. This raises the concern that the hyperparameter choice of the other methods might also not be optimal when using the different simulations, including their own simulations and the aggregation of single-cell resolution datasets.

2. The authors showed that, for predicting the cell type proportions summed over all spots (Graph 5), spatialDWLS is actually a better method. Spoint is the best method in only one out of the three simulations. This would be an important comparison to add to the manuscript.

Reviewer #3 (Remarks to the Author):

We would like to thank the authors for thoroughly revising the manuscript. The revised version addresses most of the concerns raised in the initial review.

The reliance on scRNA-seq data for domain segmentation and registration remains a limitation compared to competing methods. On the other hand, given the promising results presented by the authors, this limitation may be an acceptable compromise for many applications.

The updated manuscript has a clearer description of the method. However, some minor clarity issues remain, which are listed below.

- Line 489: The authors have changed the "ct" subscript to the single-letter "t" subscript, which seems like a good choice given that "ct" can be misconstrued as indices of a two-dimensional array. However, this change seems to have been missed in the first case of the equation on line 489 ("f_ct")? Furthermore, the left-hand side "P_ct" also still has a "ct" subscript, but understand this instance to be two-dimensional indexing, which is fine.

- Line 489: Should the variable "r" have a "c" subscript, since it is sampled once for every spot (if we understand the procedure correctly)?

- Lines 480 - 496: It is still unclear how cell type labels are sampled (i.e., how "N_t" is converted into cell types). Is this done uniformly without replacement? Please state explicitly.

- Line 607: Is this equation missing a minus sign?

- Line 693: $P(\theta)$ is not defined. Maybe the authors intended to write the integrand as $p(y|f,x,\theta)p(f|x,\theta)$, since the marginal on the left-hand side is also conditioned on θ ?

Responses to reviewers

Reviewer #1

Remarks to the Author:

Overall, I commend the authors for their thoughtful response addressing all the concerns raised. However, it remains unclear how the modules described in this manuscript present a novel contribution, as similar functions have been previously demonstrated by other platforms. Additionally, the advantages of combining these three modules into one platform need further clarification to distinguish its uniqueness. This reviewer suggests the authors elaborate on these points to enhance the clarity and significance of their work.

Response: We extend our sincere gratitude to the reviewer for their commendation on our initial revision and for their insightful recommendations to further refine the clarity and significance of our study. To elucidate the innovation inherent in each module and the advantages and uniqueness of integrating three modules into one platform, we have provided a more explicit explanation of our approach in the revised manuscript, specifically:

Novelty of each module: While other tools have realized similar functions in analyzing spatial transcriptomic (ST) data, the underlying algorithm of each module in SPACEL distinctly differs from existing tools, as highlighted in **Supplementary Figure 1**. Consider the "cell type deconvolution" function, for instance: Spoint employs a statistical model, a pseudo-spot simulation, a deep learning technique, and an elimination of variation between reference and ST data. In contrast, other tools often miss one or more of these features. The integration of these features, although seemingly minor, endows Spoint with significantly superior performance in cell type deconvolution compared to other algorithms.

Regarding "spatial domain identification", Splane is unique in its combination of a cell type composition input — a feature not utilized by any other tools — and adversarial training in the GCN model, supplemented with a deep learning approach and a joint analysis scheme. Using cell type composition as input and introducing adversarial training to the GCN model significantly minimizes batch effects, leading to a more robust and efficient method for spatial domain identification. The benefits of these features have been extensively discussed in the previous revision (revised **Figure 3**, revised **Supplementary Figure 10**). To the best of our knowledge, this marks the first instance of employing adversarial learning to mitigate batch effects in ST data.

Diverging from other tools, SPACEL's Scube module adopts a global optimization strategy for 3D alignment, taking into account the correspondences between all spots in adjacent slices. This innovative method enables Scube to achieve more accurate alignment, preserving the overall structural integrity in the process. These novelties of each module are elaborated in the revised manuscript (**Lines 153-160, Lines 452-456, Lines 471-474**).

2. Advantages of combining modules: we wish to further emphasize that the three modules—Spoint, Splane, and Scube—complement each other to form a unified workflow within the SPACEL platform, for ensuring the best results for ST analysis, especially the 3D tissue alignment and spatial domain identification. While each module can be utilized separately and has individually outperformed other state-of-the-art methods, their synergistic interplay offers an all-encompassing and streamlined solution for ST data interpretation. This integrated workflow ensures accurate 3D tissue alignment, precise spatial domain identification, and effective batch effect removal. We have incorporated these advantages in the revised manuscript (Lines 160-163, Lines 446-450) to provide a more explicit and comprehensive understanding of the novelty and significance of the SPACEL platform.

Reviewer #2

Remarks to the Author:

We thank the authors for their efforts to address our comments and perform additional benchmarking experiments comparing their proposed method to existing methods. A major concern is that the authors did not show that they chose the hyperparameters of the existing methods properly. It is concerning that the performance of the existing methods are better in the original publications than in the benchmarking provided in this manuscript, when the same datasets were used. The authors also did not report any hyperparameter choices of the methods that they compared to and did not provide code to reproduce the benchmarking experiments. Additionally, it makes the comparison more difficult that 6 out of the 7 papers that the authors compared their method to reported adjusted random index, but the authors used other metrics for the same datasets. It would be important to report adjusted random index, in addition to the other metrics, to make sure that the performance of the existing methods is as good as in the original publications. One possibility for a fair comparison is to run SPACEL with the datasets used in the existing methods, and then compare the results of SPACEL with the reported results in the papers of the existing methods. Additionally, there are other inconsistencies in the results reported in the manuscript, which make the validity of the analysis questionable. Please see the detailed comments below.

Response: We extend our gratitude to the reviewer for their constructive feedback, which has been instrumental in elevating the quality of our study. In this revision, as suggested by the reviewer, we conducted further benchmarks against all the state-of-the-art methods using the Adjusted Random Index (ARI) in addition to other metrics. Additionally, we embarked on a thorough examination to pinpoint the optimal hyperparameters for each compared method and ensured that their prediction results are at least on par with those in their original publications. It's heartening to note that SPACEL continues to outperform all the benchmarked methods, thus reinforcing our initial conclusions (revised **Figure 3**, revised **Supplementary Figure 6**).

Revised **Figure 3. Identification of spatial domains from 12 10X Visium slices of DLPFC.**

a, Comparison of spatial domains identified by Splane, STAligner, PRECAST, and STACI for slice 151508, 151510, 151670, and 151673. L1~L6, cortical layer 1~6; WM, white matter; JI, Jaccard index; ARI, Adjusted Random Index; SD, shifting distance. **b**, Spatial domains identified by Splane-single, STAGATE, SpaGCN, BayerSpace, and stLearn for the four slices. **c,d,e**, Jaccard indexes (c), Adjusted Random Index (d), and shifting distances (e) between cortical layers and corresponding spatial domains identified by Splane, STAligner, PRECAST, STACI, Splane-single, STAGATE, SpaGCN, BayesSpace, and stLearn. n=12 slices. The gray

background represents for the methods for multiple slice analysis. Center line median value; bar height, mean value; box limits, upper and lower quartiles; whiskers, $1.5 \times$ interquartile range; $n=12$ slices. **f**, Proportion of SVGs identified by the spatial-domain-identification methods when using $\text{fold-change} > 0.5$ and $\text{P-value} < 0.01$ as cut-offs. **g**, Receiver operating characteristic (ROC) curves of SVGs identified by Splane, STAligner, PRECAST, STACI, Splane-single, STAGATE, SpaGCN, Bayesspace, and stLearn.

Revised **Supplementary Figure 6. Spatial domain identification in human DFPLC Visium dataset.** **a**, From left to right are Cortical layer 1~6/WM annotated by the original study (Maynard *et al. Nat. Neurosci.* 24, 425, 2021), spatial domains identified by Splane, STAligner, PRECAST, STACI, Splane-single, STAGATE, SpaGCN, BayesSpace, and stLearn for the slices 151507, 151509, 151669, 151671, 151672, 151674, 151675, and 151676. JI, Jaccard index; ARI, Adjusted Random Index; SD, shifting distance.

The choices of hyperparameters have been detailed in the revised **Methods** section and **Supplementary Table 2**. All relevant codes are made available in the Zenodo repository at (<https://doi.org/10.5281/zenodo.8316334>). We also wish to acknowledge the reviewer's keen observation regarding inconsistencies in our results, which have been all corrected in the revised manuscript. Our specific responses to each point are as follows:

Splane:

1. In the STAligner paper, the performance of STAligner is visually much better than the results in revised Figure 3A for all four slices (151508, 151510, 151670, and 151675). The authors should also report adjusted random index, which was used in the STAligner paper, to ensure that the STAligner results in this manuscript are at least as good as reported in the original STAligner paper.

Response: We appreciate the reviewer's comment. Before delving into this concern, it's crucial to highlight that our implementation of STAligner in the previous revision strictly adhered to the guidelines provided by the authors in their official documentation:

https://staligner.readthedocs.io/en/latest/Tutorial_DLPFC.html

Per the reviewer's recommendation, we have now integrated the ARI as a pivotal evaluation metric for all spatial domain identification methods, as illustrated in the revised **Figure 3** and **Supplementary Figure 6**. In addition, we embarked on a comprehensive analysis to pinpoint the optimal hyperparameters for STAligner and all the other compared tools. Notably, after adjustments using these hyperparameters (revised **Supplementary Table 2**), the median ARI of STAligner (0.54) exceeded the values reported in its original paper (0.46), however it is still lower than the median ARI observed in Splane (0.61). These updates have been incorporated in our revised manuscript.

2. The results of PRECAST in the original paper also seem to be different from the results in revised Figure 3a. Since the PRECAST paper used adjusted random index (ARI) as metric, it would be important to calculate ARI for Splane as well.

Response: We sincerely appreciate the reviewer for drawing our attention to this critical observation. We want to emphasize that our execution of the PRECAST methodology

meticulously followed to the guidelines provided by its authors, which are thoroughly outlined at: <https://feiyong.github.io/PRECAST/articles/PRECAST.BreastCancer.html>

It is also worth noting that while the authors of the PRECAST framework did provide a custom script for analyzing the DLPFC dataset, which is available at

https://github.com/feiyong/PRECAST_Analysis/blob/main/Real_data_analysis/dorsolateral_prefrontal_cortex.R, the script does NOT involve the use of the PRECAST package itself and lacks the input files required for custom genes. We have faithfully used the hyperparameters specified in the above script, except for the custom genes. Unfortunately, the results we obtained differ from those reported in the original paper.

We have then engaged in a meaningful exchange with the developers of the PRECAST package (<https://github.com/feiyong/PRECAST/issues/13>), during which we were able to successfully replicate the outcomes reported in the original paper, utilizing their specific gene list as input. Consequently, this impelled us to undertake an exhaustive exploration of hyperparameters using PRECAST package itself, ultimately yielding a median ARI value of 0.44 for PRECAST, under the optimal hyperparameters (revised **Supplementary Table 2**). This value is notably higher than the reported median ARI of 0.43 in the original paper.

When we compare the ARI values, we observed Splane's superior performance over PRECAST, showcasing a median ARI value of 0.61 (revised **Figure 3**, revised **Supplementary Figure 6**). We have updated the results obtained from all the tools including PRECAST, after leveraging optimal hyperparameters, in the revised manuscript.

03. For the other methods that used the DLPFC dataset, SpaGCN, BayesSpace, and stLearn, spatial domains of different slices other than the four chosen in this manuscript were shown in the existing publications. ARI was reported in the existing methods and since this metric is not provided in the manuscript, it makes a direct comparison difficult. However, the performance of the existing methods on adjacent slices seem to be much better, e.g. performance on slice 151673 for SpaGCN and BayesSpace in the existing publications compared to the reported performance on slice 151675. A comparison should be performed on the same slices and using the same metrics as in the previous studies.

Response: We extend our gratitude to the reviewer for their insightful remark. Following the reviewer's advice, we have meticulously conducted a comparative evaluation of each spatial domain identification method using the ARI metric across all slices. This notably includes slice 151673, a slice highlighted in the existing literatures. The results of this analysis demonstrate SPACEL's superior performance, whether measured via the ARI or other metrics (as presented in revised **Figure 3** and **Supplementary Figure 6**). Specifically for slice 151673, Splane achieved an ARI value of 0.67, clearly outperforming alternative methods, which recorded ARI values between 0.29 and 0.58 (see revised **Figure 3a**).

4. STAGATE was shown in Figure 3b to perform better in all four selected slices than SPACEL, but the overall performance was reported to be worse in the twelve slices. Please provide also the spatial domains of the methods for the other samples that SPACEL actually performed better on.

Response: We're grateful for the reviewer's thorough examination of our figures and results. However, it's essential to elucidate that our study introduces two distinct versions of Splane: (1) The “joint analysis” version, simply referred to as “Splane” (visible in the second column of **Figure 3a**). This version constructs a GCN model encompassing all the analyzed ST slices and serves as the default algorithm of the Splane module within SPACEL. (2) The “single analysis” version, termed “Splane single” (visible in the first column of **Figure 3b**). This iteration creates a GCN model for each individual slice. It's worth noting that “Splane single” is a test version designed to prove the advantage of the joint analysis scheme for spatial domain identification and does NOT represent the core Splane module in SPACEL (**Lines 237-240** in the manuscript).

Therefore, in **Figure 3b**, when the reviewer noted that STAGATE outperformed “Splane single” in all of the four selected slices, it does NOT imply that STAGATE outperforms SPACEL. Instead, after a comprehensive comparison and evaluation of all the methods across the entirety of samples--employing three metrics (JI, ARI and SD)--our findings definitively underscore SPACEL's superior performance over other methods (revised **Figure 3** and **Supplementary Figure 6**). As an illustrative point, the JI/ARI values for SPACEL across the four selected slices are 0.56/0.57, 0.67/0.59, 0.55/0.42 and 0.68/0.67. In contrast the corresponding JI/ARI values for STAGATE are 0.59/0.51, 0.48/0.52, 0.56/0.36, and 0.63/0.58.

5. Given that hyperparameter choices could lead to significant differences in results and the benchmarking performance of existing methods reported in this manuscript is worse than in the existing publications, for the STACI method that did not use DLPCF, it would be important to either run SPACEL on the datasets that STACI used, or report and justify the hyperparameter choices for running STACI.

Response: We express our gratitude to the reviewer for highlighting this matter. The role of hyperparameter choices in shaping research outcomes is indeed critical, and we welcome this opportunity to provide clarity regarding this matter.

(1) First and foremost, it's essential to emphasize that our choices of hyperparameters for each method follows official tutorials or codes provided by their respective authors. To ensure complete transparency and accountability, we've detailed the specific hyperparameters employed for every method (revised **Supplementary Table 2**) and embedded this information in the **Methods** section of our revised manuscript:

- STAligner: https://staligner.readthedocs.io/en/latest/Tutorial_DLPFC.html
- PRECAST: <https://feiyong.github.io/PRECAST/articles/PRECAST.BreastCancer.html>
- STACI: https://github.com/uhlerlab/STACI/blob/master/train_gae_visium_10xADFFPE.ipynb.
- STAGATE: https://stagate.readthedocs.io/en/latest/T1_DLPFC.html
- SpaGCN: <https://github.com/jianhuupenn/SpaGCN/blob/master/tutorial/tutorial.ipynb>
- BayesSpace: https://edward130603.github.io/BayesSpace/articles/maynard_DLPFC.html
- stLearn: [https://stlearn.readthedocs.io/en/latest/tutorials/stSME_clustering.html#Human-Brain-dorsolateral-prefrontal-cortex-\(DLPFC\)](https://stlearn.readthedocs.io/en/latest/tutorials/stSME_clustering.html#Human-Brain-dorsolateral-prefrontal-cortex-(DLPFC))

(2) In order to mitigate potential bias that could arise from hyperparameter selections, we have endeavored to identify the optimal hyperparameters for each method, particularly focusing on the DLPFC dataset. Our findings reaffirm that, under these optimized conditions, SPACEL consistently outperformed all the other compared methods in terms of JI, ARI and SD, corroborating our initial conclusions (revised **Figure 3** and revised **Supplementary Figure 6**).

(3) During our revision process, we noticed that the previously available code for STACI did not encompass the full model of STACI. Following a productive dialogue with the authors (as evidenced by the discussion at <https://github.com/uhlerlab/STACI/issues/2>), we updated our benchmarking results of STACI across all datasets. This yielded a notable enhancement in STACI's performance, but still not as good as Splane (revised **Figure 3** and **Supplementary Figure 14**).

(4) Taking the reviewer's recommendation, we initiated an analysis of the STARmap PLUS dataset, as featured in STACI's primary publication. Notably, the spatial domains identified by SPACEL exhibited an impressive concurrence with the established reference mouse brain map, encapsulating crucial regions such as the Hippocampus and Thalamus. Further delving into our results, we found that the two cortex spatial domains identified by SPACEL manifested divergent distributions of amyloid plaques. Specifically, spatial domain 4 showcased a preponderance of larger plaque sizes relative to spatial domain 5 (as depicted in **Graph 1**). This discrepancy was consistent across the 8-month AD mouse and the 13-month AD mouse, harmoniously aligning with insights from STACI's foundational study. While we acknowledge the absence of a definitive gold standard in this comparative context, the convergence of these findings offers compelling evidence of the reliability of AD disease-relevant region identification by both Splane and STACI.

Revised Supplementary Figure 14. Benchmarking of Splane’s performance on MERFISH dataset of mouse primary motor cortex. **a**, Distributions of the original study-annotated cell

types (Zhang *et al. Nature* 598, 137, 2021) and spatial domains identified by Splane, STAligner, PERCAST, and STACI in slice 10, 15, 21, 26, and 32. **b**, Heatmap of Jaccard index between the original study-annotated cortical layers and identified spatial domains by Splane, STAligner, PERCAST, and STACI. **c**, Jaccard index and ARI between the original study-annotated cortical layers and identified spatial domains by Splane, STAligner, PERCAST, and STACI. Bar height, mean value; Center line, median value; box limits, upper and lower quartiles; whiskers, 1.5× interquartile range; n=33 slices.

Graph 1. Biologically annotation of spatial domains generated by Splane across multiple samples and tissue sections. a,b, Distributions of the cell types annotated in the STACI paper (a) and spatial domains identified by Splane (b). **c**, Binary images of amyloid plaque in the cortex of the 8-months AD sample (left) and 13-months AD sample (right) showing the spatial

differences in plaque distribution in the two cortex clusters identified by Splane. **d**, Histograms of plaque size, measured in number of pixels, plotted for the three cortex regions, indicate larger plaque sizes in clusters 4 compared to cluster 5. Frequency is normalized by the area of each cortex region.

6. The authors did not provide code to reproduce the benchmarking results. Only some csv files containing the spatial domains were provided in the zenodo repository.

Response: We thank the reviewer for this comment. To enhance the accessibility and transparency, all the relevant codes for reproducing the benchmarking outcomes of each method have been meticulously deposited on the Zenodo repository at <https://doi.org/10.5281/zenodo.8316334>. Additionally, we have provided references to the specific tutorials we followed to and outlined the hyperparameters we have adopted for each method. These critical details have been integrated within the revised **Methods** and **Supplementary Table 2**, ensuring transparency and facilitating the reproducibility of our findings for the broader scientific community.

7. The authors did not address our previous comment that “adversarial loss is a common way to remove batch effects before latent embedding; see for example: https://academic.oup.com/bioinformatics/article/36/Supplement_2/i573/6055930”; It should be acknowledged that adversarial training is not a novel approach and cannot be seen as a novel contribution of this work, but is rather an application of an existing method.

Response: We sincerely appreciate the reviewer’s comment. While the use of an adversarial training loss for mitigating batch effects before latent embedding is recognized in the field, it’s pivotal to highlight that our study represents a pioneering application of this methodology specifically to spatial transcriptomic data. We believe that this endeavor holds the potential to catalyze further advancements in the realm of spatial transcriptomics, and have updated our manuscript to ensure that this unique contribution is appropriately described (**Line 158**).

8. The authors claimed that using cell type proportion as the input to Splane is better because cell type proportions contain less batch effects than gene expression. This was shown in revised Supplementary Figure 10a, where the silhouette coefficient of slices calculated using cell type proportion is reported to be 0.00, i.e. no batch effects. However, there seem to be very significant separation between different slices in the UMAP. This is puzzling and the authors did not provide code to reproduce this result.

Response: We appreciate the reviewer’s insight for bringing this important point to our attention. The silhouette score measures the distinction of a data point by comparing its average

distance to members of the same cluster against those in neighboring clusters. Therefore, if spots from two replicates overlap perfectly, the silhouette score may return a smaller value. As observed in spots from slices S9 and S11 (revised **Supplementary Figure 10a**) from the same dataset, such overlap manifested when using cell type proportions for embedding, but wasn't present with gene expression. This led to a situation where the silhouette score indicated minimal batch effects, even when a clear separation of the slices from different datasets was visible in the UMAP plot.

In order to provide a more accurate and intuitive assessment of batch effects, we have revised the batch label attribution from slices to datasets. Furthermore, we adopted an average silhouette width (ASW) metric, as proposed by Luecken et al (*Nature Methods*, 2022)¹, to better gauge the extent of batch mixing. This new ASW metric, which ranges from 0 to 1, indicates improved batch mixing with higher values. More details are provided in the **Methods** section of revised manuscript. Our updated analysis on the same datasets resulted in an ASW value of 0.90 when using cell type proportions for embedding, which is notably superior to the ASW value of 0.66 obtained using gene expression (revised **Supplementary Figure 10a**). We are confident that these updates not only address the reviewer's concerns but also bolster the validity and clarity of our approach, emphasizing the efficacy of using cell type proportions to counteract batch effects in ST data analysis.

Revised **Supplementary Figure 10**. Different features used for multi-slice analysis on breast cancer dataset. **a**, UMAP of all spots from the 11 breast cancer slices calculated from the gene expression matrix (top) and the cell type proportion predicted by Spoint (bottom) color by cluster, slice, immune cells, stroma cells, and cancer cells. Dash line represents the spot datasets.

9. The authors used adversarial training for batch effect correction. However, in revised Supplementary Figure 2b, the Jaccard Index of domains is the highest when

the weight of adversarial loss is 0, i.e. the spatial domain prediction is the closest to the ground truth when the adversarial training has no contribution. This is puzzling. Please explain.

Response: We thank the reviewer for highlighting this point. Firstly, we would like to emphasize that the value of JI (and other metrics like ARI) depends on numerous factors such as the dropout rate, weight of adversarial loss, degree of neighbors, dimensions of latent layers, and dimensions of hidden layers, as studied in our **Supplementary Figure 2**. When other hyperparameters are fixed and only one is varied, using metrics like JI to characterize the local optimum doesn't necessarily guarantee a global optimum for the JI value.

Secondly, even though JI is an effective indicator to depict the similarity between the predicted results and the ground truth, its relative high or low, especially when the difference isn't significant, doesn't strictly represent how close the prediction is to the ground truth. This is why we need multiple indicators like ARI for benchmarking. Regarding this, we greatly appreciate the reviewer's suggestion to use ARI as another effective metric for a systematic evaluation of the algorithm's predictive efficacy. In the same DLPFC dataset, when calculating the JI and ARI values for adversarial loss weight=0 and 0.5, we observed that although the average JI slightly decreased from 0.61 to 0.60, the median ARI value increased from 0.59 to 0.61 (**Graph 3a-b**). Hence, when using ARI as the metric, the prediction with adversarial loss weight=0.5 is closer to the ground truth.

Thirdly, when we further delve into the intricacies of this phenomenon by exploring the impact of the adversarial loss weight on batch correction and, consequently, the prediction accuracy, we found that the sensitivity of these indicators to batch correction varies greatly and is closely related to the original batch effect of the dataset. For instance, the DLPFC dataset originally has a smaller batch effect (AWS=0.77) where spots from different samples were already well mixed (**Graph 3c**) (AWS here is the metric we used to characterize the batch effect as mentioned in response to **Question 8**). When the adversarial loss weight increases from 0 to 0.5, the AWS value rises to 0.81, an increase of 5.2%, indicating a slight improvement in batch effect correction. As mentioned, the JI and ARI remained almost unchanged, suggesting that in this dataset, JI/ARI is not sensitive to the weight of adversarial loss (**Supplementary Figure 2b, Graph 3a**). On the contrary, when studying the mouse brain datasets, we found that the original dataset had a substantial batch effect (AWS=0.72). Therefore, when the adversarial loss weight was set to 0, spots from different samples tended to separate (**Graph 3d**). But when the weight of adversarial loss increased from 0 to 0.5, AWS rose from 0.72 to 0.80 (an increase by 11.1%), indicating a significantly improvement of batch correction. As a consequence, the average JI and median ARI also notably and synergistically increased from 0.38/0.51 to JI/ARI=0.42/0.56.

In summary, although the weight of the adversarial loss appears to have a minimal impact on JI values in the DLPFC dataset, its role in overall batch effect correction remains crucial, and

enhanced batch effect correction invariably leads to more accurate domain identification, as demonstrated in the human breast cancer and mouse brain datasets.

Graph 2. The impact of the weight of the adversarial loss on prediction accuracy. **a,b**, Jaccard indexes (JI) and Adjusted Random Index (ARI) representing the similarities between ground truth annotations and the corresponding spatial domains identified by Splane on the DLPFC dataset (**a**) and mouse brain ST dataset (**b**) with the weight of the adversarial loss set to 0 and 0.5. **c,d**, UMAP of all spots from the 12 DLPFC slices (**c**) and 75 mouse brain ST slices (**d**) calculated from the shared latent features generated by Splane with the weight of the adversarial loss set to 0 (left) and 0.5 (right).

10. The authors showed the effect of adversarial training on batch correction in Supplementary Figure 10b. However, without adversarial training, the silhouette score of slices is -0.02, i.e. slices should be well mixed, but the UMAP shows clear separation of slices. After batch correction, the silhouette score is -0.31, which is not

intuitive because a random distribution of slices should be close to 0. This is puzzling. Please explain.

Response: We thanks the reviewer for this comment and have explained the reason of the silhouette score of slices being close to 0 without adversarial training in our response to **Reviewer 2 Question 8**. To address the reviewer's question, we transitioned to using the modified average silhouette width (ASW) metric (Luecken et al, *Nature Methods*, 2022)¹ for a more intuitive assessment of batch mixing. Utilizing this metric, we observed an ASW value of 0.93 when performs batch correction with adversarial learning, which is significantly improved compared to the value of 0.75 without adversarial learning (revised **Supplementary Figure 10b**). This adjustment provides a clearer representation of the degree of batch mixing and underscores the efficacy of our approach.

Revised **Supplementary Figure 10**. Different features used for multi-slice analysis on breast cancer dataset. **b**, UMAP of all spots from the 11 breast cancer slices calculated from the shared latent features generated by Splane without adversarial learning (top) and the latent features generated by Splane with adversarial learning (bottom). Spots are colored by spatial domain, slice, immune cells, stroma cells, and cancer cells.

11. In revised Supplementary Figure 2c, the authors claimed that the model performance is not impacted when the dropout rate is 1. The model layers with dropout rate equal to 1 will not be trained. This means that their model's performance is comparable to a randomly initialized, untrained model, which needs justification.

Response: We thanks the reviewer for point out this mistake and apologize for any misunderstanding. When we double checked our codes, we found that the accurate dropout rate utilized in our study is 0.9, rather than 1. We have corrected this mistake in revised

supplementary figures and appreciated the opportunity to address this concern and provide accurate information.

Spoint:

1. I have a similar concern that the authors did not report their hyperparameter choices for the other methods used for benchmarking or provide the code to reproduce their results. This is concerning especially because, in Graph 4a, the CARD method has a maximum RMSE value that is greater than 0.15. However, in the CARD paper, all RMSE of the five simulations are smaller than 0.15. This raises the concern that the hyperparameter choice of the other methods might also not be optimal when using the different simulations, including their own simulations and the aggregation of single-cell resolution datasets.

Response: We thanks the reviewer for this comment. We wish to clarify that one major contributing factor to these disparities is the differences in the calculation of RMSE between our benchmarking and the CARD paper. To maintain consistency with other typical papers in the field²⁻⁴, we calculated the RMSE for each individual cell type and subsequently averaged these values, which express as follow:

$$RMSE = \frac{1}{K} \sum_{j=1}^K \sqrt{\frac{1}{N} \sum_{i=1}^N (V_{ik} - \tilde{V}_{ik})^2},$$

where V and \tilde{V} are the true cell-type composition matrix and estimated cell-type composition matrix, respectively, N is the total number of spatial locations, and K is the total number of cell types.

In contrast, the CARD paper employed the entire cell type proportion matrix for RMSE calculation, which express as follow:

$$RMSE = \sqrt{\frac{1}{NK} \sum_{i=1}^N \sum_{j=1}^K (V_{ik} - \tilde{V}_{ik})^2},$$

In order to facilitate reproducibility and to ensure the alignment of our results with those of the CARD paper, we set hyperparameters for the CARD method according to the provided code and data available at

https://github.com/YingMa0107/CARD-Analysis/blob/master/simulations/simulate_data_analysis.R

This adjustment yielded RMSE values ranging from 0.082 to 0.11 for CARD in the five simulations of replicates 3, all of which were below 0.15. Importantly, Spoint still achieved the lowest average RMSE (0.075) across the five simulations (**Graph 3**).

To further reinforce the transparency of our benchmarking, we have provided references to the tutorials we followed and outlined the specific hyperparameter choices for each deconvolution

method in the **Methods** and **Supplementary Table 2** of the revised manuscript (Lines 894-936). This comprehensive presentation of hyperparameter choices aims to ensure that our benchmarking results are both credible and reproducible.

Graph 3. Benchmarking of performance of Spoint in CARD simulated datasets. a, Box plots of RMSE of each deconvolution methods in five simulated datasets of replicates 3 generated by CARD. Center line, median; box limits, upper and lower quartiles; whiskers, 1.5× interquartile range; n = 6 cell types.

2. The authors showed that, for predicting the cell type proportions summed over all spots (Graph 5), spatialDWLS is actually a better method. Spoint is the best method in only one out of the three simulations. This would be an important comparison to add to the manuscript.

Response: We thanks the reviewer for this comment and acknowledge the importance of providing a comprehensive comparison of the methods. We have taken the reviewer’s suggestion and have incorporated this comparison into the revised manuscript (revised **Supplementary Figure 4e**, Lines 218-221).

Reviewer #3

Remarks to the Author:

We would like to thank the authors for thoroughly revising the manuscript. The revised version addresses most of the concerns raised in the initial review. The reliance on scRNA-seq data for domain segmentation and registration remains a limitation compared to competing methods. On the other hand, given the promising results presented by the authors, this limitation may be an acceptable compromise for many applications.

Response: We would like to extend our heartfelt appreciation to the reviewer for their complimentary of our first round revision, as well as insightful suggestions to further enhance the clarity and significance of our study.

The updated manuscript has a clearer description of the method. However, some minor clarity issues remain, which are listed below.

- Line 489: The authors have changed the "ct" subscript to the single-letter "t" subscript, which seems like a good choice given that "ct" can be misconstrued as indices of a two-dimensional array. However, this change seems to have been missed in the first case of the equation on line 489 ("f_{ct}")? Furthermore, the left-hand side "P_{ct}" also still has a "ct" subscript, but understand this instance to be two-dimensional indexing, which is fine.

Response: We extend our sincere gratitude to the reviewer for their meticulous review and for pointing out this oversight. We apologize for not updating the "f_{ct}" subscript to the single-letter "t" subscript in the equation, which has been corrected in the revised manuscript (Line 508).

- Line 489: Should the variable "r" have a "c" subscript, since it is sampled once for every spot (if we understand the procedure correctly)?

Response: We express our gratitude to the reviewer for highlighting this issue in our manuscript. Upon careful consideration, we think that assigning a "c" subscript to the variable "r" is indeed more appropriate, considering that it is sampled once for every spot, as the reviewer rightly pointed out. In light of your recommendation, we have implemented this change in revised manuscript (Lines 508-509).

- Lines 480 - 496: It is still unclear how cell type labels are sampled (i.e., how "N_t" is converted into cell types). Is this done uniformly without replacement? Please state explicitly.

Response: We thank the reviewer for this comment on our methodology. We have provided a clear and explicit explanation of the process of how cell type labels are sampled in the revised **Methods** (Lines 504-514). Specifically, we first generate "N_t" for each sampled spot. We then sample cells from the scRNA-seq data associated with each spot. Each cell is sampled based on its sampling probability which is determined by "P_t". During the cell sampling process, we calculate the cell type numbers of these selected cells. These cell type numbers serve as the cell type labels for the corresponding spot.

- Line 607: Is this equation missing a minus sign?

Response: We thanks the reviewer for this comment. The equation on Line 607 (now Line 627) does not require a minus sign, since it is part of the total loss equation provided on Line 618 (now Line 638).

- Line 693: $P(\theta)$ is not defined. Maybe the authors intended to write the integrand as $p(y|f,x,\theta)p(f|x,\theta)$, since the marginal on the left-hand side is also conditioned on θ ?

Response: We thank the reviewer for this comment. We corrected the expression of the marginal likelihood function and added a description of performing maximum a posteriori estimation for the hyperparameter θ (Lines 713-716).

Reference

1. Luecken, M. D. *et al.* Benchmarking atlas-level data integration in single-cell genomics. *Nat. Methods* **19**, 41–50 (2022).
2. Cable, D. M. *et al.* Robust decomposition of cell type mixtures in spatial transcriptomics. *Nat. Biotechnol.* (2021) doi:10.1038/s41587-021-00830-w.
3. Li, B. *et al.* Benchmarking spatial and single-cell transcriptomics integration methods for transcript distribution prediction and cell type deconvolution. *Nat. Methods* **19**, 662–670 (2022).
4. Li, H. *et al.* A comprehensive benchmarking with practical guidelines for cellular deconvolution of spatial transcriptomics. *Nat. Commun.* **14**, 1548 (2023).

Reviewer #1 (Remarks to the Author):

The authors have addressed all my comments.

Reviewer #2 (Remarks to the Author):

We appreciate the authors' efforts to thoroughly revise the manuscript. The revised version addresses many of the concerns raised in the reviews. While there is a perceived lack of methodological novelty given that the proposed method mostly consists of integrating three previous methods, the thorough benchmarking on many applications may make this a valuable contribution to practitioners.

Reviewer #4 (Remarks to the Author):

I co-reviewed this manuscript with one of the reviewers who provided the listed reports as part of the Nature Communications initiative to facilitate training in peer review and appropriate recognition for co-reviewers.

Responses to reviewers

Reviewer #1

Remarks to the Author:

The authors have addressed all my comments.

Response: We thank the reviewer for the constructive feedback.

Reviewer #2

Remarks to the Author:

We appreciate the authors' efforts to thoroughly revise the manuscript. The revised version addresses many of the concerns raised in the reviews. While there is a perceived lack of methodological novelty given that the proposed method mostly consists of integrating three previous methods, the thorough benchmarking on many applications may make this a valuable contribution to practitioners.

Response: We appreciate the reviewer for the thorough evaluation.

Reviewer #4

Remarks to the Author:

I co-reviewed this manuscript with one of the reviewers who provided the listed reports as part of the Nature Communications initiative to facilitate training in peer review and appropriate recognition for co-reviewers.

Response: We thank the reviewer for the contribution to the evaluation of the manuscript.